# STOP! A OUT-OF-DISTRIBUTION PROCESSOR WITH ROBUST SPATIOTEMPORAL INTERACTION

## ABSTRACT

Recently, spatiotemporal graph convolutional networks have attained significant success in spatiotemporal prediction tasks. However, they encounter out-of-distribution (OOD) challenges due to the sensitivity of node-to-node messaging mechanism to spatiotemporal shifts, leading to suboptimal generalization in unknown environments. To tackle these issues, we introduce the **S**patio**T**emporal **O**OD **P**rocessor (STOP), which leverages spatiotemporal MLP channel mixing as its backbone, separately incorporating temporal and spatial elements for prediction. To bolster resilience against spatiotemporal shifts, STOP integrates robust interaction including a centralized messaging mechanism and a graph perturbation mechanism. Specifically, centralized messaging mechanism configures Context Aware Units (ConAU) to capture generalizable context features, constraining nodes to interact solely with ConAU for spatiotemporal feature interaction. The graph perturbation mechanism uses Generalized Perturbation Units (GenPU) to disrupt this interaction process, generating diverse training environments that compel the model to extract invariant context features from these settings. Finally, we customized a spatiotemporal distributionally robust optimization (DRO) to enhance generalization by exposing the model to challenging environments. Through evaluations on six datasets, STOP showcases competitive generalization and inductive learning. The code is available at https://anonymous.4open.science/r/ICLR2025-STOP.

## 1 INTRODUCTION

Spatiotemporal prediction, as a critical task in urban computing, has become a prominent research area, providing valuable insights into future road conditions and enhancing transportation management systems (Xia et al., 2024; Liang et al., 2023; Miao et al., 2024; Zhang et al., 2023). Within the array of models, spatiotemporal graph convolutional networks (STGNNs) have distinguished themselves as a top choice due to their power representation capabilities for graph data.

However, the success of STGNNs hinges on the assumption of independent and identically distributed (IID) training and testing environments. The environment typically comprises two crucial components: spatiotemporal data and the graph structure. This assumption is naturally vulnerable as data temporal distributions and graph structures naturally evolves, such as the introduction of new entities (e.g., sensors or air quality monitoring stations). The temporal shift and structural shift pose the spatiotemporal out-of-distribution (ST-OOD) problem.

We conduct a performance comparison of several advanced STGNNs in both IID and OOD scenarios using LargeST-SD (Liu et al., 2023b) dataset as an example, as shown in Figure 1 (a). The results show that the performance of STGNNs can degrade rapidly when faced with ST-OOD challenges, especially in the case of structural shifts (S-OOD). One potential reason could be their reliance on global node-to-node messaging for spatiotemporal interaction, such as using GCN or Transformer as spatial learners. This implies that the node representations generated depend on message paths (i.e., graph structure) and features of neighboring nodes. As depicted in Figure 1 (b), when these elements change in the testing environment, GCNs trained on a specific distribution may encounter challenges in accurately capturing the updated node representations. This can lead to errors that propagate across the entire graph, ultimately diminishing the accuracy of the overall graph representation. Furthermore, STGNNs commonly employ a stacked architecture with multiple modules to handle

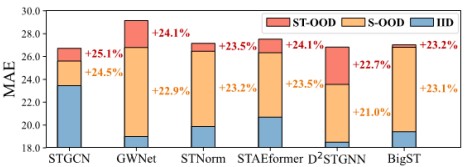 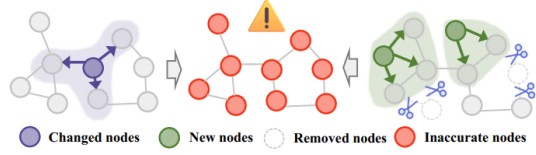

(a) Performance of STGNNs against OOD.  (b) Node-to-node fragility for spatiotemporal shift.

Figure 1: STGNNs in OOD scenarios.

the diverse dimensions of input data, ultimately producing a unique label representation. Each module depends on the precise output of the previous module, and the final prediction is decoded from this unique label representation. This setup not only leads to error accumulation but also shifts in any dimensions could lead to suboptimal predictions. Lastly, the representation capabilities of STGNNs for unseen nodes, i.e., inductive learning, have also been a concern (Zheng et al., 2024).

Several spatiotemporal OOD studies (Xia et al., 2024; Wang et al., 2024a; Ji et al., 2023) have utilized different methods to learn causal knowledge that remains robust in unknown environments. In this paper, we further undertake two efforts: First, we introduce a more robust messaging mechanism for spatiotemporal feature interactions, which can replace the node-to-node messaging. Secondly, we simplify the model structure, and the final prediction is jointly determined by both the temporal and spatial components, no longer relying on a single representation, to reduce error accumulation.

In this paper, we propose a **S**patio**T**emporal **O**OD **P**rocessor (STOP), which utilizes a lightweight MLP as its backbone. Specifically, we incorporate a spatiotemporal channel mixing module with prompt embedding and temporal decomposition techniques to enhance the temporal prediction components. For spatial learning, we introduce a robust Centralized messaging mechanism that configures Context Aware Units (ConAU) to learn invariant contextual features. The Centralized mechanism facilitates interaction between ConAU and nodes, replacing node-to-node messaging. Furthermore, we design Generalized Perturbation Units (GenPU) to perturb this interaction process, simulating spatiotemporal shifts and creating variable training environments. Additionally, we customize a spatiotemporal distributionally robust optimization (DRO) for GenPU to assist models in learning robust causal knowledge from challenging environments. We synthesize the prediction components across temporal and spatial dimensions to generate the final prediction. Our model is evaluated on six datasets, demonstrating robust generalization and inductive learning across various OOD scenarios. Notably, its efficiency is also impressive.

Our contributions can be four-folds: **(1)** We introduce a Spatio-Temporal OOD Processor (STOP), incorporating a resilient centralized messaging mechanism and a graph perturbation mechanism. **(2)** The centralized messaging constrains nodes to interact exclusively with Context Aware Units (ConAU) for feature interaction, thereby enhancing the resilience of model to spatiotemporal shifts. **(3)** The graph perturbation mechanism, equipped with Generalized Perturbation Units (GenPU), disrupts node interactions with ConAU and includes a specialized spatiotemporal distributionally robust optimization (DRO) for GenPU, facilitating the model's acquisition of causal knowledge across diverse environments. **(4)** STOP demonstrates competitive performance on six datasets.

## 2 RELATED WORK

**Spatiotemporal prediction.** As a crucial task in intelligent transportation systems (Li et al., 2024; Jin et al., 2022a;b), current popular spatiotemporal prediction models are predominantly based on spatiotemporal graph neural networks (STGNNs) (Zhang et al., 2016; Wang et al., 2024b;c). These models focus on developing advanced variants to accurately characterize spatiotemporal data, typically combining GCNs with sequential models to learn complex dynamics. For instance, D²STGNN (Shao et al., 2022b) and DCRNN integrate diffusion graph convolutional networks with RNN to effectively capture temporal patterns. Meanwhile, STAEformer (Liu et al., 2023a) and STNN (Yang et al., 2021) utilize Transformer to model long-term temporal dependencies. Some continual learning approaches (Chen et al., 2021) sequentially fine-tune models using data subsets with new distributions to adapt to spatiotemporal changes, which are introduced in Appendix A.

Unfortunately, the effectiveness of these models can only be demonstrated in environments similar to the training set, leading to challenges when encountering OOD scenarios.

**Spatiotemporal OOD learning**. Inspired by advances in time series shift learning (Liu et al., 2022) discussed in Appendix A, researchers have specifically designed spatiotemporal OOD learning models. For example, CauSTG (Zhou et al., 2023) introduces a causal framework that transfers global invariant spatiotemporal relationships to OOD scenarios. CaST (Xia et al., 2023) employs a structural causal model to elucidate the data generation process of spatiotemporal graphs. STONE (Wang et al., 2024a) proposes a causal graph structure to learn robust spatiotemporal semantic relationship. STEVE (Hu et al., 2023) encodes traffic data into two disentangled representations and utilizes spatiotemporal environments as self-supervised signals. In this paper, we reformulate their message-passing mechanism, addressing the OOD challenge from a novel perspective.

## 3 Preliminaries

We use a graph $\mathcal{G} = (\mathcal{V}, \mathbf{A})$ to represent spatiotemporal data, where $\mathcal{V}$ means the node set with $N$ nodes and $\mathbf{A} \in \mathbb{R}^{N \times N}$ is the weighted adjacency matrix of the graph $\mathcal{G}$. We use $X_t \in \mathbb{R}^{N \times c}$ to represent the observed graph signal at time step $t$, where $c$ indicates the number of feature channels.

Training environment $e^*$ is a tuple containing a training graph $\mathcal{G}^* = (\mathcal{V}^*, \mathbf{A}^*)$ and training data $(\mathcal{X}^*, \mathcal{Y}^*)$. With this training environment, spatiotemporal OOD learning aims to learn a robust function $f$, which can accurately predict values after $T_P$ time steps given observed data of past $T$ time steps $\mathbf{X} = [X_1, X_2, \ldots, X_T] \in \mathbb{R}^{T \times N \times c}$ and the graph sampled from any environment $e \sim \mathcal{E}$, where $e$ may have different spatiotemporal distributions with training environment $e^*$,

$$\arg \min_f \sup_{e \in \mathcal{E}} \mathbb{E}_{(\mathbf{X}, \mathbf{Y}) \sim p(\mathcal{X}, \mathcal{Y}|e)} \left[ \mathcal{L} \left( f \left( \mathbf{X} \right), \mathbf{Y} \right) \right], \tag{1}$$

where $\mathbf{Y} = \left[ X_{T+1}, X_{T+2}, \ldots, X_{T+T_p} \right] \in \mathbb{R}^{T_p \times N \times c}$ means the ground-truth value.

## 4 Methodology

STOP solely employs MLP to model temporal and spatial dynamics, with the final prediction jointly determined by temporal and spatial components. It also incorporates a centralized messaging for feature interaction and a graph perturbation mechanism to enhance generalization to unknown environments. The details of STOP are shown in Figure 2 and Algorithm 1.

### 4.1 Temporal modeling and predition

**Temporal decomposition.** In time series analysis, researchers (Cleveland et al., 1990; Wu et al., 2021; Zeng et al., 2023) often decompose time series data into components at various time scales. Some long-term patterns, such as seasonal or periodic trends, are relatively stable, while short-term patterns, like hourly traffic fluctuations, are unstable (Wang et al., 2024b). Intuitively, when the traffic distribution on nodes changes over time, long-term patterns may remain robust. Hence, we employ temporal decomposition techniques to learning causal knowledge in the temporal dimension. Specifically, we use the padding moving average kernel $\mathrm{AvgPool}\left(\cdot; \xi\right)$ with kernel size $\xi$ to decouple the input $\mathbf{X} \in \mathbb{R}^{T \times N \times c}$ into long-term patterns $\mathbf{X}_l$ and short-term patterns $\mathbf{X}_s$:

$$\mathbf{X}_l = \mathrm{AvgPool}\left(\mathbf{X}; \xi\right) \in \mathbb{R}^{T \times N \times c}, \tag{2}$$

$$\mathbf{X}_s = \mathbf{X} - \mathbf{X}_l \in \mathbb{R}^{T \times N \times c}. \tag{3}$$

where we employ padding operation AvgPool in $(\cdot; \xi)$ along temporal dimension, ensuring a consist time length. Subsequently, two distinct $\mathrm{MLP}\left(\cdot\right) : \mathbb{R}^{T \times N \times c} \to \mathbb{R}^{T \times N \times d}$ are leveraged to model the temporal interdependencies within these kinds of patterns. Finally, the outputs are mixed to yield the data representation,

$$\mathbf{H}_0 = \mathrm{MLP}_1\left(\mathbf{X}_l\right) + \mathrm{MLP}_2\left(\mathbf{X}_s\right) \in \mathbb{R}^{T \times N \times d_0}. \tag{4}$$

**Temporal prompt.** To boost the spatiotemporal learning capabilities, we also integrate the prompt learning method, which is a prevalent strategy in the domains of computer vision (Jia et al., 2022)

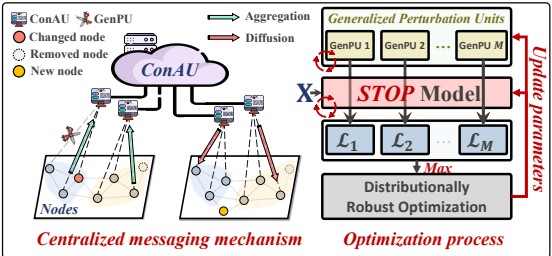

Figure 2: Details of the proposed model. Overall architecture in the left figure. Robust spatiotemporal interaction mechanism in the right figure.

and natural language processing (Vaswani et al., 2017). In this paper, we use a learnable prompt pool $\mathbf{E} \in \mathbb{R}^{(N_t * N_d) \times d_p}$ to encode temporal prior information, which can extract the spatiotemporal invariant patterns recurring on a weekly basis (Yuan et al., 2024). $N_d = 7$ is the number of days in one week. $N_t$ indicates the number of sampling points in a day. For example, for some PeMS datasets, the data sampling frequency of traffic flow is five minutes, so $N_t$ is set to $60 \times 24/5 = 288$. $d_p$ is the dimension of each embedding in the prompt pool.

For the input data $\mathbf{X} \in \mathbb{R}^{T \times N \times c}$, we use $\widetilde{\mathbf{T}} \in \mathbb{R}^{T \times N \times 1}$ to denote its temporal prior information, where $\widetilde{\mathbf{T}}(i) \in \{1, 2, ..., N_t * N_d\}^N$ represents the $i$-th row of $\widetilde{\mathbf{T}}$, indicating the relative position of this $i$-th time step of $\mathbf{X}$ in the total time steps of a week. Then we extract the appropriate embeddings in the prompt pool $\mathbf{E}$ based on this location and generate temporal prior embedding $\mathbf{E}_\mathrm{T}$:

$$\mathbf{E}_\mathrm{T} = \left[ \mathbf{E}\left(\widetilde{\mathbf{T}}(1)\right), \mathbf{E}\left(\widetilde{\mathbf{T}}(2)\right), \ldots, \mathbf{E}\left(\widetilde{\mathbf{T}}(T)\right) \right] \in \mathbb{R}^{T \times N \times d_p}. \tag{5}$$

In addition, we also use the positional embedding $\mathbf{P}$ followed by Transformer (Vaswani et al., 2017) to encode the position of each data point in $\mathbf{X}$. Finally, we integrate temporal prior embedding and data positional embedding to generate the output $\mathbf{Z}_\mathrm{I}$ denoted as the input representation:

$$\mathbf{Z}_\mathrm{I} = \mathrm{Concat}\left(\mathbf{H}_0 + \mathbf{P}, \mathbf{E}\right) \in \mathbb{R}^{T \times N \times (d_0 + d_p)}. \tag{6}$$

**Spatiotemporal channel mixing.** To capture temporal dynamics, we first mix-up the channel and temporal dimensions of the output $\mathbf{Z}_\mathrm{I}$ into shape $N \times d_t$, where $d_t = T * (d_0 + d_p)$. Subsequently, we use $L$ MLP layers for hybrid modeling. Given the input of $l$-th MLP layer with residual connection technology $\mathbf{Z}_\mathrm{T}^{(l)}$, where $\mathbf{Z}_\mathrm{T}^{(0)} = \mathbf{Z}_\mathrm{I}$, the forward process of $l$-th MLP layer is as follows:

$$\mathbf{Z}_\mathrm{T}^{(l+1)} = \mathrm{GELU}\left(\mathbf{Z}_\mathrm{T}^{(l)} \mathbf{W}_1^{(l)} + \mathbf{b}_1^{(l)}\right) \mathbf{W}_2^{(l)} + \mathbf{b}_2^{(l)} + \mathbf{Z}_\mathrm{T}^{(l)} \in \mathbb{R}^{N \times d_t}, \tag{7}$$

where $l \in \{0, 1, ..., L - 1\}$ and $\mathrm{GELU}(\cdot)$ (Hendrycks & Gimpel, 2016) is activation function. $\mathbf{W}_1^{(l)} \in \mathbb{R}^{d_t \times 4d_t}$, $\mathbf{W}_2^{(l)} \in \mathbb{R}^{4d_t \times d_t}$, $\mathbf{b}_1^{(l)} \in \mathbb{R}^{4d_t}$, and $\mathbf{b}_2^{(l+1)} \in \mathbb{R}^{d_t}$ are learnable parameters. After $L$ MLP layers, we get the temporal representation denoted as $\mathbf{Z}_\mathrm{T} = \mathbf{Z}_\mathrm{T}^{(L)} \in \mathbb{R}^{N \times d_t}$. Finally, we use a linear transformation as decoder to generate a temporal prediction component $\mathbf{Y}_t$ as follows,

$$\mathbf{Y}_t = \mathbf{Z}_\mathrm{T} \mathbf{W}_t + \mathbf{b}_t \in \mathbb{R}^{N \times (T_P * c)} \tag{8}$$

where $\mathbf{W}_t \in \mathbb{R}^{d_t \times (T_P * c)}$ and $\mathbf{b}_t \in \mathbb{R}^{T_P * c}$ are learnable parameters.

### 4.2 SPATIAL MODELING AND PREDICTION

#### 4.2.1 CENTRALIZED MESSAGING MECHANISM

STGNN conventionally leverages a global node-to-node messaging mechanism for spatiotemporal feature interactions, which, unfortunately, is vulnerable to structural variations (Finkelshtein et al., 2023; Han et al., 2024b), hindering its generalization capability to unknown graph structures.

To address these limitations, we propose the adoption of a resilient centralized messaging approach that diverges from the traditional node-to-node communication paradigm. Our novel method incorporates context aware units, enabling each graph node to interact solely with these units to gather contextual features, mimicking a centralized messaging manner.

**Context Aware Units.** We first set $K$ context aware units (ConAU), where $K$ is a hyperparameter and $K \ll N$. Then we adopt a learnable feature vector $\boldsymbol{c} \in \mathbb{R}^{d_t}$ for each ConAU, where $d_t$ indicates the number of feature channels. Thus, we can get a series of context feature vectors $\mathbf{C} = [\boldsymbol{c}_1, \boldsymbol{c}_2, \dots, \boldsymbol{c}_K] \in \mathbb{R}^{K \times d_t}$. Next, we propose a multi-head low-rank attention method to achieve the interaction between nodes and ConAU.

**Multi-head Low-rank Attention.** This mechanism consists of two processes: aggregating node features to update context features and diffusing context features to generate node representations. It takes $\mathbf{Z}_{\mathrm{T}} \in \mathbb{R}^{N \times d_t}$ and $\mathbf{C}$ as input. Inspired by the multi-head mechanism (Vaswani et al., 2017), we utilize distinct linear layers to project Query, Key, and Value separately into $d_h = d_t/h$ dimensions with $h$ heads. Specifically, for the $i$-th head where $i = \{1, 2, \dots, h\}$, the calculation of low-rank attention is as follows:

$$\mathbf{Z}_c^{(i)} = \mathcal{A}\left(\mathbf{Q}, \mathbf{K}, \mathbf{V}\right) = \underbrace{\mathrm{softmax}\left(\alpha \mathbf{Q}\mathbf{K}^\top\right)}_{\textbf{Diffusion}} \times \underbrace{\mathrm{softmax}\left(\alpha \mathbf{K}\mathbf{Q}^\top\right)}_{\textbf{Aggregation}} \mathbf{V}, \qquad (9)$$

$$\text{where} \quad \mathbf{Q} = \mathbf{Z}_{\mathrm{T}}\mathbf{W}_q^{(i)} \in \mathbb{R}^{N \times d_h}, \ \mathbf{K} = \mathbf{C}\mathbf{J}_{d_t}^{(i)} \in \mathbb{R}^{K \times d_h}, \ \mathbf{V} = \mathbf{Z}_{\mathrm{T}}\mathbf{J}_{d_t}^{(i)} \in \mathbb{R}^{N \times d_h}. \qquad (10)$$

here $\alpha$ is a scaling factor and equals to $1/\sqrt{d_h}$. $\mathbf{W}_q^{(i)} \in \mathbb{R}^{d_t \times d_h}$ is a learnable parameter matrix, and $\mathbf{J}_{d_t}^{(i)} \in [0, 1]^{d_t \times d_h}$ is a column submatrix of $d_t$-order identity matrix $\mathbf{I}_{d_t} \in [0, 1]^{d_t \times d_t}$, which contains all rows and the columns $(d_h * (i - 1) + 1)$ to $(d_h * i)$ of $\mathbf{I}_{d_t}$. $\mathbf{J}_{d_t}^{(i)}$ is used to project the feature subspace corresponding to the $i$-th head. The computed attention matrix is low-rank with high efficiency, which is explained in Appendix E. Finally, we splice outputs of multiple heads to generate representation for nodes: $\mathbf{Z}_c = \mathrm{Concat}\left(\mathbf{Z}_c^{(1)}, \mathbf{Z}_c^{(2)}, \dots, \mathbf{Z}_c^{(h)}\right) \in \mathbb{R}^{N \times d_t}$.

This attention comprises both aggregation and diffusion processes, as shown in the right half of Figure 2. The aggregation process, denoted by $\mathbf{K}\mathbf{Q}^\top \in \mathbb{R}^{K \times N}$, extracts node features for updating context features. Conversely, the diffusion process, denoted by $\mathbf{Q}\mathbf{K}^\top \in \mathbb{R}^{N \times K}$, disperses the context features to individual nodes to facilitate feature interaction and node representation generation.

**Robustness Analysis.** The proposed centralized messaging mechanism is constrained to operate between nodes and ConAU, effectively avoiding the complexity associated with direct node-to-node interactions. ConAU in this mechanism assimilates contextual features, which is used to generate output representations for individual nodes. These features are coarse-grained and high-level, which exhibits resilience to temporal variations for individual nodes. Furthermore, structural changes (such as adding or removing nodes) do not significantly disrupt the message-passing pathways between nodes and ConAU. New nodes can also leverage these contextual features to develop information-rich representations, thereby enhancing inductive learning capabilities. In summary, our approach demonstrates remarkable resilience to spatiotemporal variations and strong in OOD environments.

### 4.2.2 SPATIOTEMPORAL CHANNEL MIXING

Following the acquisition of context features for each node, we proceed to refine personalized features for individual nodes to enhance the overall node representation. This refinement involves subtracting the context features from the temporal representations to isolate the personalized feature representation of each node, denoted as $\mathbf{Z}_p$, as depicted below:

$$\mathbf{Z}_p = \mathbf{Z}_{\mathrm{T}} - \mathbf{Z}_c \in \mathbb{R}^{N \times d_t}. \qquad (11)$$

Subsequently, we concatenate the decoupled context features $\mathbf{Z}_c$ and personalized features $\mathbf{Z}_p$, and then linearly map them back to the initial representation.

$$\mathbf{Z}_t' = \mathrm{GELU}\left(\mathrm{Concat}\left(\mathbf{Z}_p, \mathbf{Z}_c\right)\mathbf{W}_1 + \mathbf{b}_1\right)\mathbf{W}_2 + \mathbf{b}_2 \in \mathbb{R}^{N \times d_t}, \qquad (12)$$

$$\widetilde{\mathbf{Z}}_t = \mathrm{LayerNorm}\left(\mathbf{Z}_t' + \mathbf{Z}_{\mathrm{T}}\right) \in \mathbb{R}^{N \times d_t}, \qquad (13)$$

where $\mathbf{W}_1 \in \mathbb{R}^{d_t \times 4d_t}$, $\mathbf{W}_2 \in \mathbb{R}^{4d_t \times d_t}$, $\mathbf{b}_1 \in \mathbb{R}^{4d_t}$, and $\mathbf{b}_2 \in \mathbb{R}^{d_t}$ are learnable parameters. We then decouple spatial components by calculating the difference between the input representation $\mathbf{Z}_{\mathrm{I}}$ and the temporal representation $\widetilde{\mathbf{Z}}_t$, denoted as $\mathbf{Z}_s^{(0)} = \mathbf{Z}_{\mathrm{I}} - \widetilde{\mathbf{Z}}_t$. Next, we utilize $L$ MLP layers to capture spatial high-dimensional features, with the final output denoted as the spatial representation $\mathbf{Z}_{\mathrm{S}} = \mathbf{Z}_s^{(L)}$. The forward process of the $l$-th MLP layer is as follows:

$$\mathbf{Z}_s^{(l+1)} = \mathrm{GELU}\left(\mathbf{Z}_s^{(l)}\mathbf{W}_3^{(l)} + \mathbf{b}_3^{(l)}\right)\mathbf{W}_4^{(l)} + \mathbf{b}_4^{(l)} + \mathbf{Z}_s^{(l)} \in \mathbb{R}^{N \times d_t}, \qquad (14)$$

where $\mathbf{W}_3^{(l)} \in \mathbb{R}^{d_t \times 4d_t}, \mathbf{W}_4^{(l)} \in \mathbb{R}^{4d_t \times d_t}, \mathbf{b}_3^{(l)} \in \mathbb{R}^{4d_t}$, and $\mathbf{b}_4^{(l)} \in \mathbb{R}^{d_t}$ are learnable parameters. Finally, same as the temporal part, we also use a linear layer to decode the spatial representation $\mathbf{Z}_s$ to produce a prediction from the spatial component:

$$\mathbf{Y}_s = \mathbf{Z}_S \mathbf{W}_s + \mathbf{b}_s \in \mathbb{R}^{N \times (T_P * c)}, \tag{15}$$

where $\mathbf{W}_s \in \mathbb{R}^{d_t \times (T_P * c)}$ and $\mathbf{b}_s \in \mathbb{R}^{T_P * c}$ are learnable parameters.

## 4.3 Final prediction

We sum the predictions from the spatial and temporal dimensions to get finial prediction $\widehat{\mathbf{Y}}$ as follow,

$$\widehat{\mathbf{Y}} = \mathbf{Y}_t + \mathbf{Y}_s \in \mathbb{R}^{N \times (T_P * c)}. \tag{16}$$

Finally, we reshape the predictions $\widehat{\mathbf{Y}}$ into $T_P \times N \times c$ to align the dimensions.

## 4.4 Graph perturbation mechanism

In this section, we introduce the Generalized Perturbation Units (GenPU) to perturb the interaction process of centralized messaging to improving generalization of the model to unknown environments. Additionally, we specifically design a Distributionally Robust Optimization (DRO) (Duchi & Namkoong, 2019) objective to optimize models and GenPU.

**Generalized Perturbation Units (GenPU).** To acquire robust contextual features, our strategy involves disrupting the aggregation process of the centralized messaging mechanism, which is responsible for updating context features. This approach enables us to circumvent the significant computational overhead associated with directly perturbing the data. Specifically, we create $M$ learnable perturbation vector in the training process, denoted $\mathbb{G} = \{\boldsymbol{g}_1, \boldsymbol{g}_2, \ldots, \boldsymbol{g}_M\}$, where $\boldsymbol{g}_i \in \mathbb{R}^N$ with $i \in \{1, 2, \cdots, M\}$ means $i$-th GenPU. Then, we use $\mathrm{softmax}$ operation to normalize $\boldsymbol{g}_i \in \mathbb{R}^N$ to get the corresponding masking probability vector $\boldsymbol{g}_i' = \mathrm{softmax}\,(\boldsymbol{g}_i) \in (0,1)^N$. Subsequently, we create a multinomial distribution $\mathcal{M}\,(\boldsymbol{g}_i'; s)$. Based on this distribution, we sample a masking indices $\widetilde{\boldsymbol{g}}_i \sim \mathcal{M}\,(\boldsymbol{g}_i'; s) \in \{0,1\}^N$, where $s \in (0, N)$ indicates the number of sample hits (i.e. the number of values equal to 1 in $\widetilde{\boldsymbol{g}}_i$). Finally, we create $K$ replicas of $\widetilde{\boldsymbol{g}}_i$ corresponding to $K$ ConAU. As a result, we can obtain a mask matrix with log operation as follows:

$$\mathbf{G}_i = \log\left([\widetilde{\boldsymbol{g}}_i, \widetilde{\boldsymbol{g}}_i, \cdots, \widetilde{\boldsymbol{g}}_i]\right) \in \{-\infty, 0\}^{K \times N}. \tag{17}$$

If $\mathbf{G}_i[m, n] = -\infty$, the aggregation interaction between $m$-th node and $n$-th ConAU is masked. Then we integrate $\mathbf{G}_i$ into low-rank attention mechanism to control the aggregation process:

$$\widetilde{\mathcal{A}}_i\left(\mathbf{Q}, \mathbf{K}, \mathbf{V}; \mathbf{G}_i\right) = \mathrm{softmax}\left(\alpha \mathbf{Q}\mathbf{K}^\top\right) \times \underbrace{\mathrm{softmax}\left(\alpha \mathbf{K}\mathbf{Q}^\top + \mathbf{G}_i\right)}_{\textbf{Perturbation operation}} \mathbf{V}. \tag{18}$$

From the ConAU's perspective during the aggregation process of perturbing contextual features, they perceive varying environment to learn context features, thereby compelling the model to acquire generalizable knowledge. In the training phase, we leverage $M$ GenPU in parallel to conduct the perturbation operation. Accordingly, according to Equation 16, the model will individually generate predictions for these $M$ environments, represented as $\{\widehat{\mathbf{Y}}_1, \widehat{\mathbf{Y}}_2, \ldots, \widehat{\mathbf{Y}}_M\}$.

**Spatiotemporal Distributionally Robust Optimization.** To promote effective learning from the diverse variable environments created, we introduce a spatiotemporal out-of-distribution (OOD) optimization objective that adheres to the principles of distributionally robust optimization (DRO) (Duchi & Namkoong, 2019), as explained in Appendix F. With $M$ predictions generated from different environments, our spatiotemporal DRO does not require optimizing all $M$ branches sequentially; instead, it selects the branch with the highest loss for gradient descent, as shown in the right half of Figure 2. This approach indicates that the model performs worst in that particular environment, thereby enhancing training efficiency and encouraging the model to learn purely invariant knowledge. We designate the GenPU responsible for generating this environment as $\boldsymbol{g}$. The specific optimization objective is defined as follows:

$$\min_f \sup_{\boldsymbol{g} \in \mathbb{R}^N} \mathbb{E}_{(\mathbf{X}, \mathbf{Y}) \sim (\mathcal{X}, \mathcal{Y} | e^*)} \left[\mathcal{L}\left(f\left(\mathbf{X}\right), \mathbf{Y}; \boldsymbol{g}\right)\right], \quad \text{s.t. } ||\widehat{\boldsymbol{g}}||_0 = s \in (0, N). \tag{19}$$

where "sup" means the supremum, and $|| \cdot ||_0$ stands for zero norm. GenPU participate in the learning process by influencing the sampling distribution of the mask matrix, which is essentially non-differentiable, rather than participating in the backpropagation process as part of the parameters. Thus, we optimize the model parameters and GenPU alternately, as shown in Algorithm 2.

**Robustness Analysis**. The GenPU introduces random perturbations in the spatial interaction process, effectively generating diversified training environments. This strategy prevents the model from becoming overly reliant on a single training environment, thereby promoting the learning of more generalizable features. Spatiotemporal DRO compels the model to engage with the most challenging instances within the generated environments, which can further enhance the model's robustness.

## 5 EXPERIMENTS

In this section, we conduct a comprehensive evaluation of the proposed model. We will answer the following potential questions. **Q.1**. What is the generalization performance of STOP in spatiotemporal OOD scenarios? **Q.2**.What is the inductive learning ability of STOP for new nodes? **Q.3**. How do model hyperparameters affect model performance? **Q.4**. Is each component of the model valid for OOD capabilities? **Q.5**. Is STOP effective in both T-OOD and S-OOD separate scenarios? **Q.6**. What are the insights of model efficiency and embedding?

### 5.1 EXPERIMENT SETTING

**Setting.** We set both the input and prediction windows to 12 in traffic prediction and 24 in atmospheric prediction. Temporal decomposition kernel size $\xi$ is equal to 3 in traffic datasets and 7 in KnowAir. The number of ConAU $K$ is set to $\{8, 24, 32, 64, 8, 4\}$ and the number of GenPU $M$ is equal to $\{3, 3, 3, 3, 2, 4\}$ in six datasets in Table 1. The dimensions of embeddings are set to 64. We use 8 heads in multi-head low-rank attention. We implement all models using PyTorch framework of Python 3.8.3 and leveraging the Nvidia A100-PCIE-40GB as support, and adopt Adam optimizer with a learning rate 0.002. MAE, RMSE, and MAPE are used as metrics for comparison.

**Datasets & baselines.** We conduct a comprehensive evaluation of our model on six spatiotemporal datasets spanning multiple years across two domains. These datasets include LargeST (Liu et al., 2024) and PEMS3-Stream (Chen et al., 2021) in the traffic domain, and KnowAir (Wang et al., 2020) in the atmospheric domain. The dataset summary is presented in Table 1. Our comparison involves advanced spatiotemporal models and spatiotemporal OOD learning methods. The spatiotempo-

Table 1: Spatiotemporal datasets.

| Dataset | Nodes | Edges | Years |
|---|---|---|---|
| LargeST-SD | 716 | 17,319 | 2017-2021 |
| LargeST-GBA | 2,352 | 61,246 | 2017-2021 |
| LargeST-GLA | 3,834 | 201,363 | 2017-2021 |
| LargeST-CA | 8,600 | 525,888 | 2017-2021 |
| PEMS3-Stream | 655 | 1,577 | 2011-2017 |
| KnowAir | 184 | 3,796 | 2015-2018 |

ral models include STGCN (Yu et al., 2017), GWNet (Wu et al., 2019), STNorm (Deng et al., 2021), STID (Shao et al., 2022a), STAEformer (Liu et al., 2023a), STNN (Yang et al., 2021), D²STGNN (Shao et al., 2022b), BigST (Han et al., 2024a), and RPMixer (Yeh et al., 2024). The spatiotemporal OOD models include CaST (Xia et al., 2024) and STONE (Wang et al., 2024a). *Some models require the removal of non-essential components (such as node embedding in STID or adaptive graph learning method in GWNet) to adapt them to the ST-OOD setting, as the parameters of them are intertwined with the scale of the graph structure*, as elaborated in Appendix C.1.

**ST-OOD Datasets.** For the evaluation of temporal shift, we train the models using data from the first year and test them on each subsequent year. The training set comprises the first 60% of data from the initial year dataset, while the following 20% of data is used as the validation set. In each subsequent year, the last 20% of data is designated as the test set. This setup aims to accentuate the temporal distribution difference between the test and training sets, while maintaining a ratio of approximately 6:2:2 for the training, validation, and test sets. Regarding structural shift evaluation, we select a subset of nodes for training and validation. In the test set, we decrease the number of nodes by 10% and introduce 30% new nodes to simulate shifts in the graph structure and scale. More detailed settings can be found in Appendix C.2.

Table 2: OOD performance comparisons on four datasets. The unit of MAPE is percent (%). We bold the best-performing model results in **red** and underline the sub-optimal model results in blue.

| | | Method | Imp. | **Ours** | STONE | CaST | RPMixer | BigST | $D^2$STGNN | STNN | STAEformer | STID | STNorm | GWNet | STGCN |
|---|---|---|---|---|---|---|---|---|---|---|---|---|---|---|---|
| SD | 3 | MAE | +3.96% | **17.71** | 18.44 | 21.35 | 24.92 | 18.56 | 18.70 | 36.46 | 18.70 | 19.68 | 18.82 | 20.15 | 18.68 |
| | | RMSE | +1.79% | **28.45** | 29.55 | 33.28 | 39.88 | 29.93 | 29.31 | 56.84 | 28.97 | 29.56 | 30.06 | 31.34 | 29.61 |
| | | MAPE | +3.69% | **11.73** | 12.32 | 16.04 | 15.63 | 12.18 | 13.04 | 26.91 | 12.62 | 13.18 | 12.82 | 14.44 | 12.92 |
| | 6 | MAE | +5.90% | **23.62** | 25.10 | 29.28 | 42.37 | 25.66 | 25.13 | 36.91 | 25.80 | 25.87 | 26.00 | 28.07 | 25.25 |
| | | RMSE | +2.73% | **37.71** | 39.66 | 45.24 | 66.45 | 40.61 | 38.77 | 57.59 | 40.73 | 40.86 | 41.20 | 43.00 | 39.48 |
| | | MAPE | +8.94% | **15.99** | 17.56 | 21.49 | 26.15 | 18.03 | 17.46 | 27.15 | 17.59 | 18.03 | 18.03 | 21.17 | 17.34 |
| | 12 | MAE | +9.85% | **32.59** | 37.12 | 42.40 | 77.31 | 37.89 | 36.35 | 41.69 | 37.17 | 38.30 | 38.08 | 39.75 | 36.15 |
| | | RMSE | +3.32% | **51.82** | 54.60 | 64.05 | 115.62 | 58.74 | 53.60 | 64.99 | 57.81 | 59.40 | 59.24 | 61.08 | 55.74 |
| | | MAPE | +11.62% | **22.89** | 25.90 | 31.73 | 49.48 | 27.12 | 25.98 | 31.32 | 27.07 | 26.90 | 27.89 | 31.46 | 26.41 |
| GBA | 3 | MAE | +3.98% | **18.33** | 20.19 | 21.85 | 24.79 | 19.92 | 19.10 | 40.61 | 20.91 | 19.09 | 20.86 | 20.65 | 21.49 |
| | | RMSE | +5.41% | **29.70** | 33.65 | 34.32 | 39.59 | 32.33 | 32.64 | 60.07 | 33.59 | 31.40 | 32.92 | 32.21 | 33.57 |
| | | MAPE | +5.01% | **13.64** | 15.10 | 18.61 | 17.06 | 14.75 | 14.29 | 33.77 | 14.93 | 14.36 | 15.57 | 15.70 | 14.79 |
| | 6 | MAE | +4.22% | **24.75** | 25.84 | 29.70 | 40.77 | 28.64 | 26.10 | 40.50 | 28.61 | 26.90 | 31.24 | 28.39 | 30.05 |
| | | RMSE | +7.77% | **38.48** | 41.96 | 45.16 | 62.24 | 43.93 | 41.72 | 59.96 | 44.03 | 42.15 | 46.69 | 42.60 | 44.97 |
| | | MAPE | +3.44% | **20.48** | 21.24 | 25.77 | 29.48 | 22.25 | 21.26 | 33.68 | 22.41 | 21.79 | 25.57 | 22.74 | 22.84 |
| | 12 | MAE | +3.60% | **34.93** | 39.56 | 42.60 | 72.51 | 42.87 | 36.26 | 44.62 | 41.68 | 39.36 | 45.73 | 39.61 | 43.29 |
| | | RMSE | +5.48% | **53.10** | 56.18 | 63.33 | 104.93 | 63.06 | 56.23 | 65.61 | 62.28 | 59.60 | 65.62 | 58.33 | 62.34 |
| | | MAPE | +3.39% | **31.09** | 32.18 | 36.88 | 56.28 | 34.52 | 32.23 | 38.28 | 34.99 | 33.43 | 41.02 | 33.67 | 35.23 |
| PEMS3-Stream | 3 | MAE | +11.09% | **11.39** | 13.27 | 15.43 | 14.68 | 12.79 | 12.89 | 17.04 | 12.81 | 12.96 | 13.03 | 12.97 | 13.39 |
| | | RMSE | +7.02% | **19.48** | 21.48 | 24.53 | 23.73 | 20.79 | 21.14 | 28.47 | 21.02 | 20.95 | 21.07 | 21.11 | 21.60 |
| | | MAPE | +6.25% | **15.45** | 17.06 | 32.15 | 18.02 | 17.30 | 16.58 | 23.63 | 16.48 | 16.66 | 20.44 | 16.41 | 16.71 |
| | 6 | MAE | +11.81% | **12.47** | 14.30 | 17.13 | 17.41 | 14.05 | 14.08 | 17.26 | 14.14 | 14.18 | 14.51 | 14.14 | 14.63 |
| | | RMSE | +6.77% | **21.62** | 23.68 | 27.63 | 28.61 | 23.07 | 23.26 | 29.27 | 23.38 | 23.19 | 23.67 | 23.31 | 23.82 |
| | | MAPE | +9.08% | **16.02** | 18.23 | 33.77 | 20.90 | 19.54 | 17.62 | 25.63 | 19.71 | 18.52 | 22.43 | 17.91 | 18.33 |
| | 12 | MAE | +11.79% | **14.36** | 16.28 | 20.96 | 24.00 | 16.65 | 16.55 | 18.19 | 16.71 | 16.56 | 17.04 | 16.37 | 17.25 |
| | | RMSE | +8.64% | **24.95** | 28.41 | 33.82 | 39.64 | 27.46 | 27.44 | 30.14 | 27.92 | 27.31 | 27.94 | 27.10 | 28.20 |
| | | MAPE | +10.89% | **18.66** | 20.94 | 39.07 | 27.84 | 23.59 | 20.12 | 30.81 | 20.95 | 21.25 | 25.30 | 20.29 | 21.30 |
| Know-Air | 6 | MAE | +5.10% | **24.37** | 25.68 | 26.20 | 30.56 | 26.89 | 26.43 | 27.85 | 26.19 | 26.49 | 28.46 | 27.84 | 27.92 |
| | | RMSE | +2.74% | **36.56** | 37.59 | 38.42 | 45.34 | 39.16 | 37.91 | 39.07 | 37.82 | 38.90 | 41.47 | 40.25 | 39.47 |
| | | MAPE | +0.90% | **51.94** | 52.41 | 59.53 | 69.06 | 57.45 | 58.39 | 65.74 | 52.90 | 57.84 | 65.26 | 52.42 | 58.32 |
| | 12 | MAE | +6.66% | **27.03** | 28.96 | 29.49 | 38.45 | 29.77 | 30.06 | 30.48 | 29.45 | 30.85 | 30.86 | 31.11 | 31.63 |
| | | RMSE | +3.40% | **40.29** | 42.64 | 41.98 | 55.26 | 41.75 | 42.52 | 42.67 | 41.71 | 44.59 | 43.87 | 43.65 | 43.71 |
| | | MAPE | +11.48% | **54.45** | 71.99 | 70.15 | 87.60 | 68.39 | 67.10 | 71.05 | 61.64 | 68.44 | 71.83 | 61.51 | 69.83 |
| | 24 | MAE | +6.09% | **28.70** | 30.56 | 31.63 | 42.67 | 31.57 | 30.94 | 31.48 | 30.96 | 32.78 | 32.52 | 32.99 | 34.68 |
| | | RMSE | +6.78% | **42.39** | 45.48 | 45.21 | 61.30 | 44.52 | 46.21 | 44.72 | 43.48 | 46.67 | 44.80 | 44.14 | 47.19 |
| | | MAPE | +17.01% | **57.96** | 75.11 | 75.36 | 94.76 | 76.76 | 69.84 | 74.14 | 65.31 | 74.02 | 81.32 | 70.84 | 80.49 |

## 5.2 OOD Performance Comparison(Q.1)

As shown in Table 2, we report the average values across all years of test sets on four datasets. Experiments on large datasets can be found in Appendix C.3, and detailed year-specific reports can be found in Appendix C.8.

GCN-based models like STGCN and GWNet underperform in OOD settings due to their reliance on the global messaging mechanism of GCN, rendering them highly sensitive to spatiotemporal shifts. Transformer-based models such as STAEformer and D²STGNN exhibit improved predictive accuracy by leveraging self-attention mechanisms to aggregate global node features, effectively addressing spatiotemporal shift errors. Conversely, MLP-based models like STID and BigST, which treat nodes as independent channels, suffer from reduced performance due to the lack of spatial interaction information. Despite these advancements, STGNNs still face challenges in generalizing weights for unseen graph structures. On the other hand, spatiotemporal OOD baselines like STONE introduce diverse training environments utilizing perturbation-generated semantic relations to learn invariant causal knowledge, resulting in enhanced performance.

STOP demonstrates significant improvements across various metrics, with a maximum enhancement of 17.01%. This improvement can be attributed to its robust centralized messaging mechanism, which facilitates effective spatial feature interaction.

## 5.3 Inductive learning performance of STOP (Q.2)

To compare the inductive learning performance of models, we report the their performance on new nodes in Table 3. Specifically, Transformer-based models, such as D²STGNN, demonstrates strong generalization capabilities because the self-attention mechanism generates accurate representations for new nodes to some extent. GCN-based models exhibit the weakest generalization capabilities because the trained model parameters are coupled with the original graph structure, and new nodes cannot generate accurate representations by aggregating neighboring nodes. The performance of Transformer-based models is poor because the attention mechanism cannot generate robust aggregate weights for new nodes. On the other hand, the spatiotemporal OOD learning framework

Table 3: OOD inductive learning performance comparisons on SD and GBA datasets of new nodes.

| | Method | | Imp. | Ours | STONE | CaST | RPMixer | BigST | D²STGNN | STNN | STAEformer | STID | STNorm | GWNet | STGCN |
|---|---|---|---|---|---|---|---|---|---|---|---|---|---|---|---|
| SD | 3 | MAE | +2.68% | **17.02** | 17.56 | 20.51 | 23.67 | 17.75 | 18.74 | 40.27 | 17.94 | 17.73 | 18.12 | 20.33 | 18.01 |
| | | RMSE | +2.26% | **26.85** | 27.59 | 31.42 | 36.94 | 27.82 | 28.92 | 62.87 | 28.06 | 27.63 | 28.31 | 31.18 | 27.88 |
| | | MAPE | +4.51% | **11.64** | 14.65 | 16.55 | 15.35 | 12.19 | 14.50 | 29.10 | 12.63 | 13.38 | 12.89 | 15.00 | 13.12 |
| | 6 | MAE | +6.84% | **22.73** | 25.60 | 28.22 | 40.08 | 24.62 | 25.51 | 40.65 | 24.85 | 24.82 | 25.13 | 28.82 | 24.40 |
| | | RMSE | +4.61% | **35.99** | 38.16 | 43.50 | 62.09 | 38.42 | 39.06 | 63.50 | 38.85 | 38.80 | 39.45 | 43.85 | 37.73 |
| | | MAPE | +8.98% | **15.91** | 18.02 | 22.09 | 25.32 | 17.48 | 19.08 | 29.23 | 17.59 | 18.04 | 18.06 | 22.32 | 17.64 |
| | 12 | MAE | +9.94% | **31.53** | 35.01 | 41.16 | 73.30 | 36.52 | 36.20 | 45.14 | 35.96 | 36.83 | 37.06 | 41.48 | 35.03 |
| | | RMSE | +7.20% | **50.13** | 54.02 | 62.63 | 109.29 | 56.60 | 54.61 | 70.34 | 56.06 | 57.24 | 57.95 | 63.34 | 54.11 |
| | | MAPE | +15.07% | **22.88** | 26.94 | 32.58 | 47.83 | 27.10 | 28.95 | 33.47 | 27.15 | 26.98 | 27.98 | 33.92 | 26.97 |
| GBA | 3 | MAE | +3.21% | **18.08** | 18.68 | 21.43 | 24.40 | 19.59 | 19.12 | 40.58 | 20.57 | 18.76 | 20.55 | 20.86 | 24.87 |
| | | RMSE | +4.75% | **29.26** | 30.72 | 33.69 | 38.84 | 31.76 | 32.55 | 60.04 | 33.02 | 30.86 | 32.40 | 32.38 | 38.36 |
| | | MAPE | +0.74% | **13.35** | 15.67 | 18.11 | 16.79 | 14.41 | 14.20 | 33.23 | 14.61 | 13.45 | 12.89 | 15.92 | 18.58 |
| | 6 | MAE | +6.48% | **24.41** | 27.30 | 29.06 | 40.09 | 28.09 | 26.10 | 40.46 | 28.09 | 26.38 | 30.74 | 29.03 | 29.72 |
| | | RMSE | +7.81% | **37.91** | 41.12 | 44.24 | 61.01 | 43.05 | 41.86 | 59.92 | 43.19 | 41.33 | 45.89 | 43.32 | 44.44 |
| | | MAPE | +4.24% | **20.10** | 20.99 | 25.02 | 28.93 | 21.68 | 20.35 | 33.16 | 21.82 | 21.04 | 24.53 | 23.60 | 22.39 |
| | 12 | MAE | +6.51% | **34.48** | 39.61 | 41.59 | 71.38 | 41.96 | 36.88 | 44.28 | 40.80 | 38.49 | 44.95 | 41.05 | 42.82 |
| | | RMSE | +4.65% | **52.48** | 55.04 | 62.03 | 103.18 | 61.76 | 56.67 | 65.15 | 61.01 | 58.44 | 64.49 | 60.19 | 61.66 |
| | | MAPE | +8.52% | **30.70** | 33.78 | 35.80 | 55.35 | 33.56 | 32.72 | 37.46 | 34.00 | 39.75 | 39.72 | 35.65 | 34.51 |

STONE uses a novel embedding method that computes the distances between nodes and anchor points to generate initial embeddings for new nodes, resulting in good performance. However, our model excels in extending performance to new nodes by leveraging the centralized messaging mechanism to access contextual features and enhance representations.

## 5.4 HYPERPARAMETER SENSITIVITY ANALYSIS (Q.3)

In this section, we analyze the sensitivity of the numer of ConAU and GenPU on the SD and KnowAir datasets. The performance is presented in Figure 3. When the number of ConAU $K$ is set to 8 in SD dataset and 4 in KnowAir dataset. When $K$ exceeds this value, the model creates too many ConAU, making it unable to focus on extracting invariant contextual features, thus introducing noise. When $K$ is less than this value, too few perception units fail to learn sufficient invariant knowledge, leading to a decrease in the model's generalization performance. The number of GenPU $M$ is set to 3 in SD dataset and 4 in KnowAir dataset. A smaller $M$ may not provide sufficient training environment diversity, resulting in performance degradation. On the other hand, an excessive number of GenPU does not necessarily improve performance. Too large $M$ means that the generated environment is too complex, which increases the learning difficulty of the model to extract causal knowledge.

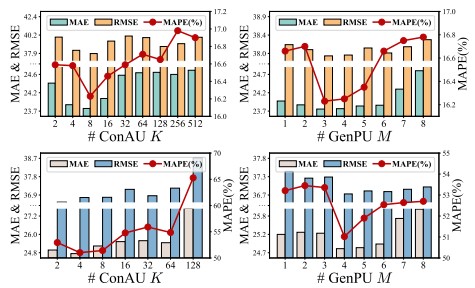

Figure 3: Sensitivity experiments of STOP.

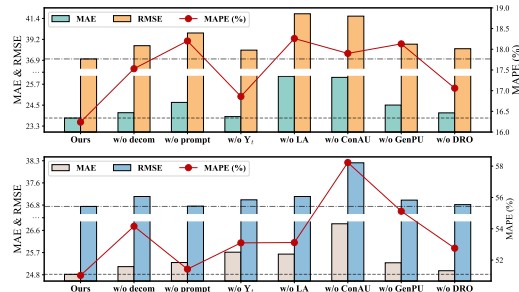

Figure 4: Ablation experiments on two datasets.

## 5.5 ABLATION STUDY (Q.4)

We conduct an ablation study to examine the effectiveness of each component on SD and KnowAir datasets. "w/o decom" removes the time decomposition module, "w/o prompt" eliminates the spatiotemporal prompting method, "w/o $\mathbf{Y}_t$" uses only spatial prediction as the final prediction. "w/o LA" means we use vanilla self-attention mechanism to replace the low-rank attention module.

As illustrated in Figure 4, the results show that each component of the model helps to improve the model's OOD capabilities. "w/o $\mathbf{Y}_t$" achieves poor prediction performance, which proves that the proposed parallel component is effective for OOD. "w/o ConAU" removes ConAU and achieves high errors, demonstrating that spatial features is crucial to improving the generalization ability of the model. "w/o GenPU" has higher prediction errors because GenPU can help the model extract

Table 4: Performance comparison in T-OOD and S-OOD scenarios.

| | Method | **Ours** | STONE | D$^2$STGNN | STNN | STGCN | GWNet |
|---|---|---|---|---|---|---|---|
| S-OOD | MAE | **23.21** | 25.00 | 26.56 | 35.06 | 29.74 | 26.79 |
| | RMSE | **36.95** | 39.12 | 42.77 | 55.12 | 44.45 | 41.47 |
| | MAPE | **14.45** | 16.72 | 19.80 | 23.42 | 21.79 | 18.16 |
| T-OOD | MAE | **22.91** | 25.41 | 24.23 | 36.14 | 25.73 | 23.38 |
| | RMSE | **37.17** | 37.56 | 39.04 | 56.26 | 40.07 | 37.63 |
| | MAPE | **15.35** | 16.38 | 17.37 | 26.46 | 17.68 | 16.58 |

causal knowledge and enhance model robustness. We perform comprehensive ablation experiments including double ablation in Appendix C.9.

### 5.6 PERFORMANCE IN S-OOD AND T-OOD SCENARIOS (Q.5)

With LargeST-SD dataset, we investigate the performance of models in T-OOD and S-OOD scenarios. Used two datasets are simplified versions of ST-OOD. For S-OOD, we use the last 20% of the 2017 data as the test set with the graph structure unchanged. For T-OOD, we maintain the graph structure consistent between the training and testing environments, aligning the data selection with ST-OOD. The experimental results are shown in Table 4, and we can observe that STGNNs exhibit poor performance in the S-OOD scenario, mainly due to the sensitivity of the node-to-node interaction method to structural shifts. The poor performance of STNN can be attributed to its use of Transformer, which lacks robustness against noise introduced by temporal and spatial shifts. Our model has achieved competitive performance in both T-OOD and S-OOD scenarios.

### 5.7 CASE STUDY (Q.6)

**Embedding visualization**. Using LargeST-SD dataset as example, we visualize the temporal prompt embedding $\mathbf{E}$ in Figure 5 (a), Personalized features $\mathbf{Z}_p$, and contextual features $\mathbf{Z}_c$ in Figure 5 (b). We can see that temporal embeddings unveil essential periodic patterns for OOD scenarios. Both node personalized and context features exhibit strong discriminative capabilities. Context features capture shared node patterns, ensuring resilience to individual node variations. Meanwhile, personalized features enhance the model's ability to tailor predictions for each node effectively.

**Efficiency study**. The training time of peer epoch is illustrated in Figure 5 (c), we can see that STOP demonstrates remarkable effectiveness and efficiency on the SD dataset. This is becauase our model primarily uses lightweight MLP layers to model temporal and spatial dynamics. Compared to the SOTA model D$^2$STGNN, our model have improved the efficiency by about 20 times.

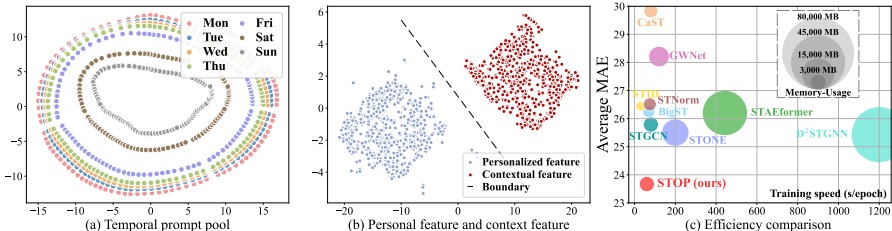

Figure 5: Visual case and efficiency study of STOP on LargeST-SD dataset.

## 6 CONCLUSION

In this paper, we present a Spatio-Temporal Out-of-Distribution Processor, namely STOP, which incorporates a spatial interaction mechanism and a graph perturbation mechanism to enhance resilience against spatiotemporal shifts. The spatial interaction mechanism employs a centralized messaging pattern for nodes to engage with ConAU, facilitating spatial feature interactions. Through the graph perturbation mechanism, random disruptions are introduced to diversify training environments, bolstering the model's robustness. Assessment across numerous datasets in various OOD scenarios showcases the model's robust generalization, inductive learning, and efficiency.

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

Table 5: Abbreviations used in the paper, along with their full names and descriptions

| Abbreviation | Full name or description |
| --- | --- |
| IID | Independent and identically distribution |
| OOD | Out-of-distribution |
| S-OOD | Spatial out-of-distribution with structural shifts |
| ST-OOD | Spatiotemporal out-of-distribution with structural and temporal shifts |
| MLP | Multilayer perceptron |
| GCN | Graph Convolutional Network |
| STGNN | Spatiotemporal Graph Neural Network |
| STOP | Our model: Spatiotemporal OOD Processor |
| ConAU | Context Aware Units |
| GenPU | Generalized Perturbation Units |
| DRO | Distributionally robust optimization |

## A  RELATED WORK

### A.1  CONTINUAL LEARNING WITH SPATIOTEMPORAL SHIFTS

Several studies (Chen et al., 2021; Wang et al., 2023a; Lee & Park, 2024; Chen & Liang, 2024) have proposed continual learning strategies to tackle spatiotemporal graph prediction in scenarios with spatiotemporal shifts. When the spatiotemporal data distribution undergoes changes, these models engage in fine-tuning using a subset of new data to adjust to the updated data distribution. For instance, TrafficStream (Chen et al., 2021) recommends utilizing subsets of newly added nodes and significant temporal pattern data changes for fine-tuning the model. PECPM (Wang et al., 2023b) identifies conflicting nodes to enhance the fine-tuning process, focusing on nodes that have experienced substantial changes. DLF (Wang et al., 2024b) introduces a streaming training strategy to continuously fine-tune the model to adapt to the dynamic nature of spatiotemporal data. TF-MoE (Lee & Park, 2024) partitions traffic flow into multiple homogeneous groups and assigns an expert model responsible for each group, enabling each expert model to specialize in learning and adapting to specific patterns. However, these models often compromise performance to improve learning efficiency, resulting in lower performance compared to traditional spatiotemporal models. Primarily, these models train and fine-tune on a sufficient amount of new distribution data (approximately 21 days in one month) and test on the new data distribution, thereby adhering to the IID assumption and encountering difficulties in OOD learning.

### A.2  TEMPORAL SHIFT IN TIME SERIES

Various models have been developed in the time series domain to address temporal shifts in time series data, particularly focusing on OOD learning challenges. For instance, RevIN (Kim et al., 2021) employs a symmetric structure to eliminate and reconstruct distribution information based on the input window's statistics. AdaRNN (Du et al., 2021) categorizes historical time sequences into different classes and dynamically matches input data to these classes for contextual information identification. Additionally, a reversible instance normalization technique, proposed by (Kim et al., 2021), aims to mitigate temporal distribution shift issues. Non-stationary Transformers (Liu et al., 2022) introduce a normalization-denormalization technique to stabilize time series data, mainly for transformer-based models. SAF (Arik et al., 2022) suggests test-time adaptation through a self-supervised objective to enhance adaptation against distribution shifts. DIVERSIFY (Lu et al., 2024) aims to leverage subdomains within a dataset to mitigate challenges arising from non-stationary generalized representation learning. However, these models often overlook the modeling of spatial dependencies. Spatial modeling is crucial in the field of spatiotemporal prediction, as it can examine the states of neighboring nodes to enhance prediction performance, given the strong correlations that often exist among neighboring nodes (Jin et al., 2023; Shao et al., 2023).

# B ALGORITHM & OPTIMISATION

We have provided the pseudocode of the algorithm in Algorithm 1, where we can observe that STOP makes final predictions based on the temporal component and spatial component. This includes a perturbation process to extract robust knowledge. This perturbation process only occurs in the training phase and we no longer use it in the test phase. We also provide the optimization flow of GenPU and model parameters in Algorithm 2. As shown, we interleaved the optimization of GenPU and model parameters.

---

**Algorithm 1:** STOP for spatiotemporal prediction

**Input:** Historical data $\mathbf{X} \in \mathbb{R}^{T \times N \times c}$.
**Output:** Future prediction $\widehat{\mathbf{Y}} \in \mathbb{R}^{T_P \times N \times c}$.

1  **# Data encode;**
2  $\mathbf{H}_0$ in Eq. 2 $\sim$ 4;                                      // Temporal decomposition
3  $\mathbf{Z}_\mathrm{I} \leftarrow \mathbf{H}_0, \widetilde{\mathbf{T}}, \mathbf{E}$ in Eq. 5 $\sim$ 6;                    // Input representation
4  **# Temporal modeling and prediction;**
5  $\mathbf{Z}_\mathrm{T} \leftarrow \mathbf{Z}_\mathrm{I}$ in Eq. 7;                        // Temporal representation learning
6  $\mathbf{Y}_t \leftarrow \mathbf{Z}_\mathrm{T}$ in Eq. 8;                          // Temporal prediction component
7  **# Spatial modeling and prediction;**
8  **if** *test phase* **then**                                // centralized messaging mechanism
9  $\quad \mid \quad \mathbf{Z}_c \leftarrow \mathbf{Z}_\mathrm{T}, \mathbf{C}$ in Eq. 9 $\sim$ 10;                                       // ConAU
10 **if** *training phase* **then**
11 $\quad \mid \quad \mathbf{Z}_c \leftarrow \mathbf{Z}_\mathrm{T}, \mathbf{C}, \boldsymbol{g}$ in Eq. 9 $\sim$ 10, 17 $\sim$ 18;                    // ConAU & GenPU
12 $\mathbf{Z}_s \leftarrow \mathbf{Z}_c, \mathbf{Z}_\mathrm{T}$ in Eq. 11 $\sim$ 14;                // Spatial representation learning
13 $\mathbf{Y}_s \leftarrow \mathbf{Z}_s$ in Eq. 15;                      // Spatial prediction component
14 **# Final prediction;**
15 $\widehat{\mathbf{Y}} \leftarrow \mathbf{Y}_t + \mathbf{Y}_s$ in Eq. 16;                            // Final prediction

---

**Algorithm 2:** Optimization process of STOP

**Input:** Historical data $\mathbf{X} \in \mathbb{R}^{T \times N \times c}$, GenPU $\mathbb{G} = \{\boldsymbol{g}_1, \boldsymbol{g}_2, \ldots, \boldsymbol{g}_M\} \subseteq \mathbb{R}^N$, sample hits $s \in (0, N)$, future label $\mathbf{Y} \in \mathbb{R}^{T_P \times N \times c}$, loss function $\mathcal{L}$, initialized parameters $\Theta$ of STOP function $f$, patience $P$, learning rates $\alpha$ and $\beta$.
**Output:** Well-trained parameters $\Theta^*$ of STOP.

1  **while** *maximum epochs nor reached or not converged* **do**
2  $\quad$ **for** *patience* $= 1, 2, \ldots, P$ **do**
3  $\quad \quad$ **for** $j = 1, 2, \ldots, M$ **do**
4  $\quad \quad \quad \mid \quad \boldsymbol{g}_j' \leftarrow \mathrm{softmax}\,(\boldsymbol{g}_j)$;
5  $\quad \quad \quad \mid \quad \widetilde{\boldsymbol{g}}_j \leftarrow$ sampling from multinomial distribution $\mathcal{M}\,(\boldsymbol{g}_j'; s)$;
6  $\quad \quad \quad \mid \quad \mathbf{G}_j \leftarrow \widetilde{\boldsymbol{g}}_j$ in Eq. 17;                    // Generalized Perturbation Units
7  $\quad \quad \quad \mid \quad \mathcal{L}_j \leftarrow \mathcal{L}\,(f\,(\mathbf{X})\,, \mathbf{Y}; \mathbf{G}_j)$;
8  $\quad \quad$ **end**
9  $\quad \quad \mathcal{L}^* \leftarrow \max\{\mathcal{L}_1, \mathcal{L}_2, \ldots, \mathcal{L}_M\}$;
10 $\quad \quad \Theta \leftarrow \Theta - \alpha \nabla_\Theta \mathcal{L}^*$;                        // Update the parametners of STOP
11 $\quad$ **end**
12 $\quad i \leftarrow \arg\max\{\mathcal{L}_1, \mathcal{L}_2, \ldots, \mathcal{L}_M\}$;
13 $\quad \boldsymbol{g}_i \leftarrow \boldsymbol{g}_i + \beta\left((\mathbf{1} - \widetilde{\boldsymbol{g}}_i)\,\boldsymbol{g}_i^\top - \log\|\exp\boldsymbol{g}_i\|_1\right)\mathcal{L}^*$;                    // Update GenPU
14 **end**

---

## C EXPERIMENTS

### C.1 BASELINE DETAIL

In experiments, we compare a lot of spatiotemporal prediction models with spatiotemporal OOD models. However, the original versions of many of these models are not compatible with the OOD setting. Consequently, we had to remove certain non-essential code related to graph structures, particularly node embedding techniques and adaptive graph structure learning techniques.

**Node embedding technology.** The researchers set a node embedding vector $E \in \mathbb{R}^{N \times d_s}$ to capture node patterns adaptively, which are coupled with the size $N$ of the graph structure. Therefore, when the model is trained, it cannot be run directly into the test environment with ST-OOD. STID, STAEformer, and BigST use this technology.

**Adaptive graph learning.** This method generally use two noode embedding vectors $E_1 \in \mathbb{R}^{N \times d_s}$ and $E_2 \in \mathbb{R}^{N \times d_s}$, and they multiply these two node embedding matrices, $A_s = E_1 E_2^\top \in \mathbb{R}^{N \times N}$, to generate an adaptive adjacency matrix $A_s$ for learning the adjacency matrix, which is then used for GCN. GWNet, D$^2$STGNN, and CaST adopt this method.

### C.2 EXPERIMENTAL DATASET DETAILS

In this paper, we utilized six datasets to evaluate the effectiveness of the models in OOD scenarios, primarily from the domains of transportation and atmosphere. These datasets often span multiple years. Among them, LargeST (Liu et al., 2024) collected five years of data from 8600 records, sampled at a frequency of five minutes. PEMS3-Stream (Chen et al., 2021) is a naturally streaming traffic dataset, recording data from July each year from 2011 to 2017, where the traffic structure expands year by year, naturally representing spatiotemporal shifts. Knowair (Wang et al., 2020) collected 18 atmospheric features sampled at an hourly frequency. We followed the following rules to create spatiotemporal OOD datasets.

Temporal shift: We used the first 60% of data from the first year as the training set, followed by 20% of data for the validation set. We used the last 20% of data from subsequent years for the test set. This longer time interval ensures changes in temporal distribution characteristics.

Structural shift: Apart from the PeMS3-Stream dataset, we selected a subset of nodes for training and validation, approximately 75% of the total number, in the test set, we randomly masked 10% of nodes to simulate node disappearance and added 30% of nodes as new nodes. This is because for spatiotemporal systems, cities or detection systems generally tend to expand. Since PeMS3-Stream is a natural streaming data set, we use it directly.

Table 6: The details of used datasets.

| Dataset | Training set | | Test set | | |
| | Time range | Graph Nodes | Temporal shift | Structural shift | |
| | | | | New nodes | Removed Nodes |
|---|---|---|---|---|---|
| LargeST-SD | First 60% data in 2017 | 550 | Last 20% data in 2018-2021 | 165 | 55 |
| LargeST-GBA | First 60% data in 2017 | 1809 | Last 20% data in 2018-2021 | 542 | 180 |
| LargeST-GLA | First 60% data in 2017 | 2949 | Last 20% data in 2018-2021 | 884 | 294 |
| LargeST-CA | First 60% data in 2017 | 6615 | Last 20% data in 2018-2021 | 1984 | 661 |
| KnowAir | First 60% data in 2011 | 141 | Last 20% data in 2012-2017 | 42 | 14 |
| PEMS3-Stream | First 60% data in 2015 | 655 | Last 20% data in 2016-2021 | (60, 131, 167, 179, 195, 216) | 0 |

### C.3 OOD PERFORMANCE COMPARISON ON LARGE DATASETS

As the largest collection of spatiotemporal data available in open source today, CA represents an invaluable test case for the OOD capability of the model. The performance of STOP and the baseline is evaluated on large-scale and large-scale spatiotemporal datasets, respectively, under identical conditions.

Based on the same partitioning strategy as described in Section 5.1, we divide the LargeST dataset into the two largest subdatasets, GLA and CA. Due to the parameter complexity of Transformer-

based baselines such as STAEformer, STNN, D$^2$STGNN, and STONE, which scales at least quadratically with the number of nodes, deploying these models on GLA and CA datasets is not feasible.

As shown in Table 7, STOP consistently outperforms the baselines on both the large-scale spatiotemporal OOD dataset in terms of overall performance and performance on newly added nodes, with improvements of up to 14.01%. On large-scale spatiotemporal datasets, the performance of baselines based on global message passing mechanisms declines significantly due to the introduction of more new nodes. STID, which does not involve node interactions, achieves the second-best performance among the baselines. In contrast, STOP benefits from ConAU by decomposing large-scale spatiotemporal scenes into stable spatiotemporal subenvironments, leading to the best performance while ensuring node interactions. This highlights STOP's remarkable OOD capabilities even in large-scale scenarios.

Table 7: OOD performance comparisons on GLA and CA datasets. The absence of baselines indicates that the models incur out-of-memory issues.

| | | Method | Imp. | Ours | CaST | RPMixer | BigST | STID | STNorm | GWNet | STGCN |
|---|---|---|---|---|---|---|---|---|---|---|---|
| GLA | 3 | MAE | +3.72% | **19.13** | 23.36 | 25.89 | 20.32 | 19.87 | 21.05 | 21.17 | 20.51 |
| | | RMSE | +4.56% | **30.33** | 35.53 | 41.10 | 32.56 | 31.78 | 33.03 | 32.96 | 32.24 |
| | | MAPE | +0.83% | **11.93** | 21.44 | 14.90 | 12.93 | 12.03 | 13.34 | 13.87 | 12.81 |
| | 6 | MAE | +7.10% | **26.29** | 31.43 | 43.33 | 28.83 | 28.30 | 30.70 | 29.91 | 29.13 |
| | | RMSE | +7.42% | **40.66** | 47.49 | 66.65 | 44.69 | 43.92 | 46.35 | 45.47 | 44.50 |
| | | MAPE | +0.68% | **17.60** | 27.75 | 26.18 | 18.49 | 17.72 | 20.57 | 19.90 | 19.36 |
| | 12 | MAE | +10.90% | **36.87** | 43.48 | 77.32 | 42.12 | 41.38 | 46.13 | 41.81 | 43.92 |
| | | RMSE | +9.86% | **55.96** | 65.08 | 114.02 | 62.99 | 62.69 | 66.98 | 62.08 | 64.34 |
| | | MAPE | +2.97% | **27.07** | 36.46 | 53.23 | 30.33 | 27.90 | 34.63 | 28.21 | 31.14 |
| CA | 3 | MAE | +4.80% | **17.47** | 21.87 | 23.72 | 18.77 | 18.35 | 19.10 | 19.01 | 19.23 |
| | | RMSE | +5.90% | **28.24** | 34.44 | 38.43 | 30.77 | 30.01 | 30.86 | 30.30 | 30.89 |
| | | MAPE | +1.78% | **12.69** | 17.79 | 16.02 | 13.60 | 12.92 | 15.38 | 13.62 | 13.68 |
| | 6 | MAE | +9.06% | **23.70** | 29.13 | 39.52 | 26.80 | 26.06 | 27.63 | 26.64 | 27.30 |
| | | RMSE | +10.04% | **37.17** | 45.30 | 61.88 | 42.34 | 41.33 | 43.10 | 41.32 | 42.51 |
| | | MAPE | +4.86% | **18.39** | 23.63 | 27.42 | 19.98 | 19.33 | 23.24 | 19.56 | 20.23 |
| | 12 | MAE | +12.68% | **32.86** | 41.26 | 70.64 | 39.59 | 38.23 | 40.77 | 37.63 | 40.64 |
| | | RMSE | +11.90% | **50.28** | 62.85 | 105.36 | 60.24 | 59.16 | 61.20 | 57.07 | 61.01 |
| | | MAPE | +7.12% | **27.65** | 34.71 | 53.26 | 32.00 | 29.77 | 35.50 | 30.31 | 31.93 |

Table 8: Inductive learning preformance on GLA and CA datasets of new nodes. The absence of baselines indicates that the models incur out-of-memory issues.

| | | Method | Imp. | Ours | CaST | RPMixer | BigST | STID | STNorm | GWNet | STGCN |
|---|---|---|---|---|---|---|---|---|---|---|---|
| GLA | 3 | MAE | +3.65% | **18.99** | 23.09 | 25.65 | 20.17 | 19.71 | 20.92 | 21.35 | 20.36 |
| | | RMSE | +4.50% | **30.13** | 35.16 | 40.89 | 32.36 | 31.55 | 32.84 | 33.30 | 32.05 |
| | | MAPE | +0.75% | **11.94** | 21.32 | 14.86 | 12.91 | 12.03 | 13.26 | 14.01 | 12.81 |
| | 6 | MAE | +7.00% | **26.17** | 31.15 | 42.95 | 28.66 | 28.14 | 30.56 | 30.46 | 29.00 |
| | | RMSE | +7.29% | **40.57** | 47.16 | 66.35 | 44.51 | 43.76 | 46.24 | 46.51 | 44.39 |
| | | MAPE | +0.51% | **17.64** | 27.62 | 26.09 | 18.47 | 17.73 | 20.47 | 20.25 | 19.38 |
| | 12 | MAE | +10.58% | **36.78** | 43.12 | 76.60 | 41.84 | 41.13 | 45.87 | 42.97 | 43.70 |
| | | RMSE | +10.38% | **55.89** | 64.65 | 113.41 | 62.56 | 62.36 | 66.70 | 63.92 | 64.04 |
| | | MAPE | +2.51% | **27.17** | 36.35 | 53.12 | 30.24 | 27.87 | 34.35 | 28.98 | 31.14 |
| CA | 3 | MAE | +4.74% | **17.48** | 21.86 | 23.73 | 18.76 | 18.35 | 19.10 | 19.38 | 19.23 |
| | | RMSE | +5.81% | **28.39** | 34.50 | 38.59 | 30.86 | 30.14 | 30.98 | 30.87 | 30.96 |
| | | MAPE | +1.91% | **12.87** | 18.46 | 16.15 | 13.85 | 13.12 | 16.06 | 15.62 | 13.97 |
| | 6 | MAE | +8.98% | **23.71** | 29.11 | 39.50 | 26.79 | 26.05 | 27.65 | 27.47 | 27.30 |
| | | RMSE | +9.93% | **37.29** | 45.31 | 62.03 | 42.38 | 41.40 | 43.20 | 42.50 | 42.53 |
| | | MAPE | +4.92% | **18.73** | 24.37 | 27.65 | 20.34 | 19.70 | 24.45 | 22.76 | 20.62 |
| | 12 | MAE | +14.01% | **32.83** | 41.22 | 70.53 | 39.53 | 38.18 | 40.75 | 39.27 | 40.61 |
| | | RMSE | +14.93% | **50.30** | 62.78 | 105.38 | 60.14 | 59.13 | 61.20 | 59.43 | 60.94 |
| | | MAPE | +7.23% | **28.24** | 35.75 | 53.61 | 32.69 | 30.44 | 37.24 | 35.64 | 32.53 |

## C.4 INDUCTIVE LEARNING COMPARISON ON LARGE DATASETS

To evaluate the inductive learning capabilities of each model, we further report the performance of added nodes in Table 8. We can see that GCN-based models have overall poor inductive capabilities. While they can rely on message passing mechanisms to generalize learned information to unseen nodes, the spatially confused interactions cannot guarantee accurate descriptions of added nodes, leading to subpar performance. In this regard, STID achieves better predictive results because it assumes nodes are independent, allowing the model to learn time-related knowledge that is unrelated

to nodes, which can generalize to added nodes and avoid error accumulation. Our model demonstrates strong inductive learning capabilities on large-scale graphs, as added nodes can access shared context features to obtain good representations.

## C.5 PERFORMANCE ON RAPID EVOLUTING SPATIOTEMPORAL DYNAMICAL SYSTEM

In the main experiment, the proportion of added nodes is relatively small (only 30%), which may not cover rapidly developing urban scenarios. We further create a challenging scenario where we train on 30% of nodes from the year 2017 and test on the remaining 70% of nodes from subsequent years. Details of the experimental dataset are provided in Table 9.

Table 9: Rapidly growth OOD setting on SD dataset.

| Training set | | Test set | |
|---|---|---|---|
| Time range | Graph (Nodes) | Temporal shift | Strucal shift |
| Firtst 60% data in 2017 | 214 | Last 20% data in 2018-2021 | 500 new nodes & 0 removed nodes |

Table 10: OOD performance with rapidly growth on SD dataset.

| | | Method | Imp. | **Ours** | STONE | CaST | RPMixer | BigST | D²STGNN | STNN | STAEformer | STID | STNorm | GWNet | STGCN |
|---|---|---|---|---|---|---|---|---|---|---|---|---|---|---|---|
| All nodes | 3 | MAE | +3.06% | **18.04** | 18.61 | 21.47 | 25.20 | 18.85 | 20.98 | 42.24 | 18.99 | 18.78 | 19.14 | 22.62 | 20.61 |
| | | RMSE | +2.18% | **29.17** | 29.82 | 33.75 | 40.13 | 30.36 | 33.46 | 65.12 | 30.29 | 30.17 | 30.84 | 34.70 | 32.56 |
| | | MAPE | +2.14% | **12.32** | 13.74 | 15.92 | 15.64 | 12.59 | 14.50 | 33.68 | 14.87 | 12.91 | 15.79 | 17.83 | 15.15 |
| | 6 | MAE | +5.55% | **23.64** | 25.03 | 28.80 | 42.69 | 26.32 | 30.83 | 42.67 | 26.68 | 26.52 | 26.60 | 32.67 | 28.28 |
| | | RMSE | +4.56% | **38.10** | 39.92 | 44.71 | 66.85 | 41.79 | 47.76 | 65.66 | 41.82 | 41.93 | 42.20 | 49.31 | 44.40 |
| | | MAPE | +7.31% | **16.49** | 18.89 | 20.73 | 26.13 | 17.79 | 21.61 | 34.18 | 22.79 | 18.89 | 22.20 | 25.61 | 20.54 |
| | 12 | MAE | +16.35% | **32.29** | 38.97 | 41.95 | 77.90 | 38.60 | 45.12 | 46.48 | 38.76 | 39.44 | 38.59 | 48.05 | 40.62 |
| | | RMSE | +6.27% | **51.74** | 55.20 | 63.40 | 116.56 | 60.11 | 68.19 | 71.04 | 59.68 | 60.79 | 60.35 | 71.78 | 63.28 |
| | | MAPE | +14.11% | **22.95** | 26.80 | 31.80 | 49.35 | 26.72 | 31.83 | 36.88 | 33.16 | 30.24 | 35.07 | 40.56 | 29.35 |
| New nodes | 3 | MAE | +2.97% | **18.28** | 18.84 | 21.53 | 25.24 | 18.97 | 21.26 | 44.82 | 19.16 | 19.37 | 19.37 | 22.97 | 20.86 |
| | | RMSE | +2.13% | **29.47** | 30.11 | 33.67 | 39.93 | 30.34 | 33.81 | 68.42 | 30.43 | 30.22 | 31.06 | 35.27 | 32.98 |
| | | MAPE | +1.73% | **12.50** | 14.07 | 16.20 | 15.66 | 12.72 | 14.98 | 35.29 | 15.21 | 13.08 | 16.38 | 18.45 | 15.54 |
| | 6 | MAE | +5.40% | **24.02** | 25.39 | 28.94 | 42.76 | 26.55 | 31.34 | 45.26 | 27.00 | 26.79 | 27.02 | 33.27 | 28.71 |
| | | RMSE | +4.51% | **38.57** | 40.39 | 44.74 | 66.81 | 41.94 | 48.51 | 68.97 | 42.19 | 42.17 | 42.74 | 50.15 | 45.06 |
| | | MAPE | +6.94% | **16.77** | 19.40 | 21.04 | 26.13 | 18.02 | 22.34 | 35.86 | 23.34 | 19.22 | 23.22 | 26.58 | 21.08 |
| | 12 | MAE | +15.57% | **32.81** | 39.40 | 42.19 | 77.95 | 38.86 | 45.73 | 48.99 | 39.22 | 39.78 | 39.24 | 48.98 | 41.26 |
| | | RMSE | +6.05% | **52.31** | 55.68 | 63.55 | 116.44 | 60.25 | 68.94 | 74.23 | 60.24 | 61.08 | 61.32 | 72.91 | 63.97 |
| | | MAPE | +14.82% | **23.33** | 27.39 | 32.22 | 49.28 | 26.88 | 32.76 | 38.59 | 33.87 | 30.86 | 37.24 | 41.97 | 30.12 |

We observe that for baseline models based on Transformer and GCN, such as D²STGNN and GWNet, the rapid and large influx of new nodes significantly disrupts the model's learning of message passing mechanisms, leading to a decrease in performance for models relying on such global message passing mechanisms. Models like BigST based on linear attention mechanisms and STONE based on relaxed mapping perform better than the former in out-of-distribution (OOD) scenarios with rapid growth. On the other hand, STID, based on node independence, shows limitations in generalizing features to new nodes when faced with a large number of additional nodes. In contrast, STOP benefits from its innovative ConAU and GenPU-oriented low-order attention mechanism, capturing flexible adaptations to changes in the overall spatiotemporal environment through sub-environments, showing the highest relative improvement rate at 16.35% and demonstrating robustness in scenarios with rapid node growth.

## C.6 COMPARE CONTINUOUS LEARNING METHOD

We compared STOP with several continual learning methods on out-of-distribution (OOD) tasks. Taking the PEMS3-Stream dataset as an illustration, when encountering spatiotemporal shifts, these models require fine-tuning using 21-day data from the new distribution. To ensure a fair comparison, we aligned the OOD task settings by conducting tests directly in the subsequent years following the initial year of training. This training methodology is denoted as 'static-STModel' in TrafficStream (Chen et al., 2021), 'SurSTG-Static' in PEMCP (Wang et al., 2023b), and 'Static-TFMoE' in TFMoE (Lee & Park, 2024). We directly extracted their experimental results from the PEMS3-Stream dataset. For an intuitive comparison, we have added the predicted performance of STGCN.

As depicted in Table 11, the performance of continual learning strategies is notably inferior to traditional prediction models because they trade performance for accelerated training processes. And our model significantly surpasses existing continual learning models in OOD tasks.

It is noteworthy that in this experiment, the performance indicated by STOP is slightly superior to that in the primary experiment because *the results amalgamate the performance of testing data in the first year*, which was omitted in the primary experiment to emphasize the disparities in data distribution between the test and training sets as much as possible.

Table 11: Compared with spatiotemporal continuous learning methods on PEMS3-Stream dataset.

| Model | 15min | | | 30min | | | 60min | | |
|---|---|---|---|---|---|---|---|---|---|
| | MAE | RMSE | MAPE | MAE | RMSE | MAPE | MAE | RMSE | MAPE |
| PECMP | 13.37 | 21.10 | 28.35 | 14.78 | 23.54 | 30.88 | 16.32 | 27.20 | 34.28 |
| TrafficStream | 13.98 | 21.88 | 29.36 | 15.12 | 23.98 | 31.67 | 17.46 | 28.01 | 36.44 |
| TFMoE | 12.95 | 21.18 | 18.97 | 14.51 | 23.90 | 19.62 | 18.07 | 29.87 | 24.92 |
| STGCN | 13.27 | 21.03 | 16.64 | 14.47 | 23.64 | 18.03 | 17.05 | 27.95 | 21.04 |
| Ours | **11.37** | **19.16** | **15.38** | **12.41** | **21.18** | **15.92** | **14.24** | **24.39** | **18.51** |

## C.7  EFFICIENCY STUDY

The training time per epoch is depicted in Figure 6, showcasing the remarkable effectiveness and efficiency of STOP on the KnowAir dataset. Transformer-based models like STNN, STARformer, and D$^2$STGNN exhibit substantial computational time and high memory usage due to their utilization of self-attention mechanisms to calculate dependencies between node pairs, resulting in a time and space complexity that scales quadratically with the number of nodes. Similarly, GCN-based models rely on GCN mechanisms for spatial feature interactions, leading to a time complexity that is also quadratic with the number of nodes. In contrast, our model, with a complexity linear with the number of nodes, significantly reduces the computational complexity.

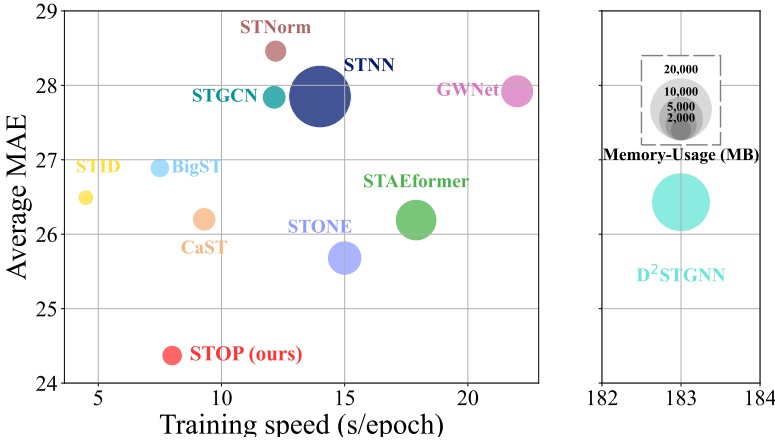

Figure 6: Visual case and efficiency study of STOP on KnowAir dataset.

## C.8  DETAILED PERFORMANCE ANALYSIS OF OOD IN EACH YEAR

In the main experiment, we reported the average OOD performance over multiple years. To provide a more detailed comparison, we present the performance changes of each model in the spatiotemporal OOD dataset for each year. As shown in Table 12 to 16, the results demonstrate that in fine-grained performance analysis, our model remains highly effective.

Table 12: OOD performance on LargeST-2018 dataset

| | | Method | Ours | CaST | RPMixer | BigST | STID | STNorm | GWNet | STGCN | STONE | D²STGNN | STNN | STAEformer |
|---|---|---|---|---|---|---|---|---|---|---|---|---|---|---|
| SD | 3 | MAE | **17.80** | 22.07 | 26.22 | 19.13 | 18.65 | 19.48 | 19.84 | 18.68 | 18.83 | 18.38 | 35.32 | 19.01 |
| | | RMSE | **28.23** | 34.38 | 41.83 | 30.76 | 29.87 | 30.75 | 30.62 | 29.29 | 29.97 | 28.51 | 55.12 | 30.25 |
| | | MAPE | **10.76** | 14.68 | 15.28 | 11.47 | 12.25 | 11.96 | 12.50 | 11.56 | 11.24 | 11.47 | 23.38 | 11.58 |
| | 6 | MAE | **23.40** | 30.03 | 44.55 | 26.41 | 26.01 | 26.76 | 26.97 | 24.53 | 26.38 | 23.98 | 35.77 | 26.21 |
| | | RMSE | **37.13** | 46.41 | 69.75 | 41.64 | 40.92 | 41.64 | 41.36 | 38.01 | 38.53 | 37.72 | 55.73 | 41.04 |
| | | MAPE | **14.52** | 19.82 | 25.81 | 16.28 | 16.57 | 16.71 | 17.98 | 15.22 | 15.02 | 15.03 | 23.72 | 16.22 |
| | 12 | MAE | **32.06** | 43.42 | 80.17 | 38.86 | 38.31 | 38.93 | 37.66 | 34.78 | 38.77 | 32.97 | 41.51 | 38.25 |
| | | RMSE | 51.45 | 66.43 | 120.41 | 60.05 | 59.78 | 60.02 | 58.90 | 53.80 | 51.94 | **51.27** | 64.44 | 59.22 |
| | | MAPE | **20.59** | 29.10 | 48.67 | 24.68 | 24.39 | 25.31 | 26.42 | 22.96 | 22.15 | 21.70 | 28.11 | 24.79 |
| GBA | 3 | MAE | **19.87** | 24.51 | 28.16 | 22.44 | 21.70 | 22.33 | 21.96 | 22.61 | 20.58 | 20.48 | 41.67 | 23.27 |
| | | RMSE | **32.71** | 38.63 | 45.26 | 36.86 | 35.55 | 36.07 | 34.62 | 36.12 | 36.49 | 35.06 | 62.49 | 37.74 |
| | | MAPE | **15.74** | 23.37 | 21.21 | 18.03 | 17.34 | 18.08 | 18.71 | 17.26 | 16.65 | 16.67 | 39.87 | 18.17 |
| | 6 | MAE | **25.44** | 32.60 | 45.23 | 30.73 | 29.52 | 30.96 | 28.76 | 30.21 | 26.15 | 26.05 | 41.57 | 31.18 |
| | | RMSE | **40.59** | 49.60 | 69.42 | 47.93 | 46.23 | 47.50 | 43.56 | 46.13 | 42.86 | 42.49 | 62.41 | 48.28 |
| | | MAPE | **23.19** | 32.35 | 37.04 | 27.13 | 27.55 | 27.63 | 26.81 | 26.59 | 24.30 | 24.63 | 39.73 | 27.59 |
| | 12 | MAE | **33.94** | 45.49 | 77.43 | 43.71 | 41.78 | 43.38 | 38.42 | 41.67 | 36.96 | 34.94 | 46.13 | 44.53 |
| | | RMSE | **53.45** | 67.92 | 113.45 | 65.75 | 63.68 | 64.22 | 57.12 | 61.36 | 56.02 | 54.96 | 68.65 | 66.98 |
| | | MAPE | **33.85** | 45.54 | 69.67 | 41.11 | 42.03 | 43.41 | 38.80 | 40.22 | 37.77 | 37.31 | 46.05 | 43.32 |
| GLA | 3 | MAE | **19.70** | 24.78 | 28.12 | 21.81 | 21.03 | 22.27 | 21.52 | 21.73 | | | | |
| | | RMSE | **31.31** | 37.70 | 44.37 | 34.75 | 33.54 | 34.76 | 33.47 | 33.99 | | | | |
| | | MAPE | **11.25** | 19.65 | 14.84 | 12.42 | 11.58 | 12.62 | 12.69 | 11.91 | | | | |
| | 6 | MAE | **26.38** | 33.17 | 47.11 | 30.71 | 29.39 | 31.64 | 29.57 | 30.22 | | | | |
| | | RMSE | **41.01** | 50.28 | 72.19 | 47.55 | 45.86 | 47.81 | 45.05 | 46.24 | | | | |
| | | MAPE | **16.20** | 25.56 | 26.16 | 17.46 | 16.73 | 18.70 | 17.76 | 17.50 | | | | |
| | 12 | MAE | **36.15** | 45.37 | 83.11 | 44.34 | 42.28 | 45.84 | 40.51 | 44.17 | | | | |
| | | RMSE | **55.61** | 68.36 | 121.95 | 66.37 | 64.77 | 67.03 | 60.64 | 65.32 | | | | |
| | | MAPE | **24.33** | 33.70 | 53.19 | 27.87 | 25.94 | 29.77 | 24.75 | 26.93 | | | | |
| CA | 6 | MAE | **18.66** | 23.86 | 26.15 | 20.59 | 20.07 | 20.41 | 19.96 | 20.84 | | Out of Memory | | |
| | | RMSE | **30.26** | 37.58 | 42.44 | 33.80 | 32.72 | 33.06 | 31.84 | 33.53 | | | | |
| | | MAPE | **13.24** | 18.53 | 17.31 | 14.28 | 13.71 | 14.50 | 14.02 | 14.24 | | | | |
| | 12 | MAE | **24.67** | 31.61 | 43.27 | 28.95 | 28.08 | 28.61 | 27.00 | 29.04 | | | | |
| | | RMSE | **39.12** | 49.24 | 67.85 | 45.85 | 44.60 | 45.05 | 41.99 | 45.49 | | | | |
| | | MAPE | **18.98** | 24.83 | 29.84 | 20.89 | 20.56 | 21.59 | 19.74 | 20.89 | | | | |
| | 24 | MAE | **33.54** | 44.03 | 75.81 | 41.51 | 40.52 | 40.85 | 37.02 | 42.04 | | | | |
| | | RMSE | **52.42** | 67.48 | 113.33 | 63.71 | 63.08 | 62.43 | 56.46 | 63.80 | | | | |
| | | MAPE | **27.78** | 36.25 | 57.70 | 32.88 | 31.63 | 32.32 | 30.10 | 32.09 | | | | |

Table 13: OOD performance on LargeST-2019 dataset

| | | Method | Ours | CaST | RPMixer | BigST | STID | STNorm | GWNet | STGCN | STONE | D²STGNN | STNN | STAEformer |
|---|---|---|---|---|---|---|---|---|---|---|---|---|---|---|
| SD | 3 | MAE | **18.42** | 22.51 | 26.61 | 19.71 | 19.51 | 19.56 | 21.12 | 19.63 | 19.42 | 19.54 | 37.94 | 19.52 |
| | | RMSE | **29.68** | 35.36 | 42.80 | 32.18 | 31.39 | 31.25 | 33.00 | 31.15 | 30.88 | 30.68 | 59.76 | 31.62 |
| | | MAPE | **11.73** | 16.02 | 16.03 | 12.27 | 13.18 | 12.65 | 14.57 | 12.99 | 12.76 | 12.91 | 28.17 | 12.58 |
| | 6 | MAE | **24.16** | 30.58 | 44.89 | 26.85 | 26.88 | 26.32 | 28.93 | 26.05 | 25.86 | 25.76 | 38.28 | 26.58 |
| | | RMSE | **38.73** | 47.64 | 70.91 | 43.07 | 42.76 | 41.85 | 44.61 | 40.84 | 40.54 | 39.74 | 60.37 | 42.38 |
| | | MAPE | **15.81** | 21.39 | 26.91 | 17.42 | 17.95 | 17.44 | 20.96 | 17.25 | 17.80 | 17.12 | 28.40 | 17.50 |
| | 12 | MAE | **32.78** | 43.96 | 80.12 | 39.17 | 39.31 | 38.24 | 40.49 | 37.07 | 36.89 | **35.59** | 43.02 | 38.22 |
| | | RMSE | **52.84** | 67.24 | 120.77 | 61.26 | 61.64 | 60.13 | 62.82 | 57.94 | 54.16 | 54.24 | 67.88 | 60.11 |
| | | MAPE | **22.22** | 31.30 | 50.17 | 26.46 | 26.42 | 26.47 | 30.52 | 26.08 | 26.00 | 25.06 | 32.33 | 26.42 |
| GBA | 3 | MAE | **19.95** | 23.90 | 27.07 | 21.52 | 21.01 | 21.98 | 21.95 | 22.43 | 22.26 | 21.46 | 41.80 | 22.59 |
| | | RMSE | **32.18** | 37.26 | 42.81 | 34.69 | 34.16 | 34.68 | 33.94 | 35.05 | 34.94 | 35.45 | 62.20 | 35.99 |
| | | MAPE | **15.45** | 21.66 | 19.51 | 16.54 | 15.95 | 17.48 | 17.87 | 16.16 | 16.04 | 16.84 | 38.21 | 16.74 |
| | 6 | MAE | **26.30** | 31.96 | 43.87 | 29.93 | 29.05 | 31.42 | 29.34 | 30.51 | 28.25 | 27.84 | 41.72 | 30.51 |
| | | RMSE | **40.91** | 48.34 | 66.45 | 45.96 | 45.13 | 46.99 | 43.72 | 45.62 | 42.30 | 43.83 | 62.10 | 46.78 |
| | | MAPE | **23.51** | 30.05 | 34.24 | 24.85 | 25.50 | 27.53 | 25.63 | 25.05 | 25.05 | 24.59 | 38.23 | 25.25 |
| | 12 | MAE | **36.07** | 44.88 | 76.42 | 43.52 | 41.75 | 44.80 | 39.71 | 43.12 | 36.93 | 37.50 | 45.94 | 43.56 |
| | | RMSE | **54.99** | 67.05 | 110.42 | 65.05 | 63.30 | 64.97 | 58.44 | 62.39 | 56.18 | 57.40 | 67.93 | 65.74 |
| | | MAPE | **34.75** | 42.42 | 65.49 | 37.91 | 38.94 | 43.22 | 37.05 | 38.59 | 35.69 | 36.41 | 43.50 | 38.83 |
| GLA | 3 | MAE | **19.69** | 24.47 | 27.31 | 21.23 | 20.76 | 21.51 | 21.69 | 21.10 | | | | |
| | | RMSE | **30.93** | 37.04 | 43.19 | 33.92 | 32.95 | 33.70 | 33.58 | 33.12 | | | | |
| | | MAPE | **11.74** | 21.06 | 15.06 | 12.69 | 12.01 | 12.88 | 13.66 | 12.35 | | | | |
| | 6 | MAE | **26.68** | 32.83 | 45.88 | 29.83 | 29.44 | 30.68 | 30.29 | 29.50 | | | | |
| | | RMSE | **41.06** | 49.58 | 70.56 | 46.45 | 45.54 | 46.59 | 45.96 | 45.29 | | | | |
| | | MAPE | **17.13** | 27.37 | 26.73 | 17.97 | 17.68 | 19.31 | 19.35 | 18.42 | | | | |
| | 12 | MAE | **36.79** | 45.09 | 81.30 | 42.90 | 42.73 | 44.89 | 41.81 | 43.33 | | | | |
| | | RMSE | **56.03** | 67.71 | 119.66 | 64.97 | 64.92 | 66.28 | 62.28 | 64.54 | | | | |
| | | MAPE | **25.92** | 36.07 | 55.21 | 28.69 | 27.67 | 31.03 | 27.11 | 28.64 | | | | |
| CA | 6 | MAE | **18.50** | 23.33 | 25.21 | 19.90 | 19.51 | 19.91 | 19.81 | 20.19 | | Out of Memory | | |
| | | RMSE | **29.65** | 36.55 | 40.72 | 32.59 | 31.74 | 32.19 | 31.40 | 32.40 | | | | |
| | | MAPE | **13.34** | 18.47 | 16.80 | 14.01 | 13.44 | 15.19 | 14.04 | 13.94 | | | | |
| | 12 | MAE | **24.94** | 30.93 | 41.86 | 28.08 | 27.58 | 28.22 | 27.27 | 28.23 | | | | |
| | | RMSE | **38.97** | 48.06 | 65.53 | 44.55 | 43.67 | 44.30 | 42.14 | 44.22 | | | | |
| | | MAPE | **19.55** | 24.73 | 29.04 | 20.55 | 20.30 | 22.95 | 20.04 | 20.56 | | | | |
| | 24 | MAE | **34.31** | 43.25 | 73.86 | 40.68 | 40.08 | 40.80 | 37.76 | 41.13 | | | | |
| | | RMSE | **52.57** | 66.25 | 110.23 | 62.63 | 62.31 | 62.15 | 57.27 | 62.72 | | | | |
| | | MAPE | **29.07** | 36.29 | 56.63 | 32.69 | 31.37 | 34.17 | 30.78 | 31.72 | | | | |

Table 14: OOD performance on LargeST-2021 dataset

| | Method | | Ours | CaST | RPMixer | BigST | STID | STNorm | GWNet | STGCN | STONE | D²STGNN | STNN | STAEformer |
|---|---|---|---|---|---|---|---|---|---|---|---|---|---|---|
| SD | 3 | MAE | **18.24** | 21.42 | 25.11 | 19.24 | 18.92 | 19.29 | 20.97 | 19.33 | 18.61 | 19.54 | 38.32 | 19.69 |
| | | RMSE | **29.23** | 33.16 | 39.71 | 30.70 | 30.10 | 30.74 | 32.61 | 30.60 | 30.01 | 30.73 | 58.72 | 27.31 |
| | | MAPE | **12.02** | 15.99 | 15.64 | 12.27 | 13.33 | 12.80 | 15.29 | 13.21 | 12.95 | 13.81 | 27.23 | 12.91 |
| | 6 | MAE | **24.22** | 29.31 | 42.46 | 26.47 | 26.50 | 26.78 | 29.11 | 26.33 | 26.09 | 26.55 | 38.64 | 27.21 |
| | | RMSE | **38.64** | 44.84 | 65.77 | 41.67 | 41.67 | 42.10 | 44.43 | 41.13 | 40.94 | 41.07 | 59.25 | 42.59 |
| | | MAPE | **16.31** | 21.48 | 26.12 | 18.57 | 18.41 | 17.96 | 21.94 | 17.82 | 17.58 | 18.35 | 27.34 | 18.10 |
| | 12 | MAE | **33.06** | 42.01 | 77.24 | 38.48 | 38.91 | 38.60 | 40.88 | 37.16 | 37.77 | 37.26 | 42.59 | 38.66 |
| | | RMSE | **52.31** | 62.75 | 114.49 | 59.15 | 60.06 | 59.64 | 62.34 | 57.35 | 56.07 | 56.52 | 65.51 | 59.78 |
| | | MAPE | **22.98** | 31.50 | 49.02 | 26.74 | 27.21 | 27.48 | 31.56 | 26.67 | 26.02 | 26.77 | 31.06 | 27.48 |
| GBA | 3 | MAE | **17.44** | 20.37 | 23.25 | 18.68 | 17.64 | 20.12 | 19.76 | 21.12 | 18.23 | 18.32 | 39.34 | 19.63 |
| | | RMSE | **28.33** | 32.32 | 37.24 | 30.49 | 29.58 | 31.67 | 30.91 | 32.83 | 29.56 | 31.67 | 57.83 | 31.84 |
| | | MAPE | 11.46 | 13.84 | 13.74 | 11.88 | 10.73 | 13.67 | 12.54 | 12.59 | **10.72** | 12.06 | 26.84 | 12.03 |
| | 6 | MAE | **24.12** | 27.81 | 38.87 | 27.59 | 25.21 | 31.23 | 27.70 | 30.22 | 25.87 | 25.42 | 39.18 | 26.96 |
| | | RMSE | **37.13** | 42.60 | 59.14 | 42.14 | 39.93 | 46.19 | 41.54 | 44.89 | 39.06 | 41.14 | 57.64 | 41.73 |
| | | MAPE | **17.16** | 19.18 | 23.40 | 17.63 | 16.38 | 22.01 | 18.15 | 18.99 | 17.21 | 17.75 | 26.75 | 17.39 |
| | 12 | MAE | **35.00** | 39.89 | 70.24 | 41.91 | 37.03 | 45.97 | 39.12 | 43.87 | 36.03 | 36.09 | 42.72 | 39.04 |
| | | RMSE | **52.48** | 59.20 | 100.88 | 60.59 | 56.12 | 64.95 | 57.59 | 62.31 | 58.97 | 55.86 | 62.52 | 58.17 |
| | | MAPE | **26.56** | 27.65 | 44.32 | 27.42 | 26.84 | 34.75 | 26.88 | 29.04 | 26.67 | 26.96 | 29.43 | 26.71 |
| GLA | 3 | MAE | **18.86** | 22.84 | 24.75 | 19.69 | 20.31 | 20.59 | 20.98 | 19.98 | | | | |
| | | RMSE | **30.05** | 34.85 | 39.70 | 31.81 | 32.13 | 32.59 | 32.77 | 31.76 | | | | |
| | | MAPE | **11.99** | 21.13 | 14.56 | 12.68 | 12.89 | 13.15 | 13.95 | 12.59 | | | | |
| | 6 | MAE | **26.22** | 30.69 | 41.37 | 27.88 | 28.59 | 30.16 | 29.73 | 28.42 | | | | |
| | | RMSE | **40.56** | 46.41 | 63.92 | 43.35 | 43.85 | 45.63 | 45.22 | 43.60 | | | | |
| | | MAPE | **18.04** | 27.41 | 25.56 | 18.23 | 18.60 | 20.36 | 20.05 | 19.14 | | | | |
| | 12 | MAE | **37.19** | 42.11 | 74.06 | 40.33 | 41.17 | 45.21 | 41.30 | 43.05 | | | | |
| | | RMSE | **56.17** | 62.84 | 110.23 | 60.29 | 61.56 | 65.44 | 61.33 | 62.86 | | | | |
| | | MAPE | **28.09** | 35.91 | 51.89 | 29.45 | 28.61 | 34.08 | 27.84 | 30.83 | | | | |
| CA | 6 | MAE | **16.89** | 21.15 | 22.68 | 17.93 | 17.58 | 18.46 | 18.47 | 18.46 | Out of Memory | | | |
| | | RMSE | **27.45** | 33.42 | 36.75 | 29.54 | 28.99 | 30.00 | 29.52 | 29.84 | | | | |
| | | MAPE | **11.87** | 16.48 | 14.87 | 12.56 | 12.02 | 15.25 | 12.76 | 12.60 | | | | |
| | 12 | MAE | **23.08** | 28.08 | 37.95 | 25.66 | 25.06 | 26.82 | 26.22 | 26.20 | | | | |
| | | RMSE | **36.07** | 43.74 | 59.24 | 40.57 | 39.82 | 41.83 | 40.67 | 40.85 | | | | |
| | | MAPE | **17.21** | 21.83 | 25.41 | 18.40 | 17.88 | 23.02 | 18.39 | 18.58 | | | | |
| | 24 | MAE | **32.27** | 39.63 | 68.35 | 37.99 | 36.71 | 39.46 | 37.29 | 39.01 | | | | |
| | | RMSE | **49.11** | 60.16 | 101.84 | 57.47 | 56.62 | 58.96 | 56.56 | 58.40 | | | | |
| | | MAPE | **26.04** | 31.94 | 49.28 | 29.39 | 27.37 | 34.64 | 28.40 | 29.37 | | | | |

Table 15: OOD performance of each year in PEMS3-Stream dataset.

| | Method | | Ours | STONE | CaST | RPMixer | BigST | D²STGNN | STNN | STAEformer | STID | STNorm | GWNet | STGCN |
|---|---|---|---|---|---|---|---|---|---|---|---|---|---|---|
| 2012 | 3 | MAE | 10.65 | 12.50 | 14.61 | 14.03 | 12.08 | 12.13 | 15.12 | 12.13 | 12.30 | 12.21 | 12.21 | 12.67 |
| | | RMSE | 16.92 | 18.96 | 22.30 | 21.71 | 18.59 | 18.67 | 23.67 | 18.71 | 18.78 | 18.67 | 18.72 | 19.33 |
| | | MAPE | 14.48 | 16.88 | 29.76 | 17.43 | 16.37 | 15.93 | 19.86 | 15.94 | 16.04 | 18.14 | 15.81 | 16.15 |
| | 6 | MAE | 11.49 | 13.33 | 16.14 | 16.50 | 13.14 | 13.12 | 15.03 | 13.27 | 13.31 | 13.24 | 13.14 | 13.69 |
| | | RMSE | 18.37 | 21.21 | 25.02 | 25.96 | 20.44 | 20.31 | 23.54 | 20.59 | 20.50 | 20.45 | 20.30 | 21.10 |
| | | MAPE | 15.74 | 17.58 | 31.36 | 20.21 | 18.27 | 16.95 | 19.70 | 18.91 | 17.82 | 19.43 | 17.18 | 17.66 |
| | 12 | MAE | 13.04 | 15.88 | 19.66 | 22.66 | 15.44 | 15.18 | 15.68 | 15.43 | 15.32 | 15.31 | 14.94 | 16.00 |
| | | RMSE | 21.02 | 23.86 | 30.80 | 36.24 | 24.17 | 23.63 | 24.52 | 24.32 | 23.89 | 23.88 | 23.25 | 24.89 |
| | | MAPE | 17.71 | 19.55 | 36.40 | 26.84 | 21.63 | 19.20 | 21.77 | 20.13 | 20.24 | 21.71 | 19.29 | 20.49 |
| 2013 | 3 | MAE | 10.90 | 12.74 | 15.23 | 14.35 | 12.35 | 12.42 | 16.17 | 12.40 | 12.53 | 12.50 | 12.51 | 12.99 |
| | | RMSE | 17.82 | 19.66 | 23.65 | 22.80 | 19.55 | 19.59 | 26.08 | 19.64 | 19.68 | 19.61 | 19.69 | 20.37 |
| | | MAPE | 14.57 | 16.41 | 31.18 | 17.55 | 16.60 | 15.95 | 20.75 | 16.02 | 16.09 | 19.44 | 15.91 | 16.23 |
| | 6 | MAE | 11.92 | 13.76 | 17.03 | 17.17 | 13.63 | 13.61 | 16.09 | 13.76 | 13.74 | 13.80 | 13.66 | 14.23 |
| | | RMSE | 19.62 | 21.46 | 26.87 | 27.85 | 21.76 | 21.64 | 25.93 | 21.99 | 21.75 | 21.89 | 21.72 | 22.52 |
| | | MAPE | 15.85 | 17.69 | 32.89 | 20.54 | 18.66 | 17.01 | 20.56 | 19.17 | 17.93 | 21.31 | 17.42 | 17.88 |
| | 12 | MAE | 13.71 | 15.55 | 21.07 | 23.95 | 16.22 | 16.05 | 16.81 | 16.31 | 16.11 | 16.21 | 15.88 | 16.83 |
| | | RMSE | 22.64 | 24.48 | 33.26 | 39.09 | 25.93 | 25.46 | 26.96 | 26.32 | 25.62 | 25.75 | 25.24 | 26.69 |
| | | MAPE | 18.00 | 19.84 | 38.16 | 27.53 | 22.37 | 19.45 | 22.66 | 20.41 | 20.64 | 23.90 | 19.81 | 20.83 |
| 2014 | 3 | MAE | 11.60 | 13.44 | 15.86 | 15.14 | 13.02 | 13.07 | 17.31 | 13.07 | 13.18 | 13.27 | 13.15 | 13.64 |
| | | RMSE | 19.31 | 21.15 | 24.81 | 23.94 | 20.52 | 20.85 | 28.39 | 20.73 | 20.61 | 20.94 | 20.90 | 21.43 |
| | | MAPE | 16.07 | 17.91 | 35.03 | 19.18 | 18.85 | 17.55 | 26.73 | 17.49 | 17.73 | 22.35 | 17.42 | 17.75 |
| | 6 | MAE | 12.58 | 14.42 | 17.62 | 17.87 | 14.28 | 14.16 | 17.84 | 14.35 | 14.33 | 14.70 | 14.20 | 14.82 |
| | | RMSE | 21.16 | 23.00 | 27.97 | 28.75 | 22.65 | 22.72 | 30.30 | 22.90 | 22.57 | 23.33 | 22.81 | 23.44 |
| | | MAPE | 17.24 | 18.08 | 36.71 | 22.10 | 21.53 | 18.55 | 31.25 | 20.92 | 19.67 | 24.90 | 18.91 | 19.45 |
| | 12 | MAE | 14.35 | 16.19 | 21.65 | 24.82 | 16.89 | 16.57 | 19.01 | 16.91 | 16.69 | 17.14 | 16.39 | 17.51 |
| | | RMSE | 24.39 | 26.23 | 34.58 | 40.36 | 27.00 | 26.75 | 30.76 | 27.42 | 26.54 | 27.49 | 26.47 | 27.95 |
| | | MAPE | 19.45 | 21.29 | 42.31 | 29.40 | 26.32 | 20.98 | 40.87 | 22.02 | 22.50 | 28.18 | 21.31 | 22.52 |
| 2015 | 3 | MAE | 11.55 | 13.39 | 15.53 | 14.85 | 12.92 | 12.99 | 17.40 | 12.92 | 13.09 | 13.21 | 13.10 | 13.50 |
| | | RMSE | 19.67 | 21.51 | 24.44 | 23.65 | 20.79 | 20.94 | 29.50 | 20.86 | 20.96 | 21.13 | 21.01 | 21.46 |
| | | MAPE | 15.67 | 17.51 | 33.81 | 18.68 | 18.07 | 17.13 | 25.68 | 16.95 | 17.21 | 21.55 | 16.87 | 17.22 |
| | 6 | MAE | 12.62 | 14.46 | 17.21 | 17.60 | 14.20 | 14.16 | 17.92 | 14.29 | 14.35 | 14.82 | 14.27 | 14.77 |
| | | RMSE | 21.98 | 23.82 | 27.71 | 28.92 | 23.29 | 23.20 | 31.31 | 23.51 | 23.54 | 24.10 | 23.43 | 23.93 |
| | | MAPE | 16.89 | 18.73 | 35.36 | 21.57 | 20.59 | 18.09 | 28.98 | 20.39 | 19.05 | 24.11 | 18.32 | 18.81 |
| | 12 | MAE | 14.29 | 16.13 | 20.85 | 23.99 | 16.66 | 16.49 | 18.80 | 16.77 | 16.59 | 17.20 | 16.39 | 17.23 |
| | | RMSE | 25.16 | 30.70 | 33.78 | 39.91 | 27.73 | 27.46 | 31.64 | 28.21 | 27.74 | 28.36 | 27.28 | 28.32 |
| | | MAPE | 19.01 | 20.85 | 40.51 | 28.42 | 25.01 | 20.45 | 35.42 | 21.32 | 21.67 | 27.24 | 20.59 | 21.67 |
| 2016 | 3 | MAE | 11.25 | 13.19 | 15.09 | 14.23 | 12.47 | 12.73 | 17.67 | 12.51 | 12.64 | 12.86 | 12.77 | 13.14 |
| | | RMSE | 22.46 | 24.30 | 26.12 | 25.12 | 22.56 | 23.77 | 32.55 | 23.54 | 22.80 | 23.12 | 23.29 | 23.69 |
| | | MAPE | 14.26 | 16.10 | 31.35 | 17.12 | 16.54 | 16.11 | 26.31 | 15.66 | 15.74 | 19.93 | 15.59 | 16.02 |
| | 6 | MAE | 12.37 | 14.21 | 16.74 | 16.95 | 13.75 | 13.94 | 18.15 | 13.85 | 13.90 | 14.53 | 14.00 | 14.41 |
| | | RMSE | 25.09 | 26.93 | 29.14 | 30.11 | 25.06 | 26.04 | 34.04 | 26.11 | 25.38 | 26.10 | 25.83 | 25.94 |
| | | MAPE | 15.52 | 17.36 | 32.92 | 20.01 | 18.93 | 17.14 | 30.98 | 18.79 | 17.57 | 22.21 | 17.07 | 17.52 |
| | 12 | MAE | 14.21 | 16.05 | 20.50 | 23.42 | 16.40 | 16.51 | 19.32 | 16.50 | 16.33 | 17.13 | 16.33 | 17.03 |
| | | RMSE | 28.89 | 35.73 | 35.16 | 41.23 | 29.87 | 30.78 | 35.16 | 31.11 | 30.02 | 30.83 | 30.13 | 30.43 |
| | | MAPE | 17.81 | 22.65 | 38.28 | 27.08 | 23.27 | 19.72 | 39.49 | 20.05 | 20.32 | 25.37 | 19.52 | 20.46 |
| 2017 | 3 | MAE | 12.54 | 14.38 | 16.26 | 15.49 | 13.88 | 14.01 | 18.61 | 13.84 | 14.02 | 14.13 | 14.06 | 14.37 |
| | | RMSE | 21.47 | 23.31 | 25.88 | 25.15 | 22.73 | 23.02 | 30.63 | 22.64 | 22.89 | 22.98 | 23.06 | 23.35 |
| | | MAPE | 15.72 | 17.56 | 31.78 | 18.18 | 17.35 | 16.78 | 22.47 | 16.82 | 17.14 | 21.20 | 16.85 | 16.90 |
| | 6 | MAE | 13.79 | 15.63 | 18.04 | 18.38 | 15.31 | 15.47 | 18.54 | 15.34 | 15.46 | 15.99 | 15.56 | 15.87 |
| | | RMSE | 23.81 | 25.65 | 29.05 | 30.08 | 25.23 | 25.63 | 30.53 | 25.20 | 25.43 | 26.12 | 25.77 | 25.99 |
| | | MAPE | 17.12 | 19.96 | 33.40 | 20.98 | 19.23 | 18.00 | 22.31 | 20.06 | 19.07 | 22.64 | 18.53 | 18.63 |
| | 12 | MAE | 16.01 | 17.85 | 22.04 | 25.15 | 18.27 | 18.48 | 19.50 | 18.34 | 18.31 | 19.23 | 18.28 | 18.88 |
| | | RMSE | 27.65 | 29.47 | 35.33 | 41.03 | 30.06 | 30.58 | 31.79 | 30.14 | 30.08 | 31.35 | 30.24 | 30.94 |
| | | MAPE | 19.59 | 21.43 | 38.75 | 27.76 | 22.93 | 20.89 | 24.62 | 21.74 | 22.08 | 25.40 | 21.22 | 21.84 |

Table 16: OOD performance of each year in KnowAir dataset from 2016 to 2018

| | Method | | **Ours** | STONE | CaST | RPMixer | BigST | D²STGNN | STNN | STAEformer | STID | STNorm | GWNet | STGCN |
|---|---|---|---|---|---|---|---|---|---|---|---|---|---|---|
| 2016 | 3 | MAE | **29.98** | 32.49 | 31.46 | 37.20 | 31.53 | 31.58 | 32.85 | 31.49 | 32.12 | 34.84 | 33.79 | 33.40 |
| | | RMSE | **47.95** | 49.23 | 49.01 | 55.72 | 48.21 | 48.05 | 49.42 | 48.28 | 50.06 | 54.14 | 51.66 | 50.09 |
| | | MAPE | **48.93** | 51.15 | 53.22 | 69.42 | 54.59 | 53.61 | 58.98 | 49.51 | 53.55 | 60.90 | 49.75 | 54.55 |
| | 6 | MAE | **32.92** | 35.47 | 34.03 | 46.82 | 34.96 | 35.14 | 36.11 | 34.49 | 36.62 | 37.02 | 36.32 | 36.64 |
| | | RMSE | 52.67 | 54.13 | **51.59** | 68.51 | 53.87 | 52.80 | 53.47 | 52.01 | 56.59 | 57.21 | 54.27 | 54.52 |
| | | MAPE | **50.10** | 62.40 | 60.37 | 90.19 | 62.45 | 60.73 | 65.56 | 55.48 | 61.12 | 65.40 | 55.88 | 62.49 |
| | 12 | MAE | **34.34** | 36.70 | 36.72 | 51.98 | 36.96 | 36.31 | 37.37 | **35.51** | 37.78 | 37.12 | 36.83 | 39.03 |
| | | RMSE | **54.14** | 56.03 | 56.21 | 77.45 | 56.03 | 53.54 | 57.76 | 53.69 | 57.42 | 54.81 | 53.97 | 57.19 |
| | | MAPE | **52.25** | 70.09 | 64.81 | 101.17 | 68.75 | 63.99 | 65.97 | 57.10 | 64.08 | 71.09 | 61.89 | 71.04 |
| 2017 | 3 | MAE | **20.98** | 23.87 | 23.57 | 28.21 | 22.25 | 23.52 | 24.42 | 23.14 | 23.30 | 25.54 | 24.81 | 25.56 |
| | | RMSE | **29.94** | 32.70 | 32.84 | 41.33 | 31.57 | 32.02 | 32.56 | 31.77 | 32.34 | 35.30 | 33.89 | 34.97 |
| | | MAPE | **51.29** | 53.57 | 63.10 | 75.25 | 58.00 | 60.40 | 66.94 | 54.19 | 60.12 | 70.36 | 55.37 | 60.32 |
| | 6 | MAE | **22.97** | 25.88 | 26.92 | 34.99 | 24.77 | 26.19 | 26.64 | 26.12 | 27.11 | 27.01 | 28.15 | 29.99 |
| | | RMSE | **32.35** | 34.79 | 36.65 | 49.30 | 34.54 | 35.53 | 35.78 | 35.29 | 36.87 | 36.09 | 37.24 | 39.70 |
| | | MAPE | **53.15** | 66.05 | 73.63 | 93.48 | 66.65 | 64.60 | 71.04 | 62.89 | 70.15 | 73.78 | 64.47 | 75.48 |
| | 12 | MAE | **24.89** | 26.79 | 28.55 | 37.48 | 27.14 | 26.88 | 26.95 | 27.15 | 29.25 | 28.88 | 29.99 | 33.17 |
| | | RMSE | **35.15** | 37.15 | 38.79 | 51.86 | 37.20 | 36.38 | 36.37 | 36.38 | 39.38 | 38.14 | 38.95 | 43.26 |
| | | MAPE | **58.07** | 70.38 | 80.71 | 93.46 | 75.56 | 68.35 | 74.25 | 66.59 | 77.75 | 85.63 | 73.96 | 84.94 |
| 2018 | 3 | MAE | **22.15** | 24.31 | 23.58 | 26.28 | 23.25 | 24.21 | 26.29 | 23.94 | 24.04 | 25.00 | 24.91 | 25.56 |
| | | RMSE | **31.79** | 35.55 | 33.41 | 38.96 | 33.00 | 33.65 | 35.22 | 33.41 | 34.32 | 34.99 | 35.20 | 34.97 |
| | | MAPE | **55.59** | 52.51 | 62.27 | 62.51 | 59.76 | 61.15 | 71.29 | 55.00 | 59.84 | 64.53 | 52.15 | 60.32 |
| | 6 | MAE | **25.18** | 27.96 | 27.53 | 33.54 | 27.16 | 28.86 | 28.68 | 27.75 | 28.84 | 28.54 | 28.84 | 29.99 |
| | | RMSE | **35.85** | 39.00 | 37.69 | 47.96 | 36.83 | 39.22 | 38.75 | 37.85 | 40.32 | 38.30 | 39.45 | 39.70 |
| | | MAPE | **60.09** | 72.52 | 76.44 | 79.13 | 76.08 | 75.96 | 76.56 | 66.55 | 74.04 | 76.32 | 64.18 | 75.48 |
| | 12 | MAE | **26.88** | 28.19 | 29.64 | 38.53 | 30.60 | 29.63 | 30.13 | 30.22 | 31.30 | 31.57 | 32.15 | 33.17 |
| | | RMSE | **37.88** | 43.26 | 40.62 | 54.58 | 40.32 | 39.71 | 40.03 | 40.38 | 43.21 | 41.44 | 42.48 | 43.26 |
| | | MAPE | **63.56** | 84.86 | 80.56 | 89.64 | 85.98 | 77.17 | 82.21 | 72.25 | 80.22 | 87.24 | 76.68 | 84.94 |

## C.9 ABLATION EXPERIMENT

We conduct thorough ablation experiments to evaluate the effectiveness of each component. The variants we created are shown in Table 17 and the experiments are shown in Table 17.

For the time module, we found that time decomposition and prompting provided the model with better capabilities to capture the temporal patterns from the sequence perspective, while the introduction of $\mathbf{Y}_t$ to make predictions from multiple components enhanced the model's robustness.

Regarding the C&S messaging mechanism, the "w/o ConAU" variant, which removes the spatial interaction module, resulted in a significant increase in error, indicating that the spatial interaction is still necessary in OOD scenarios. The "w/o LA" variant, which removes the low-rank attention mechanism in the C&S spatial interaction module, performed poorly in prediction, as the traditional node-to-node messaging mechanism is less robust to spatiotemporal shifts. The "w/o LA + DRO" variant performed better than the "w/o LA + RandomDrop" variant, demonstrating that the proposed graph perturbation mechanism is more effective than directly perturbing the dataset to generate diverse training environments in helping the model extract robust representations.

The "w/o DRO" variant exhibited a larger prediction error, suggesting that the inability to effectively optimize the deployed GenPU mask matrix increased the complexity of the model's learning process. The "w/o (GenPU&DRO)" variant also showed a considerable increase in error, further highlighting the crucial importance of the proposed graph perturbation mechanism in enhancing the model's robustness, as it allows the model to learn resilient representations from the perturbed environments.

These ablation studies can demonstrate the positive impact of each designed component on enhancing the overall performance of the model in out-of-distribution scenarios.

Table 17: Variants and their definitions in ablation experiment.

| Variant | Definition |
|---|---|
| w/o decom | Remove the decoupling mechanism |
| w/o prompt | Remove the temporal prompt learning |
| w/o (decom & prompt) | Remove the decoupling mechanism and temporal prompt learning |
| w/o $\mathbf{Y}_t$ | Remove the temporal prediction component |
| w/o $\mathbf{Y}_s$ | Remove the spatiotemporal prediction component |
| w/o ConAU | Completely remove the spatial centralized messaging mechanism |
| w/o LA | Use naive self-attention mechanism to replace Low-rank attention |
| w/o LA + GenPU | Add GenPU term with the variant w/o LA |
| w/o LA + GenPU +DRO | Add GenPU and spatiotemporal DRO with the variant w/o LA |
| w/o LA + RandomDrop | Randomly mask 20% training nodes and then train variant w/o LA |
| w/o DRO | Remove spatiotemporal DRO |
| w/o (GenPU) | Remove spatiotemporal DRO and GenPU |
| w/o (GenPU&DRO) + RandomDrop | Remove spatiotemporal DRO and GenPU and randomly mask 20% training nodes to simulate temporal and spatial shifts |

## C.10 ADDITIONAL SENSITIVITY EXPERIMENTS

In addition to the hyperparameter experiment in Section5.4 of the main body, we additionally deployed conduct experiments on four datasets—SD, GBA, GLA, and CA—to analyze the sensitivity of two hyperparameters, the number of ConAU $K$ and the number of GenPU $M$. The numebr of nodes *for training* in these six datasets range from 141 to 6615 nodes. The results on six datasets are shown in Figure 8.

**The number of ConAU** $K$. ConAU is the coarsening unit set up to interact with the node. Thus, the number of ConAU $K$ is closely related to the spatial scale. Based on our observations, we find that setting $K$ to approximately 1% of the spatial scale is a good choice. A larger number of ConAU can hinder the model's ability to focus on capturing generalizable contextual features.

**The number of GenPU** $M$. The hyperparameter $M$ represents the number of GenPU, which are used to modulate the interaction process between nodes and ConAU. Each GenPU corresponds to a different training environment. We have observed that the number of GenPU $M$ is universally

Table 18: Ablation experiments on SD and KnowAir datasets.

| Variant | SD | | | KnowAir | | |
|---|---|---|---|---|---|---|
| | MAE | RMSE | MAPE | MAE | RMSE | MAPE |
| **Ours** | **23.79** | **37.94** | **16.24** | **24.78** | **36.77** | **51.02** |
| w/o decom | 24.09 | 38.49 | 17.53 | 25.10 | 37.10 | 54.16 |
| w/o prompt | 24.67 | 39.83 | 18.20 | 25.27 | 36.78 | 51.42 |
| w/o decom & prompt | 25.23 | 40.46 | 19.01 | 25.83 | 37.25 | 54.33 |
| w/o $\mathbf{Y}_t$ | 23.87 | 38.02 | 16.86 | 25.70 | 36.99 | 53.10 |
| w/o $\mathbf{Y}_s$ | 26.25 | 41.25 | 18.76 | 27.04 | 39.21 | 63.68 |
| w/o ConAU | 26.06 | 41.47 | 17.56 | 26.88 | 38.22 | 58.23 |
| w/o LA | 26.14 | 41.86 | 18.26 | 25.62 | 37.10 | 53.12 |
| w/o LA + GenPU | 26.29 | 42.15 | 18.71 | 25.61 | 36.86 | 55.81 |
| w/o LA + GenPU + DRO | 26.11 | 41.73 | 17.58 | 25.10 | 36.91 | 54.73 |
| w/o LA + RandomDrop | 27.41 | 43.11 | 18.32 | 25.77 | 37.16 | 59.09 |
| w/o DRO | 24.08 | 38.17 | 17.06 | 24.93 | 37.24 | 54.86 |
| w/o (GenPU&DRO) | 24.52 | 38.65 | 18.13 | 25.26 | 36.98 | 55.12 |
| w/o (GenPU&DRO) + RandomDrop | 24.77 | 38.90 | 18.48 | 25.45 | 36.87 | 55.90 |

effective when set to between 2 and 4. When $M$ is set to a smaller value, an overly complex training environment can disrupt learning stability. Conversely, if there are too few GenPU, the limited training environments may not provide sufficient diversity for the model to extract invariant knowledge. Interestingly, this hyperparameter is insensitive to spatial scale.

We further analyze the sensitivity of this hyperparameter to the temporal span of the dataset. Long-range SD, GBA, GLA, and CA datasets contain a full year of training data, and TrafficStream is a short-range dataset containing one month data for training. And we can see that $M$ is not highly correlated with the time span of the data.

**Summary**. Based on the above analysis, we recommend setting the initial values $K$ to 1% of the number of training nodes and the initial values of $M$ between 2 and 4 for hyperparameter tuning in out-of-distribution (OOD) scenarios.

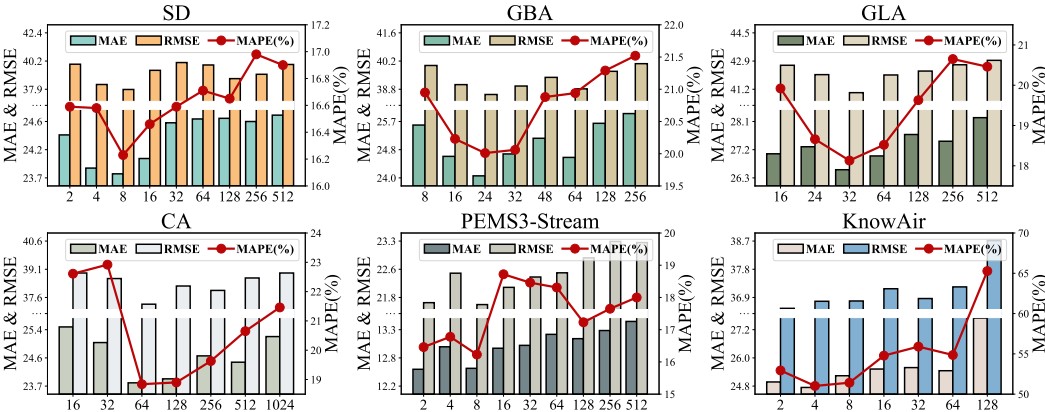

Figure 7: Sensitivity experiments of ConAU.

## D  DISCUSSION

The effectiveness of traditional spatiotemporal prediction models is typically demonstrated only in testing environments that closely resemble the training environment. While some studies on spatiotemporal OOD challenges have recognized the issues stemming from distribution shifts due to spatiotemporal variations and have proposed various strategies, however, both traditional models and OOD learning model reliance on node-to-node global interaction mechanisms constrains their generalization performance in the face of such shifts. To address this inherent limitation, we introduce an innovative spatiotemporal interaction mechanism that replaces the traditional node-to-node approach. This new mechanism incorporates ConAU that can perceive contextual features from

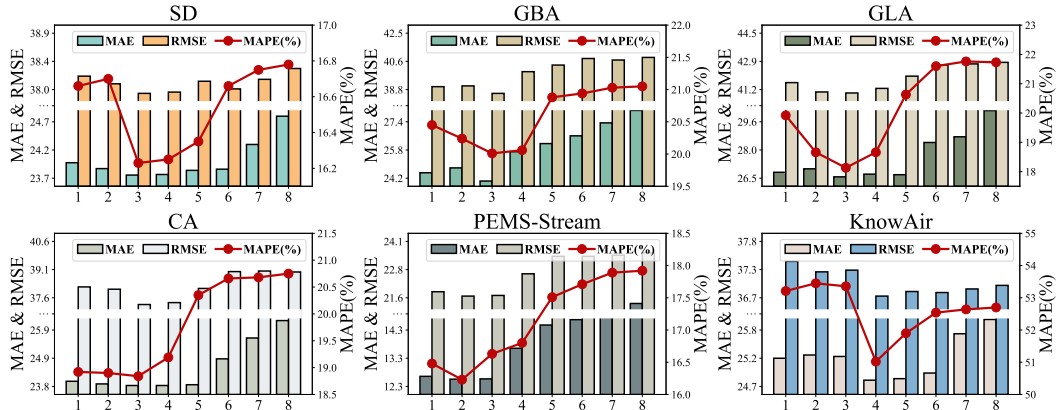

Figure 8: Sensitivity experiments of GenPU.

nodes, which helps maintain high generalization in unknown environments. Additionally, we design graph perturbation mechanism to further enhance robustness. Our method have been validated across eight OOD datasets, demonstrating performance improvements of up to +17.01%. More importantly, our findings provide valuable insights for future OOD researchers: (1) The core message-passing mechanisms in GCNs and Transformers are limited in OOD scenarios, indicating a need to explore alternatives beyond traditional GCN/Transformer with sequential model architectures; (2) A lightweight yet powerful architecture, such as Multi-Layer Perceptrons (MLPs), may be more suitable for OOD learning, as complex GCN or Transformer architectures can overfit to the training environment and compromise their generalization capabilities. However, there are still some limitations for future research:

**Exploring a Wider Range of OOD Scenarios**. Current OOD problems are typically defined within the confines of single-modal data and single tasks. However, spatiotemporal data exhibits diverse modalities and varied tasks. We believe that an improved spatiotemporal OOD handler should be capable of addressing challenges such as cross-task and cross-modal processing, areas that have not been thoroughly explored in the spatiotemporal domain.

**Integrating Large Language Models for zero-shot learning**. In OOD scenarios, accurately predicting new nodes poses a significant challenge, as these nodes have not been encountered by the model during training—commonly referred to as the zero-shot challenge. Large language models excel in this context, as their representational capabilities, developed from extensive training on massive datasets, can enhance a model's zero-shot learning ability. While this has been successfully demonstrated in the time series community, it remains relatively unexplored within the spatiotemporal domain. In future work, we plan to integrate large language models into the STOP framework to further enhance its scalability for predicting new nodes.

**Validating the Broad Impact of STOP**. The spatial interaction module integrated within the STOP framework is inherently generic, suggesting its potential for broader applicability. In upcoming research, we will propose replacing the graph convolutional networks utilized by other spatiotemporal backbones with the spatial interaction module to validate its effectiveness across various contexts. This initiative will help us better understand the potential value and applicability of the STOP module in a wide range of application domains.

# E  EXPLANATION OF CENTRALIZED MESSAGING MECHANISM

In this section, we analyze the centralized messaging mechanism. First, we demonstrate that the attention it uses satisfies the low-rank property, then explain its potential advantage: low computational complexity. Finally, we conduct qualitative analysis to illustrate how the centralized messaging mechanism exhibits enhanced resilience compared to global node-to-node message passing mechanisms (such as GCN or self-attention).

### E.1 LOW-RANK ATTENTION

In the centralized messaging mechanism, We first define low-rank attention as follows:

$$\mathbf{Z}_c^{(i)} = \mathcal{A}\left(\mathbf{Q}, \mathbf{K}, \mathbf{V}\right) = \underbrace{\mathrm{softmax}\left(\alpha\mathbf{Q}\mathbf{K}^\top\right)}_{\textbf{Diffusion}} \times \underbrace{\mathrm{softmax}\left(\alpha\mathbf{K}\mathbf{Q}^\top\right)}_{\textbf{Aggregation}} \mathbf{V}, \tag{20}$$

$$\text{where} \quad \mathbf{Q} = \mathbf{Z}_\mathrm{T}\mathbf{W}_q^{(i)} \in \mathbb{R}^{N \times d_h}, \ \mathbf{K} = \mathbf{C}\mathbf{J}_{d_t}^{(i)} \in \mathbb{R}^{N \times d_h}, \ \mathbf{V} = \mathbf{Z}_\mathrm{T}\mathbf{J}_{d_t}^{(i)} \in \mathbb{R}^{N \times d_h}. \tag{21}$$

Let $\mathbf{S}_a = \mathrm{softmax}\left(\alpha\mathbf{K}\mathbf{Q}^\top\right) \in \mathbb{R}^{K \times N}$ be the aggregation component of the attention score, and $\mathbf{S}_d = \mathrm{softmax}\left(\alpha\mathbf{Q}\mathbf{K}^\top\right) \in \mathbb{R}^{N \times K}$ be the diffusion component of the attention score, hence the attention score matrix $\mathbf{S} \in \mathbb{R}^{N \times N}$ can be expressed as

$$\mathbf{S} = \mathbf{S}_d \times \mathbf{S}_a \in \mathbb{R}^{N \times N}. \tag{22}$$

And the rank of $\mathbf{S}$ is satisfied,

$$\mathrm{rank}\left(\mathbf{S}\right) = \mathrm{rank}\left(\mathbf{S}_d \times \mathbf{S}_a\right) \leq \min\left(\mathrm{rank}\left(\mathbf{S}_d\right), \mathrm{rank}\left(\mathbf{S}_a\right)\right) \leq K \ll N, \tag{23}$$

The final inequality is a consequence of the fact that the maximum rank of a matrix is no more than the minimum of the ranks of its rows and columns (Greub, 2012). The rank of $\mathbf{S}$, up to $K$, is much lower than its size $N$, i.e., the number of rows and columns, hence the attention score matrix of our attention mechanism is a low-rank matrix. This constitutes the basis for the low ranking observed in our low-rank attention mechanism.

The low-rank characteristic in the centralized messaging mechanism offers two key advantages. Firstly, it exhibits linear complexity compared to the self-attention mechanism, allowing for a larger spatiotemporal efficiency. Secondly, it provides a lower error bound for the global node-to-node message passing mechanism, enhancing its resilience to errors.

### E.2 EFFICIENCY ANALYSIS

The low-rank attention function in Equation 20 can be rewritten as follows,

$$\mathcal{A}\left(\mathbf{Q}, \mathbf{K}, \mathbf{V}\right) = \mathbf{S}\mathbf{V} = \left(\mathbf{S}_d\mathbf{S}_a\right) \times \mathbf{V} = \mathbf{S}_d \times \left(\mathbf{S}_a\mathbf{V}\right), \tag{24}$$

Consequently, in contrast to the unlike vanilla self-attention mechanism (Vaswani et al., 2017), which necessitates the pre-computation of the attention score matrix with complexity $\mathcal{O}\left(N^2 d_h\right)$, we have the option of computing $\mathbf{S}_a\mathbf{V} \in \mathbb{R}^{K \times d_h}$ initially with complexity $\mathcal{O}\left(KNd_h\right)$ and subsequently determining $\mathbf{S}_d \times \left(\mathbf{S}_a\mathbf{V}\right) \in \mathbb{R}^{N \times d_h}$ with same complexity $\mathcal{O}\left(KNd_h\right)$, resulting in the efficient computation of low-rank attention with linear time complexity $\mathcal{O}\left(N\right)$ by $K \ll N$. As shown, we reduce the computational complexity from quadratic to nearly linear. This enables our method to effectively process graph data with a large number of nodes without requiring excessive GenPU memory resources. See Figure 5 and Figure 6 for experimental analysis.

## F ANALYSIS ON DISTRIBUTIONALLY ROBUST OPTIMIZATION

We theoretically analyzed STOP's generalization performance. Since STOP's optimization objective belongs to the distributionally robust optimization class (Duchi & Namkoong, 2019), which exhibits good generalization properties. Note that distributionally robust optimization class is a general term for optimization objectives that satisfy specific conditions - our contribution lies in how to implement optimization strategies that meet these conditions in the spatiotemporal OOD problem. First, we will introduce what constitutes a distributionally robust optimization class and the necessary conditions for membership, then analyze its beneficial properties, and finally extend these concepts to STOP.

### F.1 WHAT IS DRO?

Distributionally Robust Optimization (DRO) (Duchi & Namkoong, 2019) refers to a class of loss functions that aim to optimize by considering the worst-case scenario within a certain range of all

possible distributions of the data. In practical terms, an optimization object that takes the following form with respect to the training data distribution $e^*$ can be categorized under DRO (Duchi & Namkoong, 2019; Staib & Jegelka, 2019; Levy et al., 2020),

$$\arg\min_f \sup_{e \in \mathcal{E}} \left\{ \mathbb{E}_{(\mathbf{X},\mathbf{Y}) \sim p(\mathcal{X}, \mathcal{Y}|e)} \left[ \mathcal{L}\left( f\left( \mathbf{X} \right), \mathbf{Y} \right) \right] : \mathcal{D}\left( e, e^* \right) \leq \rho \right\}, \tag{25}$$

where $f$ is the function we optimized, usually a deep neural network with learnable parameters. $\mathcal{D}\left( \cdot, \cdot \right)$ is the distribution distance metric (Namkoong & Duchi, 2016; Shafieezadeh Abadeh et al., 2018), which is used to calculate the distance between distributions. $\rho$ is a hyperparamer to limit the extent to which the distribution is explored.

**Mark**. If an optimization satisfies: (1) modeling of different environments, (2) applying constraints, and (3) emphasizing the most challenging environments, then this optimization belongs to DRO and possesses the following beneficial properties.

### F.2 ADVANTAGES OF DRO

Recall that in the preliminary, the task of spatiotemporal OOD learning aims to learn a robust function $f$, which can accurately predict values after $T_P$ time steps given observed data of past $T$ time steps $\mathbf{X}$ and the graph sampled from any environment $e \sim \mathcal{E}$, where $e$ may have different spatiotemporal distributions with training environment $e^*$,

$$\arg\min_f \sup_{e \in \mathcal{E}} \mathbb{E}_{(\mathbf{X},\mathbf{Y}) \sim p(\mathcal{X}, \mathcal{Y}|e)} \left[ \mathcal{L}\left( f\left( \mathbf{X} \right), \mathbf{Y} \right) \right], \tag{26}$$

In a more intuitive sense, Equation. 1 is designed to find a function that reduces the loss associated with the most challenging scenario across all possible distributions $e \sim \mathcal{E}$. This task is particularly challenging because we lack access to data from any unfamiliar distributions outside of the training set (Qiao & Peng, 2023). Although traditional Empirical Risk Minimisation (Vapnik, 1998),

$$\arg\min_f \mathbb{E}_{(\mathbf{X},\mathbf{Y}) \sim p(\mathcal{X}, \mathcal{Y}|e^*)} \left[ \mathcal{L}\left( f\left( \mathbf{X} \right), \mathbf{Y} \right) \right], \tag{27}$$

which optimises solely based on the raw training environment $e^*$, performs well under the IID assumption, it is not possible to guarantee its performance in the presence of distributional drifts (Arjovsky et al., 2019). For all possible $e \in \mathcal{E}$ and function $f$, with high probability in mathematics, the following property holds,

$$\mathbb{E}_{(\mathbf{X},\mathbf{Y}) \sim p(\mathcal{X}, \mathcal{Y}|e)} \left[ \mathcal{L}\left( f\left( \mathbf{X} \right), \mathbf{Y} \right) \right] : \mathcal{D}\left( e, e^* \right) \leq \rho$$

$$\leq \mathbb{E}_{(\mathbf{X},\mathbf{Y}) \sim p(\mathcal{X}, \mathcal{Y}|e^*)} \left[ \mathcal{L}\left( f\left( \mathbf{X} \right), \mathbf{Y} \right) \right] + \mathcal{O}\left( \sqrt{ \frac{\mathrm{Var}_{(\mathbf{X},\mathbf{Y}) \sim p(\mathcal{X}, \mathcal{Y}|e^*)} \left[ \mathcal{L}\left( f\left( \mathbf{X} \right), \mathbf{Y} \right) \right]}{N_{e^*}} } \right), \tag{28}$$

where $N_{e^*}$ is the number of data point in traning environment. Therefore, due to the presence of subsequent variance terms, optimizing ERM alone cannot guarantee performance improvement in other environments $e' \in \mathcal{E} - \{e^*\}$. Compared to the IID-only condition of the ERM, distributionally robust optimization explores a certain range of challenging training data distributions, mathematically, distributionally robust optimization is equivalent to adding variance regularization to the standard ERM (Duchi & Namkoong, 2019),

$$\sup_{e \in \mathcal{E}} \left\{ \mathbb{E}_{(\mathbf{X},\mathbf{Y}) \sim p(\mathcal{X}, \mathcal{Y}|e)} \left[ \mathcal{L}\left( f\left( \mathbf{X} \right), \mathbf{Y} \right) \right] : \mathcal{D}\left( e, e^* \right) \leq \rho \right\}$$

$$= \mathbb{E}_{(\mathbf{X},\mathbf{Y}) \sim p(\mathcal{X}, \mathcal{Y}|e^*)} \left[ \mathcal{L}\left( f\left( \mathbf{X} \right), \mathbf{Y} \right) \right] + \sqrt{2\rho \, \mathrm{Var}_{(\mathbf{X},\mathbf{Y}) \sim p(\mathcal{X}, \mathcal{Y}|e^*)} \left[ \mathcal{L}\left( f\left( \mathbf{X} \right), \mathbf{Y} \right) \right]} + \varepsilon\left( f \right), \tag{29}$$

where $\epsilon(f) \geq 0$ and it is $\mathcal{O}(1/N_{e^*})$ uniformly about $f$. Therefore, if we do not consider the subsequent asymptotic terms $\epsilon(f)$, the above formula is equivalent to the following inequality,

$$\sup_{e \in \mathcal{E}} \left\{ \mathbb{E}_{(\mathbf{X},\mathbf{Y}) \sim p(\mathcal{X}, \mathcal{Y}|e)} \left[ \mathcal{L}\left( f\left( \mathbf{X} \right), \mathbf{Y} \right) \right] : \mathcal{D}\left( e, e^* \right) \leq \rho \right\}$$

$$\geq \mathbb{E}_{(\mathbf{X},\mathbf{Y}) \sim p(\mathcal{X}, \mathcal{Y}|e^*)} \left[ \mathcal{L}\left( f\left( \mathbf{X} \right), \mathbf{Y} \right) \right] + \sqrt{2\rho \, \mathrm{Var}_{(\mathbf{X},\mathbf{Y}) \sim p(\mathcal{X}, \mathcal{Y}|e^*)} \left[ \mathcal{L}\left( f\left( \mathbf{X} \right), \mathbf{Y} \right) \right]}. \tag{30}$$

DRO explores a certain range of training data distributions and tries to optimise on data distributions that may match the distribution of the test set, providing ideas for solving the OOD problem. Therefore, DRO mathematically provides more rigorous constraints than using empirical loss functions alone in OOD environments, preventing the model from over-relying on training data. This enables the model to flexibly adapt to different environments, improving its generalization performance in unknown environments.

### F.3 DOES STOP HAVE PROPERTIES OF DRO?

We will demonstrate that our optimization objective of STOP belongs to DRO, inheriting its good properties. Our optimization objective is as follows:

$$\min_{f} \sup_{\boldsymbol{g} \in \mathbb{R}^N} \mathbb{E}_{(\mathbf{X},\mathbf{Y}) \sim (\mathcal{X}, \mathcal{Y}|e^*)} \left[ \mathcal{L} \left( f\left(\mathbf{X}\right), \mathbf{Y}; \boldsymbol{g} \right) \right], \quad \text{s.t. } ||\widetilde{\boldsymbol{g}}||_0 = s \in (0, N). \tag{31}$$

Next, we demonstrate according to Mark 1 that our proposed optimization strategy satisfies the necessary conditions for DRO, thus inheriting its beneficial properties.

**Diverse environments**. STOP creates a diverse training environment by adding a perturbation process through a graph perturbation mechanism.

**Applying constraints**. Our perturbation process follows polynomial distribution sampling, and we strictly control the perturbation ratio, which imposes constraints on the generated environments.

**Exploring challenging environments:** We emphasize selecting environments with the largest gradients during training for optimization, encouraging the model to be exposed to challenging environments.

In summary, our optimization strategy belongs to DRO and thus inherits its good generalization property.

## G   NODE-TO-NODE MESSAGING LIMITATIONS

GCN has been proven to have powerful representation capabilities in various fields Wan et al.; Tan et al. (2024), which has been introduced by researchers in the field of spatiotemporal prediction. As explained in the introduction of the paper, node-to-node messaging mechanisms have the following limitations when dealing with spatiotemporal shifts: Limition.1. Coupled with the aggregation paths used during training (i.e., graph topology), structural shifts lead to inaccurate aggregation. Limition.2. Node representation errors flood throughout the entire graph, making it sensitive to temporal shifts of nodes. Limition.3. Inefficient induction ability for newly added nodes. Next, we explain three limitions and the limited role of node-to-node mechanisms in OOD scenarios.

### G.1   LIMITION.1

Using the SD dataset as an example, we first select the test data of 550 nodes and then input this data into the backbones, then we extract their output representations from their first layer that uses the node-to-node mechanism, denoted as $\alpha$.

Second, we remove 55 (10%) nodes of thse 550 nodes and add 55 new nodes, and take the new data into models again. Finally, we extract the output representations from the same layer, denoted as $\beta$.

After aligning the common nodes (495 nodes) between $\alpha$ and $\beta$, we calculate the representation error percentage using the following formula:

$$\frac{||\alpha - \beta||}{||\alpha||} \times 100\% \tag{32}$$

where $|| \cdot ||$ represents the Euclidean distance. The representation errors and final predicted performance gap are shown in the following table:

Table 19: Presentation errors due to spatial shifts.

| Model | GWNet | STGCN | STAEformer | D$^2$STGNN | **Ours** |
|---|---|---|---|---|---|
| Error | 8.71% | 6.64% | 12.96% | 11.81% | 2.68% |
| Performance gap | -25.47% | -14.71% | -20.42% | -32.63% | -1.04% |

The error percentage results demonstrate that structural shifts in the graph indeed affect GCN's accurate representation of the entire graph - even for STAEformer, the representation error percentage reaches 12.96% - thereby impacting their prediction performance.

## G.2 LIMITION.2

Using SD dataset as example again, we randomly select 30% of nodes from 550 nodes and added random noise to their data to simulate temporal shift of nodes. The errors are shown in the following table:

Table 20: Presentation errors due to temporal shifts.

| Model | GWNet | STGCN | STAEformer | D$^2$STGNN | **Ours** |
|---|---|---|---|---|---|
| Error | 2.35% | 7.29% | 6.13% | 9.17% | 1.23% |
| Performance gap | -25.47% | -19.87% | -4.34% | -15.05% | -0.83% |

When temporal distribution of nodes change, these models cannot accurately represent these nodes, and the errors also flood to the entire graph through the message passing mechanism, thereby degrading the performance of the entire graph.

## G.3 LIMITION.3

The existing node-to-node message passing mechanism's weak inductive learning capability limits models' ability to generalize learned knowledge to untrained nodes (Hamilton et al., 2017; Wang et al., 2023b). Yet new nodes frequently appear in the evoluting spatiotemporal graph. In Table 3 of the paper, we compare STOP with other models, clearly showing that our model achieves better inductive capability, with improvements of up to 15.07%. The potential reason is that the context-aware units established in our proposed interaction mechanism capture generalizable contextual features, which are common features shared across nodes. New nodes can access these features to obtain good representations.

## G.4 NODE-TO-NODE INTERACTION VS. OURS

We used two backbones: STGCN and STAEformer. The first one utilizes graph convolution as node-to-node interaction, while the latter use the self-attention mechanism for node-to-node interaction. We removed their node-to-node interaction layer and named these variants as "- graph". Additionally, we replaced their node-to-node interaction with our spatial interaction mechanism, denoting these variants as "+ Ours". We use SD and KnowAir datasets with OOD settings in our paper, and the performance results are shown in the following table:

Table 21: Add caption

| Variant | SD | | | KnowAir | | |
|---|---|---|---|---|---|---|
| | MAE | RMSE | MAPE | MAE | RMSE | MAPE |
| STGCN | 25.72 | 40.03 | 18.21 | 29.49 | 40.93 | 63.85 |
| STGCN - graph | 25.45 | 39.62 | 17.98 | 26.18 | 38.03 | 55.75 |
| STGCN + Ours | 24.87 | 38.98 | 17.65 | 25.44 | 37.42 | 52.80 |
| STAEformer | 26.20 | 41.18 | 18.39 | 27.25 | 38.93 | 56.48 |
| STAEformer - graph | 25.80 | 40.84 | 17.45 | 25.82 | 37.28 | 55.65 |
| STAEformer + Ours | 24.65 | 38.46 | 17.30 | 25.46 | 37.25 | 55.04 |

We can observe that after removing the node-to-node interaction mechanism, these variants surprisingly show better generalization performance. This demonstrates the limited (or even counterproductive) effect of node-to-node mechanisms. Meanwhile, our proposed spatial interaction module brings performance improvements, demonstrating that our proposed module is more effective than the node-to-node interaction.

