# OpenReview forum: "STOP! A Out-of-Distribution Processor with Robust Spatiotemporal Interaction"
_ICLR.cc/2025/Conference — Submitted to ICLR 2025_

### Official Review · Reviewer_4myW · 2024-10-24

**Soundness:** 2
**Presentation:** 2
**Contribution:** 2
**Rating:** 6
**Confidence:** 4

**Summary:**

In this paper, the authors focus on the problem of spatial-temporal out-of-distribution learning. Specifically, they model temporal and spatial information separately and combine them with DRO loss for prediction. At the same time, corresponding modules are designed for the capture of temporal and spatial correlation. For the structural drift problem, this method attempts to model the global correlation between node features. Empirically, experiments have been conducted on a large number of datasets to verify their effectiveness.

**Strengths:**

1. The research question is critical
2. The technical description is clear
3. Comparative experiments with SOTA methods were conducted on multiple datasets

**Weaknesses:**

In general, from my point of view, this is good work in terms of technology. However, the connection between the problem scenario and the technology used is weak, and there is a lack of certain verification and explanation. Although this method avoids the problem of structural distribution drift by discarding the original structural information and modeling the global correlation between features, it wastes data information to a certain extent. At the same time, this paper lacks ablation experiments on DRO.  Specifically:

1) The author mentioned that the reason why STGNN performs poorly under distribution drift is: global node-to-node messaging for spatiotemporal interaction. Is this verified? Can the attention propagation method mentioned in this paper be avoided?

2) In the field of spatial-temporal graph learning, many operations model temporal information and spatial information separately. At the same time, why can this operation alleviate the error accumulation mentioned? Further explanation is needed here, or corresponding references are provided.

3) Some nouns are not named properly. For example, CPUs and GPUs are proper nouns. Also, Client-Server (C2S), is there any scenario involving federated learning here?

4) The description of related work is too simple and unsystematic.

5) In the methods section, where is the low-rank reflected? Is it the low-rank of the similarity matrix? What is its advantage in this scenario? This needs explanation, and the intuition of using this technique is not clear.

6) There are no ablation experiments for the DRO learning strategy.

**Questions:**

Please see the weaknesses.

---

> ### Author Response · Authors · 2024-11-21
> **Weakness clarification**
>
> Dear Reviewer 4myW,
>
> We greatly appreciate your professional and valuable feedback, which is crucial for improving the quality of our paper.  We would like to address your concerns point by point. First, please allow us to provide an initial clarification regarding your summary of weaknesses.
>
> > **Although this method avoids the problem of structural distribution drift by discarding the original structural information and modeling the global correlation between features, it wastes data information to a certain extent.**
>
> Indeed, we don't use graph structural information in a fine-grained way; unlike traditional models, we don't perform thorough interaction through node-to-node messaging along the graph structural topology. However, we don't consider this a waste. On the contrary, it is precisely such careful use of graph structural information that causes the insufficient generalization capability of these models in structural shifts.
>
> To illustrate this point, we conducted an experiment using two OOD dataset  used in the paper: SD and KnowAir. We removed the graph convolution operations in STGCN and self-attention operations in STAEformer to eliminate the precise use of graph structural information, naming these variants STGCN-graph and STAEformer-graph. We replace their original components with our proposed interaction mechanism, and define the variables as STGCN+Ours and STAEformer+Ours. We report the performance of both variants and naive models in OOD scenarios in the following table:
>
> | | | SD| | | | KnowAir | |
> |:----:|:--:|:--:|:--:|--|-----|:--:|:--:|
> | Model| MAE| RMSE | MAPE | |MAE| RMSE| MAPE  |
> | STGCN| 25.72 | 40.03| 18.21 | |29.49 | 40.93| 63.85 |
> | STGCN-graph| 25.45 | 39.62| 17.98 | |26.18 | 38.03| 55.75 |
> | STGCN+Ours| **24.87** | **38.98**| **17.65** | |**25.44** | **37.42**   | **52.80** |
> || | | | ||| |
> | STAEformer | 26.20 | 41.18 | 18.39 | |27.25 | 38.93   | 56.48 |
> | STAEformer-graph | 25.80 | 40.84 | 17.45 | |25.82 | 37.28   | 55.65 |
> | STAEformer+Ours  | **24.65** | **38.46** | **17.20** | |**25.46** | **37.25**   | **55.04** |
> ||||
>
> The above results demonstrate that their meticulous utilization of graph structural information actually reduced generalization performance. Therefore, we abandoned the detailed utilization of graph information and proposed a more robust spatial interaction mechanism. As the table shows, our interactive component is more efficient.

---

> ### Author Response · Authors · 2024-11-21
> **W1. Node-to-node messaging**
>
> As explained in the introduction of the paper, node-to-node messaging mechanisms have the following limitations when dealing with spatiotemporal shifts: **Limition.1**. Coupled with the aggregation paths used during training (i.e., graph topology), structural shifts lead to inaccurate aggregation. **Limition.2**. Node representation errors flood throughout the entire graph, making it sensitive to temporal shifts of nodes.  **Limition.3**. Inefficient induction ability for newly added nodes. Next, we explain three limitions and the limited role of node-to-node mechanisms in OOD scenarios.
>
> ### **Limition.1**
>
> Using the SD dataset as an example, we first select the test data of 550 nodes and then input this data into four backbones, then we extract their output representations from their first layer that uses the node-to-node mechanism, denoted as $\alpha$. Second, we remove 55 (10%) nodes of thse 550 nodes and add 55 new nodes, and take the new data into models again. Finally, we extract the output representations from the same layer, denoted as $\beta$.
>
> After aligning the common nodes (495 nodes) between $\alpha$ and $\beta$, we calculate the representation error percentage using the following formula: $\frac{||\alpha-\beta||}{||\alpha||}\times100$\%, where $||\cdot||$ represents the Euclidean distance. The representation errors and final predicted performance gap are shown in the following table:
>
> | **Model** | **GWNet** | **STGCN** | **STAEformer** | **D2STGNN** |**Ours**|
> |:-:|:-:|:-:|:-:|:-:|:-:|
> | Error percentage | 8.71% | 6.64%| 12.96%| 11.81% |**2.68%**|
> |Performance gap| -25.47%  | -14.71% |-20.41%|-32.63%|**-1.04%**|
> |||
>
> The error percentage results show that structural shift affect the coupled aggregation paths of node-to-node messaging mechanisms, thereby impacting their accuracy in representing the entire graph. In contrast, our proposed interaction mechanism is more robust.
>
> ---
> ---
> ### **Limition.2**
>
> Using SD dataset as example again, we randomly select 30% of nodes from 550 nodes and added random noise to their data to simulate temporal shift of nodes, the representation is denoted as $\gamma$. The errors between $\alpha$ and $\gamma$ are shown:
>
> |**Model**|**GWNet**| **STGCN** | **STAEformer** | **D2STGNN** |**Ours**|
> |:--:|:--:|:--:|:--:|:--:|:--:|
> | Error percentage| 2.35%| 7.29%| 6.13% |9.17%|**1.23%**|
> |Performance Gap| -7.92%|  -19.87% |-4.34%|-15.05%|**-0.83%**|
> |||
>
> When temporal distribution of nodes change, these models cannot accurately represent these nodes, and the errors also flood to the entire graph through the message passing mechanism, thereby degrading the performance of the entire graph.
>
> ---
>
> ### **Limition.3**
>
> The weak inductive learning capability of  the node-to-node messaging limits the model's ability to accurately describe untrained nodes [1]. However, in OOD (Out-Of-Distribution) scenarios, new nodes frequently emerge. In Table 3 of our paper, we compared STOP with other models, and it clearly shows that our model has better inductive capability, with improvements of up to 15.07%.
>
> Ref:
>
> [1] Hamilton W, Ying Z, et al. Inductive representation learning on large graphs[J]. Advances in neural information processing systems, 2017.
>
> ---
> ### **Node-to-node interaction vs. Ours**
>
> We used two backbones: STGCN and STAEformer. The first one utilizes graph convolution as node-to-node interaction, while the latter use the self-attention mechanism for node-to-node interaction. We removed their node-to-node interaction layer and named these variants as '-graph'. Additionally, we replaced their node-to-node interaction with our spatial interaction mechanism, denoting these variants as '+ Ours'. We use SD and KnowAir datasets with OOD settings in our paper, and the performance results are shown in the following table:
>
> | | | SD| | | | KnowAir | |
> |:----:|:--:|:--:|:--:|--|-----|:--:|:--:|
> | Model| MAE| RMSE | MAPE | |MAE| RMSE| MAPE  |
> | STGCN| 25.72 | 40.03| 18.21 | |29.49 | 40.93| 63.85 |
> | STGCN-graph| 25.45 | 39.62| 17.98 | |26.18 | 38.03| 55.75 |
> | STGCN+Ours| **24.87** | **38.98**| **17.65** | |**25.44** | **37.42**   | **52.80** |
> || | | | ||| |
> | STAEformer | 26.20 | 41.18 | 18.39 | |27.25 | 38.93   | 56.48 |
> | STAEformer-graph | 25.80 | 40.84 | 17.45 | |25.82 | 37.28   | 55.65 |
> | STAEformer+Ours  | **24.65** | **38.46** | **17.20** | |**25.46** | **37.25**   | **55.04** |
> ||||
>
> We can observe that after removing the node-to-node interaction mechanism, these variants surprisingly show better generalization performance. **This demonstrates the limited (or even counterproductive) effect of node-to-node mechanisms.** Meanwhile, our proposed spatial interaction module brings performance improvements, demonstrating  that our proposed module is more effective than the node-to-node interaction.
>
> Summary. Our method effectively addresses these three limitations of traditional node-to-node messaging mechanisms and achieves better robustness for spatiotemporal OOD problem.

---

> ### Author Response · Authors · 2024-11-21
> **W2.  Error accumulation**
>
> In the introduction of this paper, we emphasize that **error accumulation stems from the stacked structure of existing spatiotemporal learning models**. These models stack multiple spatiotemporal modules sequentially, with each module depending on the output of the previous one. After spatiotemporal learning, these models only produce one representation, denoted as the prediction representation, which is finally input to the decoder for prediction. The weakness of this architecture is that errors caused by spatiotemporal shift accumulate and lead to large deviations in label representations, resulting in suboptimal accuracy.
>
> **Although temporal and spatial layers in each spatiotemporal module model temporal or spatial information separately , it cannot fundamentally address such issue**. Next, we illustrate the error accumulation due to structural and temporal shifts.
>
> ### **1. Error accumulation due to structural shift**
>
> First, we create two datasets  using the SD dataset with GWNet and D2STGNN. For the first dataset, we select the test data of 550 nodes and then input this data into the backbones, then we extract their output representations from each spatial or temporal layer. For the second dataset, we remove 55 (10%) nodes of thse 550 nodes and add 55 (10%) new nodes, and take the new data into models again.  Finally, we calculate the representation error percentage of each temporal or spatial layer, which shown in below:
>
> | $~~~$Layer | 1 | $~~~$2| $~~~~$3| $~~~~$4| $~~~~$5 | $~~~~$6 | $~~~~$7  | Prediction representation  |
> | :--: | :--: | :--: | :--: | :---: | :---: | :---: | :---: | :------------------------------: |
> | GWNet     | 0     | 2.77% | 8.61% | 13.28% | 17.14% | 20.88% | 23.52% | 27.20% |
> |D2STGNN|     0|       8.43%|   15.90%|19.03%|	    28.10%|  -  |  -   | -|
> |||
>
> The first layer is the temporal layer (TCN in GWNet and Transformer in D2STGNN) to model temporal information for each node, thus, errors is 0. Subsequently, they use node-to-node mechanism for node interaction, leading to representation errors due to structural shifts, and these error-containing representations are then fed into the next temporal module, affecting the learning of this layer.
>
> ### **2. Error accumulation due to temporal shift**
>
> We further randomly selected 30% of nodes from the first data set above and added random noise to their data for simulating temporal shift of nodes. The representation deviation at each layer is shwon below:
>
> |  $~~~$Layer | $~~~$1 | $~~~$2 | $~~~$3 | $~~~$4  | $~~~$5  | $~~~$6  | $~~~$7| Prediction representation  |
> |:---------:|:-----:|:-----:|:-----:|:------:|:------:|:------:|:------:|:------:|
> | GWNet     | 9.85%     | 11.87% | 14.40% | 16.01% | 17.44% | 18.62% | 19.26% | 23.43% |
> |D2STGNN|     23.39%|       25.45%|   26.84%|29.88%|	    34.00%|  -  |  -   | -|
> |||
>
> Changes in the data distribution of nodes lead to incorrect representation of these nodes by the model, and the message passing mechanism propagates these errors to other nodes.
>
> **Summary**. We can see that errors caused by either structural shifts or temporal shifts accumulate significant errors in prediction representation, ultimately leading to decreased prediction performance.
>
> ### **3. STOP for error accumulation**
>
> In response, **we simplify this stacked structure by using only one temporal layer and one spatial layer for spatiotemporal learning. These two layers each generate a prediction representation component, separately**. And the final prediction of our STOP is jointly determined by these two prediction components. Next, we verify the roles of these two prediction components in spatiotemporal shifts.
>
> Using the SD dataset and KnowAir dataset as examples, following the paper's settings, we created one dataset with only temporal shift and another with structural shift. We use the Shapley Value  to analyze the contributions of two prediction components in different OOD scenarios. The Shapley Value is a common indicator used to measure individual contributions in collaboration. We denote $\mathbf{Y}_t$ as the temporal prediction component and $\mathbf{Y}_s$ as the spatial prediction component.
>
> The results are shown in the following table:
>
> | Component | | SD   | KnowAir |       |
> |:-----------:|:-------:|:-------:|:---------:|:-------:|
> |           | SOOD  | TOOD  | SOOD    | TOOD  |
> | $\mathbf{Y}_t$       | 64.04 | 41.95 | 72.62   | 36.23 |
> | $\mathbf{Y}_s$       | 35.96 | 58.05 | 27.38   | 63.77 |
> |||
>
> The results reveal that in the SOOD dataset with structural distribution shift, the temporal prediction of STOP experiences less interference and its prediction contribution increases. Conversely, in TOOD scenarios characterized by temporal distribution shift, STOP would rely on spatial prediction components to make predictions. This separate prediction architecture allows STOP to maintain robustness across various OOD scenarios, as demonstrated in Tables 2 and 4 of the paper.

---

> ### Author Response · Authors · 2024-11-21
> **W3. Nouns**
>
> Thank you very much for your warm reminder.
>
> Our paper focuses solely on spatiotemporal OOD problems and does not involve federated learning scenarios.
>
> We recognize that our abbreviations caused confusion, and we have made the following efforts to address your concerns:
>
> **1. CPUs → ConAU**. This refers to Context Perception Units, to avoid any potential confusion, we will adopt the term ConPU (**Con**text **A**ware **U**nits) instead.
>
> **2. GPUs → GenPU**. GPUs originally stood for Generalized Perturbation Units, and we will revise this to GenPU (**Gen**eralized **P**erturbation **U**nits).
>
> **3. Client-Server interaction mechanism -> Centralized interaction mechanism**. We establish a small number of perception units that engage in centralized interactions with nodes, replacing traditional node-to-node decentralized interactions,  which we plan to denote as centralized interaction mechanism.
>
> 4.$~$Furthermore, except for widely recognized abbreviations (such as OOD, MLP, or STGNN), we use full terms rather than abbreviations for other terms to avoid potential confusion.
>
> Considering that these abbreviations are closely related to our method, we may not be able to implement these changes immediately in the recent revised version to avoid confusion among reviewers who may not synchronize with the abbreviation changes in a timely manner. We assure you that we will make the modifications after the discussion phase to prevent any overlap with commonly understood terms. Thank you once again for your valuable suggestions.

---

> ### Author Response · Authors · 2024-11-21
> **W4. Related work**
>
> We sincerely apologize for any confusion caused. In fact, we further discussed related work **in Appendix Section A**, including introductions to continual learning for spatiotemporal shifts and temporal OOD learning methods. In order of their relevance to our focus on OOD problems with spatiotemporal shifts, we have sequentially introduced four related subfields.
>
> (1). In the main body, we first introduced the progress of **traditional spatiotemporal prediction models**, whose limitation is that their good performance can only be demonstrated in independently and identically distributed environments.
>
> (2). In the main body,  we presented the progress in **spatiotemporal OOD learning work**. Unlike these works, our method proposes a novel spatial interaction mechanism to further enhance robustness.
>
> (3). In the appendix, we then discuss **the continual learning strategy for handling dynamic spatiotemporal graph data**.  However, since these strategies require training the model with new data to adapt to shifting distributions, they share the same limitation as traditional learning methods by adhering to the i.i.d. assumption.
>
> (4). Finally, we introduced works related to **temporal shift learning in time series**, which focus on addressing changes in statistical features. However, they ignore the dynamic nature of spatiotemporla grpah topology.
>
> To avoid confusion, we have refined the related work section and added clarifications in the main text to strengthen the connection between the main content and the related work section in the appendix. Additionally, we have summarized existing work using more refined language.

---

> ### Author Response · Authors · 2024-11-21
> **W5.  Low-rank**
>
> Sorry for the confusion. Based on the design inspiration of our spatial interaction mechanism, the attention matrix calculated by the corresponding mathematical formula is low-rank, as its rank is significantly lower than its maximum possible rank, hence we name it low-rank attention. Specifically, our attention matrix can be calculated as follows:
>
> $$
> S=\operatorname{softmax}(\alpha QK^\top)\operatorname{softmax}(\alpha KQ^\top)
> $$
>
> where $\alpha$  is a scaling factor and equals to $1 / \sqrt{d_{h}}$. $Q\in R^{N\times d_h}$, $K\in R^{K_n\times d_h}$, and $Q\in R^{N\times d_h}$ are the query, key, and value vectors, respectively $N$ is the number of nodes, and $K_n$ is the number of context perception units. Let $S_d=softmax(\alpha QK^\top)$ and $S_a=softmax(\alpha KQ^\top)$. The final attention matrix $S\in R^{N\times N}$ is a low-rank matrix because its rank is satisfied:
> $$
> \operatorname{rank}\left(\mathbf{S}\right)=\operatorname{rank}\left(\mathbf{S}_d\times\mathbf{S}_a\right)\leq\min\left(\operatorname{rank}\left(\mathbf{S}_d\right), \operatorname{rank}\left(\mathbf{S}_a\right)\right)\leq K_n\ll N
> $$
> The final inequality is a consequence of the fact that the maximum rank of a matrix is no more than the minimum of the ranks of its rows and columns. The rank of $\mathbf{S}$, which is up to $K_n$, is much lower than its size $N$, i.e., the number of rows and columns, hence the computed attention score matrix of our method is a low-rank matrix.
>
> The benefit of our used low-rand attention mechanism is that it is more efficient than the naive Transformer. The complexity of the traditional Transformer for spatial interaction is $\mathcal{O}\left(N^{2}d_h\right)$.  In our method, the complexity of calculating both $S_a$ and $S_b$ is $\mathcal{O}\left(K_n Nd_h\right)$. In this paper, $K_n\ll N$, thus, our low-rank attention has lower complexity. We thoroughly evaluated $K_n$ in Appendix C.10, which is generally 1% of $N$. We will emphasize this part in Appendix E.1 of the manuscript.

---

> ### Author Response · Authors · 2024-11-21
> **W6. Spatiotemporal DRO ablation**
>
> Sorry for the confusion. Since spatiotemporal DRO is a customized optimization strategy for generalized perturbation units ( temporarily GPU), our initially created variant 'w/o GPU' in the ablation experiments removed both DRO and generalized perturbation units together, and we overlooked conducting separate ablation experiments for DRO. To address your concern, we conducted ablation experiments across all datasets to evaluate spatiotemporal DRO's effectiveness, with the average MAE for 12 time steps shown in the following table.
>
>
> | $~$Variant | $~~$SD    | $~$GBA   | $~$GLA   | $~~$CA    | PEMS3-Stream | KnowAir |
> |:-------:|:-----:|:-----:|:-----:|:-----:|:------------:|:-------:|
> | w/o DRO | 24.08 | 25.94 | 26.86 | 24.10 | 12.71        | 24.93   |
> | w/o GPU | 24.52 | 26.10 | 27.24 | 24.39 | 12.99        | 25.26   |
> | Ours    | **23.79** | **25.09** | **26.53** | **23.84** | **12.54**        | **24.78**   |
> ||||||
>
> We can find that the prediction performance of w/o DRO is inferior because removing the proposed spatiotemporal DRO optimization strategy increases the model's learning complexity due to multiple training environments created through GPU's series of perturbations. spatiotemporal DRO, on the other hand, selects the most challenging one from these environments for training, thereby guiding the model to extract more robust knowledge. We have synchronized this discussion into the manuscript.

---

> > ### Comment · Reviewer_4myW · 2024-11-26
> >
> > Thanks to the authors for their informative response. It is interesting to find that discarding the original structure is beneficial. After reading the other reviewers' comments, I will maintain my original positive score.

---

> > > ### Author Response · Authors · 2024-11-26
> > >
> > > Dear Reviewer 4myW,
> > >
> > > Thank you very much for your professional suggestions. Your valuable and detailed review comments have significantly improved the quality of our paper. We sincerely thank you again for your time and effort.
> > >
> > > Best regards,
> > >
> > > The Authors

---

### Official Review · Reviewer_f4as · 2024-11-01

**Soundness:** 3
**Presentation:** 3
**Contribution:** 3
**Rating:** 6
**Confidence:** 3

**Summary:**

This paper presents the STOP model, designed to address OOD shifts in spatiotemporal prediction tasks. STOP employs an MLP channel-mixing backbone with robust interaction mechanisms, including C2S messaging and graph perturbation, to enhance resilience to spatiotemporal shifts. The C2S messaging mechanism utilizes CPUs for feature interactions, reducing dependence on global node-to-node messaging, while GPUs simulate spatiotemporal shifts to create diverse training environments. A customized DRO is also integrated to improve generalization.

**Strengths:**

This paper addresses the important and meaningful problem of spatiotemporal shifts. By introducing a lightweight MLP layer to model spatial and temporal dynamics, it effectively mitigates out-of-distribution (OOD) challenges while maintaining efficiency.

**Weaknesses:**

See questions part for more details.

**Questions:**

1. Section 5.4 only examines the hyperparameters \( K \) and \( M \), but since this paper focuses on model design, a more thorough exploration of hyperparameter choices (e.g., those discussed in Section 5.1) would be beneficial. Specifically, assessing the ease of finding optimal hyperparameters and their generalizability across downstream tasks is necessary. I feel the current discussion on these points is insufficient. Additionally, was the choice of \( M \) analyzed in Section 5.4 without being mentioned in Section 5.1?

2. The STOP model incorporates various component designs, including CPU and GPU interactions. While ablation studies that remove modules individually are common for validating component effectiveness, they often focus solely on the independent impact of specific components and may overlook interactions between modules. I suggest designing more comprehensive experiments, such as dual (or multiple) module ablations, to observe potential synergies or interdependencies among components.

3. This paper chooses an MLP as the backbone, differing fundamentally from the graph convolution architectures used in prior work. Aside from being more lightweight, what clear advantages does the MLP offer for addressing these types of problems?

---

> ### Author Response · Authors · 2024-11-21
> **W1.Detailed analysis of M and K**
>
> Dear Reviewer LDYw,
>
> Thank you very much for your valuable comments, which are essential for improving our paper. We would like to address your concerns point by point.
>
> We conduct experiments on six datasets to analyze the sensitivity of two hyperparameters. The spatial scales for training in these six datasets range from 141 to 6615 nodes. And we report the average performance over 12 time steps below.
>
> > **The number of CPUs $K$**
>
> The CPUs are coarsening units used for perceiving contextual features through interactions with nodes. Consequently, the number of CPUs $K$ is closely related to the spatial scale. Based on our experimental results, we find that setting $K$ to approximately 1% of the spatial scale is an effective choice. A larger number of CPUs can hinder the model's ability to focus on capturing generalizable contextual features.
>
> |            |        |       |   K     |       |       |       |       |
> |:------------:|:------:|:-----:|:-----:|:-----:|:-----:|:-----:|:-----:|
> | Dataset (The numer of training nodes)         | 2      | 4     |8    | 16    | 32    | 64    | 128   |
> | SD (550)           | 24.40   | 23.88 |  **23.79** | 24.03 | 24.59 | 24.65 | 24.66 |
> |              |        |       |       |       |       |       |       |
> | Dataset (The numer of training nodes)         | 8      | 16    | **24**    | 32    | 48    | 64    | 128   |
> | GBA (1809)          | 25.55  | 24.65 | **24.09** | 24.72 | 25.17 | 24.62 | 25.6  |
> |              |        |       |       |       |       |       |       |
> | Dataset (The numer of training nodes)       | 16     | 24    | **32**    | 48    | 64    | 128   | 256   |
> | GLA (2949)          |  27.05 | 27.28 | **26.53** | 26.98 | 27.68 | 27.65 | 27.46 |
> |              |        |       |       |       |       |       |       |
> | Dataset (The numer of training nodes)  | 16     | 32    | **64**    | 128   | 256   | 512   | 1024  |
> | CA (6615)           |  25.52 | 25.05 | **23.84** | 23.96 | 24.65 | 24.46 | 25.23 |
> |              |        |       |       |       |       |       |       |
> | Dataset (The numer of training nodes)    | 2      | 4     | **8**     | 16    | 32    | 64    | 128   |
> | PEMS3-Stream (655) |  12.52 | 12.99 | **12.54** | 12.96 | 13.02 | 13.25 | 13.16 |
> |              |        |       |       |       |       |       |       |
> | Dataset (The numer of training nodes)    | 2      | **4**     | 8     | 16    | 32    | 64    | 128   |
> | KnowAir (141)      | 25.01  | **24.78** | 25.27 | 25.55 | 25.61 | 25.48 | 27.68 |
> |||||
>
> > **The number of GPUs $M$.**
>
> The hyperparameter $M$ represents the number of GPU, which are used to modulate the interaction process between nodes and CPUs. Each GPU corresponds to a different training environment. The MAE is shown in Table below, and we have observed that the number of GPUs $M$ is universally effective when set to between 2 and 4. When $M$ is set to a smaller value, an overly complex training environment can disrupt learning stability. Conversely, if there are too few GPUs, the limited training environments may not provide sufficient diversity for the model to extract invariant knowledge. Interestingly, this hyperparameter is insensitive to spatial scale. And  $M$ is not highly correlated with the time span of the data. In the experiments, Long-range SD, GBA, GLA, and CA datasets contain a full year of training data, and TrafficStream is a short-range dataset containing one month data for training.
>
> | GPU (M)  |  1      | 2      | 3      | 4     | 5     | 6     |7| 8|
> |:------------:|:-----:|:-----:|:-----:|:-----:|:-----:|:-----:|:-----:|:-----:|
> | SD       | 24.02  | 23.90  | **23.79**  | 23.80 | 23.87 | 23.89 |24.31|24.79|
> | GBA      | 24.55  | 24.82  | **24.09**  | 25.72 | 26.17 | 26.60 |27.32|28.01|
> | GLA    |  26.77 | 26.98  | **26.53**  | 26.68 | 26.65 | 28.46 |28.78|30.25|
> | CA       |  24.00 | 23.90   | **23.84**  | 23.84 | 23.87 | 24.85 |25.64|26.30|
> | PEMS3-Stream | 12.64  | **12.54** | 12.55  | 13.63 | 14.45 | 14.64 |14.90|15.21|
> | KnowAir  | 25.21  | 25.27  | 25.24 | **24.78** | 24.81 | 24.92 |25.69|25.97|
> |||||
>
>
> Summary. Based on the above analysis, we recommend setting the initial values $K$ to 1\% of the number of training nodes and the initial values of $M$ between 2 and 4 for hyperparameter tuning in OOD scenarios.
>
> This discussion is supplemented in the Appendix Section C.10. We also have supplemented the missing report on $M$ in Section 5.1. Thank you for your kind reminder about our oversight.

---

> ### Author Response · Authors · 2024-11-21
> **W2. Multiple module ablation experiment.**
>
> We conduct thorough ablation experiments to evaluate the effectiveness of each component. The variants we created are shown below.
> |||
> |:--------:|:---------:|
> | **Variant** | **Definition**  |
> | w/o decom                | Remove the decoupling mechanism                                                                 |
> | w/o prompt               | Remove the temporal prompt learning                                                             |
> | w/o (decom & prompt)     | Remove the decoupling mechanism and temporal prompt learning                                    |
> |w/o $\mathbf{Y}_t$ | Remove the temporal prediction component|
> |w/o $\mathbf{Y}_s$| Remove the spatiotemporal prediction component|
> | w/o DRO                  | Remove spatiotemporal DRO                                                                                      |
> | w/o (GPU&DRO)            | Remove spatiotemporal DRO and GPU                                                                              |
> | w/o (GPU&DRO)+RandomDrop | Remove spatiotemporal DRO and GPU and randomly mask 20% training nodes to simulate temporal and spatial shifts |
> | w/o LA                   | Use Naive self-attention mechanism to replace Low-rank attention                                |
> | w/o LA+GPU              | Add GPU term with the variant w/o LA                                                            |
> | w/o LA +GPU +DRO        | Add GPUs and spatiotemporal DRO with the variant w/o LA                                                        |
> | w/o LA+RandomDrop        |  Randomly mask 20% training nodes and then train variant w/o LA                                 |
> | w/o CPU                 | Completely remove the spatial interaction mechanism                                             |
> ||||||||
>
> The experiments on two datasets are shown below. Regarding the C\&S messaging mechanism, the "w/o CPU" variant, which removes the spatial interaction module, resulted in a significant increase in error, indicating that the spatial interaction is still necessary in OOD scenarios. The "w/o LA" variant, which removes the low-rank attention mechanism in the C\&S spatial interaction module, performed poorly in prediction, as the traditional node-to-node messaging mechanism is less robust to spatio-temporal shifts. The "w/o LA+DRO" variant performed better than the "w/o LA+RandomDrop" variant, demonstrating that the proposed graph perturbation mechanism is more effective than directly perturbing the dataset to generate diverse training environments in helping the model extract robust representations.
>
> The "w/o DRO" variant exhibited a larger prediction error, suggesting that the inability to effectively optimize the deployed GPU mask matrix increased the complexity of the model's learning process. The "w/o (GPU\&DRO)" variant also showed a considerable increase in error, further highlighting the crucial importance of the proposed graph perturbation mechanism in enhancing the model's robustness, as it allows the model to learn resilient representations from the perturbed environments.
>
> These ablation studies can demonstrate the positive impact of each designed component on enhancing the overall performance of the model in out-of-distribution scenarios.
>
> |                                |      |     |        |  |     |       |
> |:------------------------------:|:------:|:------:|:------:|:-------:|:-----:|:-----:|
> |                                |      |   **SD**     |        |  |   **KnowAir**   |       |
> | Variant                        | MAE    | RMSE   | MAPE   | MAE     | RMSE  | MAPE  |
> | w/o decom                      | 24.09  | 38.49  | 17.53  | 25.10   | 37.10 | 54.16 |
> | w/o prompt                     | 24.67  | 39.83  | 18.20  | 25.27   | 36.78 | 51.42 |
> | w/o decom & prompt             | 25.23  | 40.46  | 19.01  | 25.83   | 37.25 | 54.33 |
> | w/o $\mathbf{Y}_t$                      | 23.87  | 38.02  | 16.86  | 25.70   | 36.99 | 53.10 |
> | w/o $\mathbf{Y}_s$| 26.25|41.25|18.76|27.04|39.21|63.68|
> |||||||
> |w/o CPU                       |26.06 |41.47 |17.56|26.88| 38.22| 58.23|
> | w/o LA                     | 26.14  | 41.86  | 18.26  | 25.62   | 37.10 | 53.12 |
> | w/o LA+GPU               | 26.29  | 42.15  | 18.71  | 25.61   | 36.86 | 55.81 |
> | w/o LA +GPU +DRO          | 26.11  | 41.73  | 17.58  | 25.10   | 36.91 | 54.73 |
> | w/o LA+RandomDrop        | 27.41  | 43.11  | 18.32  | 25.77   | 37.16 | 59.09 |
> |||||||
> | w/o DRO                        | 24.08  | 38.17  | 17.06  | 24.93   | 36.92 | 54.86 |
> | w/o (GPU&DRO)                  | 24.52  | 38.65  | 18.13  | 25.26   | 36.98 | 55.12 |
> | w/o (GPU&DRO)+RandomDrop| 24.77  | 38.90  | 18.48  | 25.45   | 36.87 | 55.90 |
> |||||||
> | Ours                           | **23.79**  | **37.94**  | **16.24**  | **24.78**   | **36.77** | **51.02** |
> |||||||
>
> We have added this ablation experiment to Section C.9 of the Appendix.

---

> ### Author Response · Authors · 2024-11-21
> **W3. The reason to choose MLP**
>
> Because sequential models like LSTM or GRU lack the ability to model spatial features and are unsuitable for handling graph-structured data, and popular node interaction structures like GCN and Transformer have limitations in OOD scenarios, we chose MLP, because this model is lightweight, which **not only provides high computational efficiency but also helps prevent overfitting to training data, thereby achieving higher generalization performance**. Of course, thanks to our clever utilization of MLP, we've enabled it to achieve sufficient spatiotemporal modeling capabilities.
>
> Specifically, in the third paragraph of the introduction, we explain that node-to-node messaging mechanisms, as core elements of popular GCN/Transformer structures, struggle to effectively handle spatiotemporal changes in OOD scenarios. This is because the graph knowledge learned through this propagation mechanism is coupled with the training graph topology. When the test environment undergoes changes in graph topology, alterations in feature aggregation paths hinder the model's ability to effectively represent the graph.
>
> To further clarify the concerns, we design three variants of the temporal module in STOP for temporal modeling, replacing MLP with TCN or LSTM for temporal modeling along the time dimension. Experimental results show that these variants exhibit poor generalization performance, and our competitive performance  is attributed to our innovative application of MLP, whose temporal modeling capability surpasses these sequential models.
>
> ||||||||
> |-------------------|-------|-------|-------|---------|-------|-------|
> | Variant  |     |   **SD**    |       |  |   **KnowAir**   |       |
> |                   | MAE   | RMSE  | MAPE  | MAE     | RMSE  | MAPE  |
> | MLP → TCN         | 24.23 | 37.99 | 17.66 | 25.21   | 37.06 | 52.65 |
> | MLP → LSTM        | 24.50 | 38.51 | 16.97 | 25.46   | 38.65 | 52.88 |
> | Ours              | **23.79** | **37.93** | **16.23** | **24.77**   | **36.77** | **51.01** |
> ||||||

---

> ### Comment · Reviewer_f4as · 2024-11-22
> **Feedback on Author's Revisions**
>
> Thank you for your prompt response, which includes extensive additional experiments. The results and analyses address my concerns and questions to a considerable extent, and I am willing to raise my rating for the paper.

---

> > ### Author Response · Authors · 2024-11-26
> >
> > Dear Reviewer f4as,
> >
> > We sincerely appreciate your detailed feedback and encouraging comments. We are delighted that our revisions have addressed your concerns satisfactorily. Your thoughtful insights have been invaluable in enhancing the quality of our paper. Thank you once again for your time and review.
> >
> > Warm regards,
> >
> > The authors.

---

### Official Review · Reviewer_LDYw · 2024-11-03

**Soundness:** 2
**Presentation:** 3
**Contribution:** 2
**Rating:** 6
**Confidence:** 3

**Summary:**

This paper proposes the Spatio-Temporal OOD Processor (STOP) to enhance the generalization of spatiotemporal graph convolutional networks in out-of-distribution (OOD) environments. STOP uses spatiotemporal MLP channel mixing to separately incorporate temporal and spatial elements. It introduces Context Perception Units (CPUs) and Generalized Perturbation Units (GPUs) to capture generalizable context features and create diverse training environments, thereby improving robustness against spatiotemporal shifts. Additionally, a customized Distributionally Robust Optimization (DRO) is developed to further enhance generalization.

**Strengths:**

1. meticulous work and comprehensive experiments: The paper presents a thorough and detailed introduction and study, with extensive experiments conducted across multiple datasets.

2. effective illustrations: The figures and diagrams included in the paper are well-designed and enhance the reader's understanding of the mechanisms introduced.

**Weaknesses:**

1. excessive use of abbreviations leading to confusion: The paper employs an abundance of abbreviations, some of which overlap with commonly understood terms like CPUs and GPUs. Abbreviations should be distinctive and aid in communication, rather than repurposing established terms in a way that might confuse the reader.

2. lack of original innovation and theoretical justification: The proposed method appears to be a combination of existing modules and techniques borrowed from other works, giving an impression of an engineering patchwork rather than introducing novel contributions. The authors have not provided sufficient theoretical explanations to justify why their design choices effectively address the research problem.

3. lack of discussion regarding the potential limitations of the proposed method.

**Questions:**

See weaknesses.

---

> ### Author Response · Authors · 2024-11-21
> **W1.  Abbreviations**
>
> Dear Reviewer LDYw,
>
> Thank you very much for your valuable comments, which are essential for improving our paper. We would like to address your concerns point by point.
>
> > **Abbreviations**
>
> **(1). CPUs → ConAU.** This refers to Context Perception (Aware) Units, to avoid any potential confusion, we will adopt the term ConPU (**Con**text **A**ware **U**nits) instead.
>
> **(2). GPUs → GenPU.** GPU originally stood for Generalized Perturbation Units, and we will revise this to GenPU (**Gen**eralized **P**erturbation **U**nits).
>
> (3). Furthermore, except for widely recognized abbreviations (such as OOD, MLP, or STGNN), we use full terms rather than abbreviations for other terms to avoid potential confusion.
>
>
> Considering that these abbreviations are closely related to the contributions of our method, we may not be able to implement these changes immediately in the revised version to avoid confusion among reviewers who may not synchronize with the abbreviation changes in a timely manner. We assure you that we will make the necessary modifications after the discussion phase to prevent any overlap with commonly understood terms. Thank you once again for your valuable suggestions.

---

> ### Author Response · Authors · 2024-11-21
> **W2 (1). Original innovation and theoretical justification.**
>
> Our contribution lies in developing a new spatial interaction mechanism equipped with a novel graph perturbation mechanism and aspatiotemporal DRO optimization strategy, addressing the a limitation that has not yet been discussed: node-to-node interactionmechanism that existing spatiotemporal models rely on. **This mechanism is not an engineering patchwork; instead, each component is configured with high synergy and purposeful motivation in addressing OOD challenges**. Below, I will explain the design logic of each component.
>
> (1) Unexplored motivation. We argue that the node-to-node interaction mechanism used in existing spatiotemporal prediction models is the potential reason for their poor robustness to OOD distributions, as it is sensitive to spatiotemporal changes. This limitation has not been explored.
>
> (2) Novel core methodology. We attempt to design a robust spatial interaction mechanism that incorporates context-aware units for node interaction, rather than through direct node-to-node interaction, thereby enhancing resilience to spatiotemporal shift. These context-aware units can sense high-level contextual features from nodes.
>
> (3) Effective enhancement strategies. We recognize that diverse training environments enable models to extract robust contextual features that can generalize to unknown scenarios; however, data-level perturbations are computationally expensive. Therefore, we propose a novel graph perturbation mechanism that perturbs the aforementioned spatial interaction process during training, effectively modeling diverse environments.
>
> (4) Customized optimization strategy. Recognizing that excessive training environments can complicate model learning, we introduce a specialized optimization strategy that carefully exposes the model to challenging environments to enhance robustness.
>
> In summary, we systematically build a comprehensive OOD learning framework, progressing from fundamental observations to sophisticated technical innovations.
>
> To avoid any further confusion, we will refine the method overview section in Section Introduction to emphasize our key contributions clearly.

---

> ### Author Response · Authors · 2024-11-21
> **W2 (2). Theoretical  explanation**
>
> We theoretically analyzed STOP's generalization performance. Since STOP's optimization objective belongs to the distributionally robust optimization class, which exhibits good generalization properties. Note that distributionally robust optimization class is a general term for optimization objectives that satisfy specific conditions - our contribution lies in how to implement optimization strategies that meet these conditions in the spatiotemporal OOD problem. First, we will introduce what constitutes a distributionally robust optimization class and the necessary conditions for membership, then analyze its beneficial properties, and finally extend these concepts to STOP.
>
> > **What is distributionally robust optimization class?**
>
> Distributionally robust optimization [1] refers to a class of optimization objectives that aims to optimize for the worst-case scenario within a certain range of all possible data distributions, and the mathematical formula is expressed as:
> $$
> \arg\min_{f}\sup_{e\in\mathcal{E}}\lbrace\mathbb{E}_{(\mathbf{X},\mathbf{Y})\sim p(\mathcal{X},\mathcal{Y}|e)}\[\mathcal{L}(f(\mathbf{X}),\mathbf{Y})\]:\mathcal{D}\left(e,e^{*}\right)\leq\rho\rbrace
> $$
>
> where $f$ is the function we optimized, usually a deep neural network with learnable parameters. $\mathcal{D}\left(\cdot,\cdot\right)$ is the distribution distance metric, which is used to calculate the distance between distributions. $e^*$ is the training environment covering multiple data distributions. $\rho$ is a hyperparamer.
>
> **Mark**. This equation tell me if an optimization satisfies: (1)  Optimize in diverse traing environments, (2) these environments are constrained, and (3) emphasizing the challenging environments for learning, then this optimization belongs to the distributionally robust optimization  and possesses the following beneficial properties.
>
> > **Properties of distributionally robust optimization**
>
> Compared to the IID-only condition of the Empirical Risk Minimisation, distributionally robust optimization explores a certain range of challenging training data distributions, mathematically, distributionally robust optimization is equivalent to adding variance regularization to the standard Empirical Risk Minimisation [1],
>
> $$
> \begin{align}
> &\sup\_\{e\in\mathcal{E}\}\{\lbrace\mathbb{E}\_\{\(\mathbf{X}\,\mathbf{Y}\)\sim p\(\mathcal{X}\,\mathcal{Y}\|e\)\}\[\mathcal{L}\(f\(\mathbf{X})\,\mathbf{Y}\)\]\:\mathcal{D}\(e\,e\^{*}\)\leq\rho\rbrace} \\\\
> \geq~&\mathbb{E}\_\{\(\mathbf{X}\,\mathbf{Y}\)\sim p\(\mathcal{X}\,\mathcal{Y}\|e\^\*\)\} \[\mathcal{L}\(f\(\mathbf{X}\)\,\mathbf{Y}\)\] \+ \sqrt{2 \rho \text{Var}\_\{\(\mathbf{X}\,\mathbf{Y})\sim p\(\mathcal{X}\,\mathcal{Y}|e^\*\)\}\[\mathcal{L}\(f\(\mathbf{X}\)\,\mathbf{Y}\)\]\}.
> \end{align}
> $$
>
> Compared to standard empirical loss functions, variance regularization has stricter bounds. Therefore, distributionally robust optimization class mathematically provides more rigorous constraints than using empirical loss functions alone in OOD environments, preventing the model from over-relying on training data [1]. This enables the model to flexibly adapt to different environments, improving its generalization performance in unknown environments.
>
> > **STOP as number of distributionally robust optimization class**
>
> We will demonstrate that our optimization objective of STOP belongs to distributionally robust optimization class, inheriting its good properties. Our optimization objective of STOP is as follows:
> $$
> \min_f\sup_{\boldsymbol{g}\in\mathbb{R}^N}\mathbb{E}_{(\mathbf{X},\mathbf{Y})\sim(\mathcal{X},\mathcal{Y}|e^*)}\left[\mathcal{L}\left(f\left(\mathbf{X}\right),\mathbf{Y};\boldsymbol{g}\right)\right],\quad\mathrm{s.t.} ||\widetilde{\boldsymbol{g}}||_0=s\in\left(0,N\right).
> $$
> In fact, this goal is aligned with the necessary conditions of distributionally robust optimization class explaned in Mark.
>
> - **Diverse traing environments**. STOP creates a diverse training environment by adding perturbations through a graph perturbation mechanism.
>
> - **Applying constraints for the generation of environment**. Our perturbation process follows polynomial distribution sampling, and we strictly control the perturbation ratio, which imposes constraints on the generated environments.
>
> - **Exploring challenging environments**. We select the environment with the largest gradients during training for optimization, encouraging the model to be exposed to challenging environments.
>
> In summary, our optimization strategy belongs to distributionally robust optimization, and therefore inherits its good generalization properties. That is, our optimization strategy can theoretically achieve better model generalization performance.
>
> Ref:
>
> [1] John Duchi and Hongseok Namkoong. Variance-based regularization with convex objectives. Jour-
> nal of Machine Learning Research, 20(68):1–55, 2019.

---

> ### Author Response · Authors · 2024-11-21
> **W3. Discussion regarding the potential limitations**
>
> Thank you very much for your suggestions. We acknowledge the following potential limitations, which may serve as valuable directions for future research：
>
> -	**Validating the Broad Impact of STOP**. The spatial interaction module integrated within the STOP framework is inherently generic, suggesting its potential for broader applicability. In upcoming research, we will propose replacing the graph convolutional networks utilized by other spatiotemporal backbones with the spatial interaction module to validate its effectiveness across various contexts. This initiative will help us better understand the potential value and applicability of the STOP module in a wide range of application domains.
>
> -	**Exploring a Wider Range of OOD Scenarios**. Current OOD problems are typically defined within the confines of single-modal data and single tasks. However, spatiotemporal data exhibits diverse modalities and varied tasks. We believe that an improved spatiotemporal OOD handler should be capable of addressing challenges such as cross-task and cross-modal processing, areas that have not been thoroughly explored in the spatiotemporal domain.
>
> -	**Integrating Large Language Models for zero-shot learning**. In OOD settings, accurately predicting new nodes poses a significant challenge, as these nodes have not been encountered by the model during training—commonly referred to as the zero-shot challenge. Large language models excel in this context, as their representational capabilities, developed from extensive training on massive datasets, can enhance a model's zero-shot learning ability. While this has been successfully demonstrated in the time series community, it remains relatively unexplored within the spatiotemporal domain. In future work, we plan to integrate large language models into the STOP framework to further enhance its scalability for predicting new nodes.
>
> We have added this discussion to Appendix Section D.

---

> > ### Comment · Reviewer_LDYw · 2024-11-24
> >
> > Thanks for the authors' detailed response, which has addressed most of my concerns. I'd like to raise my score.

---

> > > ### Author Response · Authors · 2024-11-26
> > >
> > > Dear Reviewer LDYw,
> > >
> > > We sincerely appreciate your acknowledgment and encouraging feedback. Interacting with you has been both enjoyable and invaluable, significantly enhancing the quality of our paper. Thank you once again for your time, effort, and insightful comments.
> > >
> > > Warm regards,
> > >
> > > The authors.

---

### Official Review · Reviewer_BojP · 2024-11-04

**Soundness:** 4
**Presentation:** 3
**Contribution:** 3
**Rating:** 6
**Confidence:** 3

**Summary:**

The paper introduces STOP (Spatio-Temporal Out-of-Distribution Processor), a model designed to enhance the robustness of spatiotemporal predictions in out-of-distribution (OOD) environments. Traditional spatiotemporal graph convolutional networks (STGNNs) often struggle with OOD scenarios due to their reliance on node-to-node messaging, which is highly sensitive to shifts in temporal data distributions or structural changes in graph networks. STOP overcomes these limitations by integrating a novel client-server (C2S) messaging mechanism and a graph perturbation strategy that simulates various training conditions.

**Strengths:**

1. The client-server messaging approach with Context Perception Units is a clever departure from traditional node-to-node interactions, reducing sensitivity to structural shifts and enhancing adaptability in OOD settings.
2.  STOP’s design primarily relies on lightweight MLP layers, making it more computationally efficient than complex STGNNs or Transformer-based models, which often require high processing power.
3. The model was thoroughly tested across a variety of datasets, demonstrating not only generalization but also strong performance in both temporal and spatial OOD scenarios, enhancing the model’s credibility.
4. The ablation studies are thorough and convincing.

**Weaknesses:**

See the questions listed below.

**Questions:**

1. Adding a paragraph discussing the research motivation and broader impacts of STOP could strengthen the work, particularly in terms of situating the work within the broader context of spatiotemporal OOD challenges. Additionally, the positioning of Figure 1 alongside the introductory text makes the flow slightly disjointed. A more developed introduction with a refined layout could improve the paper’s readability and help frame the significance of STOP’s contributions.

2. While the authors present the CPUs, GPUs, and DRO as core mechanisms in STOP, the paper could benefit from a more in-depth analysis of why each component specifically enhances spatiotemporal OOD detection. A theoretical analysis of how the graph perturbation strategy affects the model's ability to learn robust representations would provide a more solid grounding for their effectiveness and make the claims more convincing.

3. What's the potential limitation of STOP? The authors may add a section to discuss potential trade-offs or scenarios where STOP might not perform as well compared to other approaches, which would provide a more comprehensive evaluation of the model's capabilities and limitations.

---

> ### Author Response · Authors · 2024-11-21
> **Clarification on W1**
>
> Dear Reviewer BojP,
>
> Thank you very much for your valuable comments, which are crucial to the improvement of our paper. We would clarify your concerns point by point.
>
>
>  >  **W1. (a) Anohter paragraph for discussion**
>
> Based on your suggestions, we added a discussion paragraph in Appendix D to further emphasize our motivation and highlight the impact of our method on OOD spatiotemporal learning. For your convenience, the added paragraph is as follows:
>
> - The effectiveness of traditional spatiotemporal prediction models is typically demonstrated only in test environments very similar to the training environment. While some research on spatiotemporal OOD challenges has recognized the problems caused by distribution changes due to spatiotemporal variations and proposed various strategies, both traditional models and OOD learning models' reliance on node-to-node global interaction mechanisms limits their generalization performance in the face of spatiotemporal changes, as these interaction mechanisms are coupled with the training graph structure and flood representation errors. To address this inherent limitation, we introduce an innovative spatio-temporal interaction mechanism to replace traditional node-to-node approaches. This new mechanism incorporates CPU units that can sense contextual features from nodes, helping maintain high generalization in unknown environments. Additionally, we designed graph perturbation mechanisms to further enhance robustness. Our approach has been validated across eight OOD datasets, showing a +17.01% performance improvement.
>
> - More importantly, our research findings provide valuable insights for future OOD researchers: (1) The core message passing mechanisms in GCN and Transformer are limited in OOD scenarios, suggesting the need to explore alternatives beyond traditional GCN/Transformer sequential model architectures; (2) Existing models follow a stacked architecture, which lacks flexibility for OOD scenarios; (3) Lightweight but powerful architectures (such as MLPs) may be more suitable for OOD learning, as complex GCN or Transformer advanced backbones might overfit to the training environment and compromise their generalization ability.
>
> > **W1 (b). Figure 1 Layout**
>
> Thank you for your suggestion. In the revised version, we modified the layout of Figure 1, placing it at the connection point between the two paragraphs to avoid breaks in the reading flow.

---

> > ### Comment · Reviewer_BojP · 2024-11-26
> >
> > Thank you for your clarification and thoughtful responses. They have addressed my concerns, and I will maintain my current score.

---

> > > ### Author Response · Authors · 2024-11-26
> > >
> > > Dear Reviewer BojP,
> > >
> > > We sincerely thank you for your recognition and constructive feedback. Our interaction with you has been both enjoyable and invaluable, greatly enhancing the quality of our paper. Thank you once again for your time, effort, and insightful comments.
> > >
> > > Best regards,
> > >
> > > The Authors

---

> ### Author Response · Authors · 2024-11-21
> **W2. Robustness analysis of each component.**
>
> > **(1) CPUs robustness analysis**
>
> **We explained the robustness of the CPUs for OOD shifts in a single paragraph from lines 241 to 249 of the manuscript.** This has been highlighted in red in the revised manuscript, and we have reproduced it below for your convenience:
>
> - The proposed C2S messaging mechanism is constrained to operate between nodes and CPUs, effectively avoiding the complexity associated with direct node-to-node interactions. CPUs in this mechanism assimilates contextual features, which is used to generate output representations for individual nodes. These features are coarse-grained and high-level, which exhibits resilience to temporal variations for individual nodes. Furthermore, structural changes (such as adding or removing nodes) do not significantly disrupt the message-passing pathways between nodes and CPUs. New nodes can also leverage these contextual features to develop information-rich representations, thereby enhancing inductive learning capabilities. In summary, our approach demonstrates remarkable resilience to spatiotemporal variations and strong in out-of-distribution (OOD) environments.
>
> > **(2) GPUs robustness analysis**
>
> The GPUs are used to perturb the CPUs perceptual nodes during the feature extraction process. Essentially, these perturbations simulate a diversified training environment characterized by spatiotemporal shifts. By exposing the model to these varied training environments, it can better perceive invariant contextual features that generalize effectively to unknown environments.
>
> >**(3) Spatiotemporal DRO robustness analysis**
>
> The spatiotemporal DRO is specifically designed to enable the optimization process of the GPUs. It prunes relatively simple environments for model and directs the model to optimize towards the most challenging environments along the steepest gradient. This focus on challenging environments is advantageous, as it enables the model to extract more robust features that contribute to improved performance in OOD scenario.
>
> As GPUs and DRO serve as the foundational methods of our proposed graph perturbation and are utilized in tandem, we have also included a dedicated paragraph in Section 4.5 of the revised manuscript to underscore the robustness of the proposed graph perturbation mechanism.  The content is as follows:
> - The GPUs introduces perturbations in the spatial interaction process, effectively generating diversified training environments. This strategy prevents the model from becoming overly reliant on a single training environment, thereby promoting the learning of more generalizable features. And the incorporation of spatiotemporal DRO compels the model to engage with the most challenging instances within the generated environments, which can exposure further enhances the model's robustness.

---

> ### Author Response · Authors · 2024-11-21
> **W2 (2). Theoretical analysis and explanation.**
>
> We theoretically analyzed STOP's generalization performance. Since STOP's optimization objective belongs to the distributionally robust optimization class, which exhibits good generalization properties. Note that distributionally robust optimization class is a general term for optimization objectives that satisfy specific conditions - our contribution lies in how to implement optimization strategies that meet these conditions in the spatiotemporal OOD problem. First, we will introduce what constitutes a distributionally robust optimization class and the necessary conditions for membership, then analyze its beneficial properties, and finally extend these concepts to STOP.
>
> > **What is distributionally robust optimization class?**
>
> Distributionally robust optimization [1] refers to a class of optimization objectives that aims to optimize for the worst-case scenario within a certain range of all possible data distributions, and the mathematical formula is expressed as:
> $$
> \arg\min_{f}\sup_{e\in\mathcal{E}}\lbrace\mathbb{E}_{(\mathbf{X},\mathbf{Y})\sim p(\mathcal{X},\mathcal{Y}|e)}\[\mathcal{L}(f(\mathbf{X}),\mathbf{Y})\]:\mathcal{D}\left(e,e^{*}\right)\leq\rho\rbrace
> $$
>
> where $f$ is the function we optimized, usually a deep neural network with learnable parameters. $\mathcal{D}\left(\cdot,\cdot\right)$ is the distribution distance metric, which is used to calculate the distance between distributions. $e^*$ is the training environment covering multiple data distributions. $\rho$ is a hyperparamer.
>
> **Mark**. This equation tell me if an optimization satisfies: (1)  Optimize in diverse traing environments, (2) these environments are constrained, and (3) emphasizing the challenging environments for learning, then this optimization belongs to the distributionally robust optimization  and possesses the following beneficial properties.
>
> > **Properties of distributionally robust optimization**
>
> Compared to the IID-only condition of the Empirical Risk Minimisation, distributionally robust optimization explores a certain range of challenging training data distributions, mathematically, distributionally robust optimization is equivalent to adding variance regularization to the standard Empirical Risk Minimisation [1],
>
> $$
> \begin{align}
> &\sup\_\{e\in\mathcal{E}\}\{\lbrace\mathbb{E}\_\{\(\mathbf{X}\,\mathbf{Y}\)\sim p\(\mathcal{X}\,\mathcal{Y}\|e\)\}\[\mathcal{L}\(f\(\mathbf{X})\,\mathbf{Y}\)\]\:\mathcal{D}\(e\,e\^{*}\)\leq\rho\rbrace} \\\\
> \geq~&\mathbb{E}\_\{\(\mathbf{X}\,\mathbf{Y}\)\sim p\(\mathcal{X}\,\mathcal{Y}\|e\^\*\)\} \[\mathcal{L}\(f\(\mathbf{X}\)\,\mathbf{Y}\)\] \+ \sqrt{2 \rho \text{Var}\_\{\(\mathbf{X}\,\mathbf{Y})\sim p\(\mathcal{X}\,\mathcal{Y}|e^\*\)\}\[\mathcal{L}\(f\(\mathbf{X}\)\,\mathbf{Y}\)\]\}.
> \end{align}
> $$
>
>
> Compared to standard empirical loss functions, variance regularization has stricter bounds. Therefore, distributionally robust optimization class mathematically provides more rigorous constraints than using empirical loss functions alone in OOD environments, preventing the model from over-relying on training data [1]. This enables the model to flexibly adapt to different environments, improving its generalization performance in unknown environments.
>
> > **STOP as number of distributionally robust optimization class**
>
> We will demonstrate that our optimization objective of STOP belongs to distributionally robust optimization class, inheriting its good properties. Our optimization objective of STOP is as follows:
> $$
> \min_f\sup_{\boldsymbol{g}\in\mathbb{R}^N}\mathbb{E}_{(\mathbf{X},\mathbf{Y})\sim(\mathcal{X},\mathcal{Y}|e^*)}\left[\mathcal{L}\left(f\left(\mathbf{X}\right),\mathbf{Y};\boldsymbol{g}\right)\right],\quad\mathrm{s.t.} ||\widetilde{\boldsymbol{g}}||_0=s\in\left(0,N\right).
> $$
> In fact, this goal is aligned with the necessary conditions of distributionally robust optimization class explaned in Mark.
>
> - **Diverse traing environments**. STOP creates a diverse training environment by adding perturbations through a graph perturbation mechanism.
>
> - **Applying constraints for the generation of environment**. Our perturbation process follows polynomial distribution sampling, and we strictly control the perturbation ratio, which imposes constraints on the generated environments.
>
> - **Exploring challenging environments**. We select the environment with the largest gradients during training for optimization, encouraging the model to be exposed to challenging environments.
>
> In summary, our optimization strategy belongs to distributionally robust optimization, and therefore inherits its good generalization properties. That is, our optimization strategy can theoretically achieve better model generalization performance.
>
> Ref:
>
> [1] John Duchi and Hongseok Namkoong. Variance-based regularization with convex objectives. Jour-
> nal of Machine Learning Research, 20(68):1–55, 2019.

---

> ### Author Response · Authors · 2024-11-22
> **W3. Discussion regarding the potential limitations**
>
> Thank you very much for your advice. We acknowledge the following potential limitations, which may serve as valuable directions for future research：
>
> -	**Validating the Broad Impact of STOP**. The spatial interaction module integrated within the STOP framework is inherently generic, suggesting its potential for broader applicability. In upcoming research, we will propose replacing the graph convolutional networks utilized by other spatiotemporal backbones with the spatial interaction module to validate its effectiveness across various contexts. This initiative will help us better understand the potential value and applicability of the STOP module in a wide range of application domains.
>
> -	**Exploring a Wider Range of OOD Scenarios**. Current OOD problems are typically defined within the confines of single-modal data and single tasks. However, spatiotemporal data exhibits diverse modalities and varied tasks. We believe that an improved spatiotemporal OOD handler should be capable of addressing challenges such as cross-task and cross-modal processing, areas that have not been thoroughly explored in the spatiotemporal domain.
>
> -	**Integrating Large Language Models for zero-shot learning**. In OOD settings, accurately predicting new nodes poses a significant challenge, as these nodes have not been encountered by the model during training—commonly referred to as the zero-shot challenge. Large language models excel in this context, as their representational capabilities, developed from extensive training on massive datasets, can enhance a model's zero-shot learning ability. While this has been successfully demonstrated in the time series community, it remains relatively unexplored within the spatiotemporal domain. In future work, we plan to integrate large language models into the STOP framework to further enhance its scalability for predicting new nodes.
>
> We have added this discussion to Appendix Section D.

---

### Meta-Review · Area_Chair_8L8b · 2024-12-17

**Metareview:**

This submission proposes the Spatio-Temporal OOD Processor (STOP) to address spatiotemporal graph convolutional networks' out-of-distribution (OOD) challenges by leveraging spatiotemporal MLP channel mixing to separate spatial and temporal elements for prediction. STOP incorporates a centralized messaging mechanism with Context-Aware Units to extract generalizable features and a graph perturbation mechanism with Generalized Perturbation Units to create diverse training environments, enhancing robustness. A customized distributionally robust optimization (DRO) further improves generalization, achieving competitive results on six datasets.

Extensive experiments on various datasets are conducted which are acknowledged by most reviewers. Also, the paper's organization particularly for the ablation section clearly conveys all the findings. This submission also focuses on one of the important questions about addressing the spatiotemporal shifts for graph learning. Most weaknesses lie in sufficient in-depth analysis of different proposed components and their separate impact on addressing domain shifts. Most of the proposed components are customized from existing works, which leads to certain concerns about the innovation.

It is really a borderline case. Although we recommend a rejection this time, this submission could be stronger by incorporating all discussions during the rebuttal period.

**Additional Comments On Reviewer Discussion:**

The authors did a great job and lots of effort during the rebuttal period, which addressed the majority of concerns of reviewers in terms of more experiments, more analysis, and more innovation/insight justifications. It raises another concern that the extensive new experiments and analysis in rebuttal seem to take a major portion of the paper revision. In this case, a resubmission with all new content is more encouraged.

---

### Decision · Program_Chairs · 2025-01-22

Reject